# Better calibration of cloud parameterizations and subgrid effects increases the fidelity of E3SM Atmosphere Model version 1

Po-Lun Ma[1], Bryce E. Harrop[1], Vincent E. Larson[2,1], Richard Neale[3], Andrew Gettelman[3], Hugh Morrison[3], Hailong Wang[1], Kai Zhang[1], Stephen A. Klein[4], Mark D. Zelinka[4], Yuying Zhang[4], Yun Qian[1], Jin-Ho Yoon[5], Christopher R. Jones[1], Meng Huang[1], Sheng-Lun Tai[1], Balwinder Singh[1], Peter A. Bogenschutz[4], Xue Zheng[4], Wuyin Lin[6], Johannes Quaas[7], Hélène Chepfer[8], Michael A. Brunke[9], Xubin Zeng[9], Johannes Mülmenstädt[1], Samson Hagos[1], Zhibo Zhang[10], Hua Song[11], Xiaohong Liu[12], Hui Wan[1], Jingyu Wang[1], Qi Tang[4], Peter M. Caldwell[4], Jiwen Fan[1], Larry K. Berg[1], Jerome D. Fast[1], Mark A. Taylor[13], Jean-Christophe Golaz[4], Shaocheng Xie[4], Philip J. Rasch[1], L. Ruby Leung[1]

[1]Pacific Northwest National Laboratory, Richland, Washington, USA
[2]Department of Mathematical Sciences, University of Wisconsin-Milwaukee, Milwaukee, Wisconsin, USA
[3]National Center for Atmospheric Research, Boulder, Colorado, USA
[4]Lawrence Livermore National Laboratory, Livermore, California, USA
[5]School of Earth Sciences and Environmental Engineering, Gwangju Institute of Science and Technology, Gwangju, South Korea
[6]Brookhaven National Laboratory, Upton, New York, USA.
[7]Institute for Meteorology, Universität Leipzig, Leipzig, Germany
[8]LMD/IPSL, Sorbonne Université, Ecole Polytechnique, CNRS, Paris, France
[9]Department of Hydrology and Atmospheric Sciences, University of Arizona, Tucson, AZ, USA
[10]Department of Physics, University of Maryland, Baltimore County, Baltimore, MD, USA
[11]Science Systems and Applications, Inc., Lanham, Maryland, USA
[12]Department of Atmospheric Sciences, Texas A&M University, College Station, TX, USA.
[13]Sandia National Laboratory, Albuquerque, NM, USA,

*Correspondence to*: Po-Lun Ma (Po-Lun.Ma@pnnl.gov)

**Abstract.** Realistic simulation of the Earth's mean state climate remains a major challenge and yet it is crucial for predicting the climate system in transition. Deficiencies in models' process representations, propagation of errors from one process to another, and associated compensating errors can often confound the interpretation and improvement of model simulations. These errors and biases can also lead to unrealistic climate projections as well as incorrect attribution of the physical mechanisms governing the past and future climate change. Here we show that a significantly improved global atmospheric simulation can be achieved by focusing on the realism of process assumptions in cloud calibration and subgrid effects using the Energy Exascale Earth System Model (E3SM) Atmosphere Model version 1 (EAMv1). The calibration of clouds and subgrid effects informed by our understanding of physical mechanisms leads to significant improvements in clouds and precipitation climatology, reducing common and longstanding biases across cloud regimes in the model. The improved cloud fidelity in turn reduces biases in other aspects of the system. Furthermore, even though the recalibration does not change the global mean aerosol and total anthropogenic effective radiative forcings (ERFs), the sensitivity of clouds, precipitation, and

surface temperature to aerosol perturbations is significantly reduced. This suggests that it is possible to achieve improvements to the historical evolution of surface temperature over EAMv1 and that precise knowledge of global mean ERFs is not enough to constrain historical or future climate change. Cloud feedbacks are also significantly reduced in the recalibrated model, suggesting that there would be a lower climate sensitivity when running as part of the fully coupled E3SM. This study also compares results from incremental changes to cloud microphysics, turbulent mixing, deep convection, and subgrid effects to understand how assumptions in the representation of these processes affect different aspects of the simulated atmosphere as well as its response to forcings. We conclude that the spectral composition and geographical distribution of the ERFs and cloud feedback as well as the fidelity of the simulated base climate state are important for constraining the climate in the past and future.

## 1 Introduction

The Energy Exascale Earth System Model (E3SM) version 1 (E3SMv1) (Golaz et al., 2019;Caldwell et al., 2019) includes an atmospheric component called the E3SM atmosphere model (EAM) version 1 (EAMv1) (Rasch et al., 2019). EAMv1 was released in April 2018 together with the fully coupled E3SMv1 and all of its model components. EAMv1 uses a revised 4-mode version of the modal aerosol module (MAM)(Liu et al., 2012;Liu et al., 2016;Wang et al., 2020); an updated 2-moment cloud microphysics scheme (Gettelman and Morrison, 2015;Gettelman et al., 2015) (hereafter MG2); the Cloud Layers Unified By Binormals (CLUBB) parameterization (Golaz et al., 2002;Larson et al., 2002;Larson and Golaz, 2005;Bogenschutz et al., 2013) for turbulence, shallow convection, and cloud macrophysics, the Zhang and McFarlane (1995) (hereafter ZM) parameterization for deep convection with the addition of convective momentum transport (Richter and Rasch, 2008) and a modified dilute plume calculation (Neale et al., 2008). The model shows general success in simulating the present-day climatology, producing improved simulation compared to atmospheric simulations of previous-generation Earth system models (ESMs) (Rasch et al., 2019) that participated in the Coupled Model Intercomparison Project (CMIP) phase 5 (CMIP5)(Taylor et al., 2012).

However, EAMv1 still produces significant regional cloud and precipitation biases that are common in many ESMs (Zhang et al., 2019a;Xie et al., 2018;Brunke et al., 2019). These persistent errors include the underestimation of coastal stratocumulus (Sc), overly bright trade cumulus (Cu), mislocations of the Sc-to-Cu transitions, and a notable underestimate of the areal extent of clouds over the Indo-Pacific warm pool. EAMv1 also showed some new cloud biases compared to its predecessors, including overly bright clouds embedded within storm tracks and an unrealistically high liquid water path (LWP) in polar regions (Zhang et al., 2020). Closely related to these errors are biases in the mean, variability, and extremes of precipitation. As shown in Rasch et al. (2019), EAMv1 produces high annual mean precipitation over the global average, in high elevation regions, and in the central Pacific, but low annual mean precipitation over Amazonia and the tropical western Pacific (TWP). EAMv1 contains the signature of a double intertropical convergence zone (ITCZ) that has been problematic in ESMs for over

two decades (Mechoso et al., 1995;Dai, 2006). Furthermore, similar to many other coarse resolution models, EAMv1 produces too many light precipitation events and too few heavy precipitation events compared to observations (Stephens et al., 2010). The diurnal cycle of precipitation over regions that are strongly influenced by mesoscale convective systems (MCSs) is skewed, producing peak precipitation in the mid-day instead of late afternoon to early morning (Xie et al., 2019). These common and persistent biases in predictions of clouds and precipitation arise from coarse model resolution insufficient to represent small scale features as well as various deficiencies in parameterizations of cloud, turbulence, and convection processes. These deficiencies can, in turn, adversely affect other aspects of the atmosphere.

EAMv1 also shows large biases in the simulated present-day climatology of surface temperature and winds, similar to other global model predictions (Morcrette et al., 2018). These biases pose challenges for the fully coupled E3SMv1 to produce credible projections of the future climate. As discussed in Golaz et al. (2019), E3SMv1 appears very sensitive to perturbations of atmospheric composition (aerosols and greenhouse gases), producing differences in the observed and simulated temporal evolution of the global mean surface temperature in the 20$^{th}$ century and a relatively high estimate of equilibrium climate sensitivity (ECS) of 5.3 K compared to other ESMs.

Many factors may contribute to the behavior and biases of the model. Biases affect the interpretation of climate projections and future model development plans. The choice of parameter settings for parameterizations is a scientifically important factor in creating (and reducing) these biases. This study explores the impact of changes to parameter settings (i.e., recalibration) to improve fidelity of model climate, and implications for climate change studies. Hence, this recalibration effort can provide important physical insights into future development of E3SM as well as other ESMs.

Model calibration, or tuning, is a crucial research element in Earth system modeling. This procedure optimizes model fidelity by addressing the trade-off between optimizing individual processes and process interactions so that the model climate agrees with observables while simultaneously satisfying energy balance requirements. These multiple constraints frequently expose the presence of error compensations in ESMs. As discussed in depth in Hourdin et al. (2017) and Schmidt et al. (2017), balancing these requirements is a mix of art and science because some degree of subjectivity is inevitable and choices are made based on expert judgement. Expert judgement consists of evaluation, intercomparison, and interpretation of results. This is followed by changes to the model parameter settings to make the model better suited for answering specific science questions that originally motivated its development. During the development of EAMv1, model calibration used primarily the traditional one-at-a-time parameter adjustment approach (Rasch et al., 2019;Xie et al., 2018). In principle, automated procedures could be employed to perform such calibrations, but they are not yet used for final calibrations (for reasons discussed below). Instead, automated procedures have been performed for an ensemble of short simulations with perturbed physics to provide a systematic assessment of the parametric sensitivity (Rasch et al., 2019;Qian et al., 2018), helping to provide insight about multi-variate responses of the model to changes in single or multiple parameters.

The traditional one-at-a-time parameter adjustment approach is inefficient and expensive in terms of both computational and human resources (Zhang et al., 2012). It is a sequential and iterative process that requires a large number (e.g., hundreds) of iterations consisting of 1) running a multi-year simulation; 2) performing a comprehensive evaluation using diagnostics packages to assess the impact of the change of a single parameter value on different aspects of the simulation; and 3) designing and running the next simulation based on evaluation of the current simulation. However, there exist too many uncertain parameters within a climate model to repeat this process and perfectly optimize its climate fidelity.

The perturbed physics ensemble approach (Murphy et al., 2004) has been used for quantifying parametric uncertainty. The EAMv1 development team adopted the short simulation ensemble approach (Wan et al., 2014;Qian et al., 2018) which uses 5-day simulations rather than multi-year simulations to assess the fast physics. The approach significantly reduces the turn-around time and computational cost compared to the traditional multi-year simulation ensemble approach for a systematic assessment of the parametric uncertainty. One caveat, however, is that it requires *a-priori* knowledge of a manageable set of uncertain parameters and their physically, observationally, or empirically justifiable ranges. The parameter space is also too large to explore fully, and only a subset of parameters are typically selected based on physical intuition and expert judgement. In hindsight, the parameter set selected for the short simulation ensemble during the EAMv1 development was insufficient because parameters not included in the original ensemble were later found to be important. Another limitation is that the short simulations focus on fast physical processes and rapid adjustments. By design, important factors such as slow internal variability of the atmosphere and circulation feedbacks are not considered, so any conclusion drawn from the short simulation ensemble might not be applicable to the calibration of the ESM for climate simulations. Both limitations could be mitigated if the perturbed physics ensemble includes every possible combination of parameter choices and the simulations were decade-long, but the amount of computational resource for such an exercise is prohibitive.

The one-at-a-time and the short simulation ensemble approaches are complementary to each other, but for the purpose of tuning EAMv1, both approaches shared some common challenges: 1) insufficient computational and human resources to explore and optimize parameter choices; 2) insufficient time to perform and analyze the simulations; and 3) improvements to one aspect of the simulation in general may be made at the price of degradation in other aspects, suggesting model structural deficiency in addition to parametric uncertainty (Qian et al., 2018). Reconciling these contradictory results and further improving the model fidelity have been great challenges for the model development team.

In this study, we experimented with a different strategy to further reduce the various model biases mentioned above through recalibration. Instead of optimizing the model for more than a dozen of the metrics that the community typically relies on (Burrows et al., 2018;Hourdin et al., 2017;Mauritsen et al., 2012), we focused solely on clouds. We find that when clouds in every regime are improved, other aspects of the global atmospheric simulation are also improved, even though they are not the

direct targets for calibration. Interestingly, the recalibrated atmosphere model, denoted as EAMv1P, shows lower cloud and precipitation sensitivity to aerosol perturbation and to surface warming. This suggests that EAMv1P may lead to improvements to the simulation of the 20th century temperature evolution and a lower estimate of ECS when running as part of the fully coupled E3SM. More challenges may yet emerge in tuning fully coupled models.

We acknowledge that our recalibration approach has several caveats. First, like all current model calibration strategies, our recalibration does not lead to a unique and perfect configuration and there can be other ways to achieve a different model configuration with equally good present-day climate. We also acknowledge that there may be complications when the recalibrated atmosphere model is coupled with the ocean. Additional tuning might be required. However, the experience from this study will likely be valuable in that effort. Finally, we acknowledge that some tuning choices are better justified than others, because many of the uncertain parameters do not have a physically or observationally justifiable range. For those poorly constrained processes, the recalibration provides a way to identify the important process assumptions that affect our ability to accurately simulate the climate system. Future studies that develop theoretical or observational constraints to reduce the uncertainties associated with these process assumptions will be very valuable.

In Section 2, we provide a discussion on the recalibration. Section 3 shows the results from the recalibrated model. We draw conclusions in Section 4.

## 2 Approach

Because clouds in different regimes are governed by different processes, the recalibration first treats each regional cloud bias separately, followed by adjustments (including sea salt and dust emission factors) to refine the cloud climatology and to restore the top-of-atmosphere (TOA) energy balance. The TOA cloud radiative effects (CREs) are the primary tuning target but other cloud properties and cloud controlling factors are also assessed. We adopted the one-at-a-time parameter adjustment approach. Adjustments of uncertain parameters were driven by analysis of physical mechanisms affecting the simulation in every cloud regime. We also introduced new parameters for controlling the coupling of subgrid effects between the convection, turbulence, and surface flux parameterizations to produce better simulation of clouds. The recalibration is described in detail in this section.

### 2.1 Tropical clouds

Tropical clouds and precipitation are primarily controlled by the deep convection parameterization and ice cloud microphysics. They interact strongly with the atmospheric circulation in the tropics through their overturning and vertical mixing of moist static energy. In EAMv1, cloud cover is significantly underestimated in the TWP and the eastern Pacific. Precipitation is biased low in the TWP and over the Amazon, and biased high in the central Pacific, indicating a displacement of the Walker

circulation. These biases also reflect errors in the simulated Hadley cell, moderating subsidence in the subtropics and the distribution of stratocumulus and trade cumulus. To improve the tropical clouds and precipitation, we include the subgrid wind and temperature variance in surface flux and ZM's parcel buoyancy calculations to improve the spatial distribution of cloud and precipitation, followed by parameter adjustments to keep the magnitude of tropical CREs and precipitation within a reasonable range, as described in this section.

Harrop et al. (2018) showed that including the Redelsperger et al. (2000) gustiness effects associated with deep convection over ocean increases local surface fluxes in EAMv1 running at ~ 1-degree horizontal grid spacing. The circulation responses significantly improve clouds and precipitation over the TWP. This is because E3SMv1 uses the Large and Pond (1982) and Zeng et al. (1998) parameterizations for surface fluxes of heat, moisture, and momentum over ocean and land, respectively, and these bulk aerodynamic schemes are prone to underestimation of surface fluxes in regions where 1) large-scale winds are weak; and 2) convective episodes are frequent. Enabling gustiness effects increases surface fluxes in those regions, and, hence, increases clouds and precipitation.

The gustiness effects associated with deep convection wasn't ready in time to be included in the E3SMv1 release because including the gustiness effects requires retuning of the model. In this study, we built on the success of Harrop et al. (2018) and extended the Redelsperger et al. (2000) parameterization to operate over both land and ocean. To account for the gustiness effects associated with shallow convection and turbulence, the subgrid wind variance predicted by CLUBB was passed to the surface flux calculations. The total wind speed used for surface flux computation is expressed as

$$U^2 = U_0^2 + a_g \cdot U_{g(ZM)}^2 + b_g \cdot U_{g(CLUBB)}^2$$

where U is the total wind speed, $U_0$ is the resolved large-scale wind speed, and $U_{g(ZM)}$ and $U_{g(CLUBB)}$ are the wind speed enhancements owing to the gustiness associated with ZM and CLUBB, respectively. The use of the Redelsperger et al. (2000) parameterization over land is meant as a simple approximation to incorporate a consistent gustiness treatment globally until more targeted studies of gustiness impacts over land are made into a suitable alternative parameterization. Parameters $a_g$ and $b_g$ are tunable parameters used for calibrating the spatial distribution of surface fluxes. The $a_g$ parameter can be set to different values to account for the difference in surface roughness and to provide the flexibility to adjust the model in the face of the structural uncertainty of this parameterization. Based on sensitivity tests, we set $a_g$ to 0.9 over ocean and 1.2 over land and $b_g$ to 1.5 both over land and ocean.

Figure 1 shows that the gustiness associated with the ZM deep convection parameterization contributes about 15% to the total surface wind speed over tropical ocean, and up to 45% over tropical land, and gustiness associated with the shallow

convection and turbulence parameterization CLUBB accounts for 10-30% of the total surface wind speed globally. Therefore, including gustiness effects significantly increases surface fluxes (of sensible heat, moisture, and momentum) in these regions.

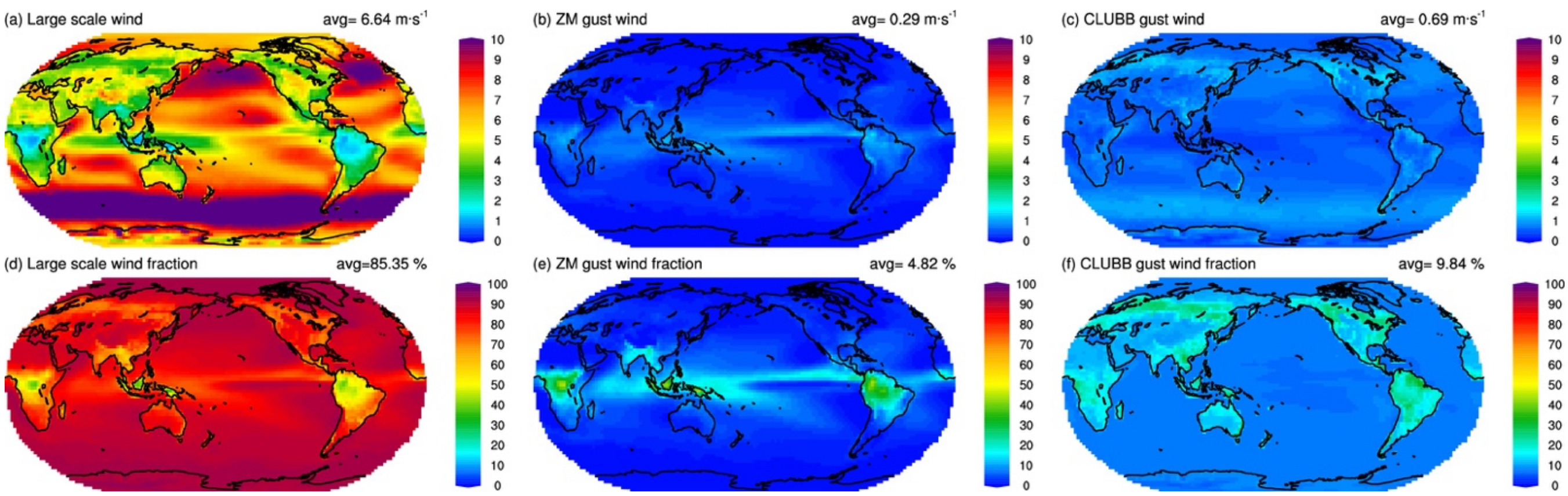

**Figure 1.** Present-day EAMv1 climatology of wind speed (m $s^{-1}$) at the lowest model level in (a) resolved motion; (b) gustiness associated with the ZM parameterization; and (c) gustiness associated with the CLUBB parameterization. (d-f) are the fractional contribution of the three components to the total wind speed.

Next, we considered the subgrid temperature perturbation in the parcel buoyancy calculation in the ZM scheme. The subgrid temperature perturbation is set to 0.5 K in the Community Atmosphere Model version 5 (CAM5) (Neale et al., 2010) and 0.8 K in EAMv1 (Rasch et al., 2019). This treatment assumes that the subgrid heterogeneity of temperature is globally uniform. However, subgrid variability of temperature should vary in space and time. In particular, subgrid temperature heterogeneity is typically larger over land than over ocean. Setting a globally uniform subgrid temperature perturbation can potentially create biases in the distribution of deep convection. To address this deficiency, we computed the subgrid temperature perturbation by taking the square root of the subgrid temperature variance (a prognostic variable in CLUBB) and passed that information to ZM's parcel buoyancy calculation to account for the variability of the subgrid temperature perturbation. Based on sensitivity tests, a scaling factor of 2.0 was introduced to enhance the effect.

Accounting for the gustiness effects and the variability of subgrid temperature variance was designed for EAMv1 running at ~1-degree horizontal grid spacing. Increasing model spatial resolution might reduce the impacts of these subgrid effects. A retuning of the subgrid effects might be needed when the model is run at a different horizontal resolution. The model configuration with only the gustiness effects and the subgrid temperature variance added to EAMv1 is labeled as EAMv1_SGV.

While EAMv1_SGV improves the spatial distribution of tropical clouds and precipitation (discussed in Section 3), tropical CREs and precipitation become overly strong after these changes, indicating a need for retuning. Among all the tunable parameters, we targeted the ones that were heavily tuned in EAMv1 and adjust their values to be closer to their theoretical or nominal values. In EAMv1, the coefficients controlling the autoconversion rate in convective clouds c0_lnd and c0_ocn (which are inversely proportional to the timescale that condensate is converted to precipitation) are set to 0.007, more than 3 times larger than the nominal rate used in Lord et al. (1982). The consequence is that less condensate is detrained from convective updrafts, producing cirrus clouds with very low water content in the upper troposphere. To compensate for the weak source of ice water, EAMv1 assumes more Aitken mode sulfate aerosols are efficient homogeneous ice nuclei. As a result, EAMv1 produces relatively high cloud ice number (Ni) with small ice water content and weak sedimentation rates, making the cirrus clouds more persistent and highly reflective. In this recalibration, we chose to 1) increase the supply of condensed water to cirrus clouds by reducing c0_lnd and c0_ocn to their nominal value 0.002; 2) reduce the deep convective cloud fraction parameter dp1; 3) increase the downdraft mass fraction parameter alfa; 4) reduce the assumed ice crystal radius detrained from deep convection (ice_deep); 5) increase the sensitivity of deep convection to surface temperature changes by reducing the number of lowest layers skipped for computing maximum moist static energy (mx_bot_lyr_adj) while maintaining numerical stability; and 6) enhance the lateral entrainment of deep convection by increasing the magnitude of dmpdz. It is worth noting that changing dmpdz has different effects on CREs in different parts of the tropics and a significant impact on the subtropical CREs, but the exact mechanism is unclear and requires further investigation. We took an iterative approach to retune the model, adjusting one parameter at a time and assess its impacts after each simulation. The model configuration with only ZM parameter changes added to EAMv1 is labeled as EAMv1_ZM (Table 1).

In addition to changes made in EAMv1_ZM, we also introduced two changes in MG2 in order to refine the tropical CRE (Table 3): 1) we increased the size threshold for sulfate aerosols to act as homogeneous ice nuclei (so4_sz_thresh_icenuc) to reduce ice number concentration and increase ice crystal size and thus the sedimentation rate; and 2) we increased ice_sed_ai to further increase the ice sedimentation rate. Combining the two MG2 changes with EAMv1_SGV and EAMv1_ZM, these adjustments increase cloudiness in the western and eastern Pacific and decrease cloudiness in the central Pacific as well as weaker subsidence in the subtropics.

**Table 1.** Description of tunable parameters and their values in EAMv1 and EAMv1_ZM.

| Parameter | Description | EAMv1 | EAMv1_ZM |
|---|---|---|---|
| alfa | Downdraft mass flux fraction adjustment | 0.1 | 0.14 |
| c0_lnd | Coefficient for converting convective cloud water to rain over land | 0.007 | 0.002 |
| c0_ocn | Coefficient for converting convective cloud water to rain over ocean | 0.007 | 0.002 |
| dmpdz | Parcel fractional mass entrainment rate ($m^{-1}$) | -0.7e-3 | -1.2e-3 |
| dp1 | Deep convective cloud fraction parameter | 0.045 | 0.018 |
| ice_deep | Ice particle radius detrained from deep convection ($10^{-6}$ m) | 16.e-6 | 14.e-6 |

| | | | |
|---|---|---|---|
| mx_bot_lyr_adj | Number of lowest layers skipped for computing maximum moist static energy | 2 | 1 |

**2.2 Subtropical low clouds**

Realistic simulation of low clouds across various cloud regimes requires not only a realistic simulation of the large-scale meteorological conditions, but also a versatile parameterization that is able to describe different subgrid characteristics of clouds and atmospheric thermodynamic conditions in different cloud regimes. The CLUBB parameterization employed in EAMv1 uses a multi-variate probability density function (PDF) to describe the subgrid variability of cloud, thermodynamic, and dynamic variables, that are closely connected to changes of the subgrid vertical velocity $w'$. The second and third moments of $w'$, $\overline{w'^2}$ and $\overline{w'^3}$, are prognostic variables in CLUBB, meaning that the skewness of the $w'$ PDF, $Sk_w \equiv (\overline{w'^3})/(\overline{w'^2}^{3/2})$, is predicted according to the governing equations. This is a critical treatment because it allows CLUBB to produce different subgrid characteristics in different regimes. As illustrated in Golaz et al. (2002), a low skewness corresponds to a rather symmetric PDF of $w'$ characteristic of the stratus and stratocumulus regimes, whereas a high skewness is more characteristic of a trade cumulus regime in which stronger and isolated updrafts embedded in subsidence occur more frequently. In principle, CLUBB can be used to represent the deep convection regime as well (Thayer-Calder et al., 2015;Guo et al., 2015), but it requires significant amount of effort to enable that feature so that EAMv1 still uses ZM for deep convection. The limit of $Sk_w \leq 4.5$ is imposed in EAMv1 in order to prevent numerical instability in CLUBB's equations. To simulate different subgrid variabilities in different regimes, CLUBB uses different damping coefficients and different widths of the $w'$ PDF as a function of $Sk_w$: For $X^*$ set to the diffusivity or variance of a CLUBB's prognostic variable (e.g., vertical velocity variance, total water variance, etc.), $X^* = Xb + (X - Xb) \cdot e^{-0.5 \cdot (\frac{Sk_w}{Xc})^2}$, where $X^*$ is a linear combination of low skewness values $X$ (C1, C11, and gamma_coef in Table 2) and high skewness values $Xb$ (C1b, C6rtb, C6rthlb, C11b, and gamma_coefb in Table 2) with a weighting factor $e^{-0.5 \cdot (\frac{Sk_w}{Xc})^2}$ where $Xc$ is a transition factor (C1c, C6rtc, C6rthlc, C11c, gamma_coefc in Table 2). For instance, the damping coefficient for $\overline{w'^2}$, C1*, is expressed as a function of skewness, C1, C1b, and C1c: $C1^* = C1b + (C1 - C1b) \cdot e^{-0.5 \cdot \left(\frac{Sk_w}{C1c}\right)^2}$.

Although this variable skewness treatment provides a way to simulate different subgrid characteristics in different regimes, it is poorly constrained—the equation describing $X^*$ and the chosen values of parameters $X$, $Xb$, and $Xc$ are somewhat ad hoc. In EAMv1, we set C1b and gamma_coefb to be the same as C1 and gamma_coef, respectively, to reduce unconstrained assumptions. This is a simple choice that reduces the number of free parameters in CLUBB but it also limits the flexibility of the CLUBB parameterization with implications for the model fidelity. As shown in (Brunke et al., 2019), EAMv1 produces overly bright shallow Cu and a significant bias in near-coast Sc. Therefore, we explored a different pathway in this study by setting C1 and C1b, and gamma_coef and gamma_coefb to different values and used the simulated low cloud CREs as the

Open Access EGU

tuning target to determine the parameter values. Improvements in the simulated clouds are significant, as shown in Section 3. However, it is worth noting that these improvements do not suggest that this treatment or the parameter settings are the correct representation of the physical processes in the real world. Rather, our study should be viewed as a demonstration that it is useful to enable the variable skewness treatment to facilitate the production of different subgrid characteristics in different cloud regimes. Reducing the level of complexity of the physics may sometimes compromise the model fidelity and can lead to further uncertainties in climate projections. As we further show in Section 3, these changes also affect aerosol-cloud interactions, cloud feedbacks, and, ultimately, climate sensitivity. Future studies that employ sufficient observations (from Doppler lidar, for example) or large eddy simulations (LES) to either constrain the parameter values in the current parameterization or develop a new parameterization to mimic the real-world subgrid characteristics in different regimes would be highly valuable.

To recalibrate CLUBB, we first increased the overall cloudiness by 1) weakening turbulent mixing in the planetary boundary layer (PBL), which reduces PBL decoupling and mixing between the PBL and the free troposphere. This was achieved by increasing C1, C1b, C6rtb, C6rthlb, and C14, increasing C_k10, and increasing the eddy length scale threshold (Figure 2a,b); 2) facilitating cloud formation by reducing the width of the $w'$ PDF via reducing gamma_coef and gamma_coefb; and 3) promoting Sc-like symmetric mixing rather than shallow Cu-like asymmetric mixing by reducing $Sk_w$ via increasing C8. Next, we allowed larger horizontal variation in subgrid characteristics by enlarging the difference in parameter values between high- and low- skewness regimes (i.e., *X's* and *Xb's*), as determined from satellite observations (Zhang et al., 2019b), and modified the *Xc* values to refine the transition between low- and high-skewness regime. The change in the width of the $w'$ PDF also affects the in-cloud cloud liquid water mixing ratio (Qc) variance, resulting in variable enhancement factors for warm rain processes in cloud microphysics. We also reduced the cloudiness in the shallow Cu regime by decreasing the lateral entrainment (i.e., reducing mu). These changes increase the skewness in the shallow Cu regime (Figure 2c,d), and produce realistic Sc-to-Cu transition (as discussed in Section 3). The model configuration with only CLUBB parameter changes added to EAMv1 is labeled as EAMv1_CLUBB (Table 2).

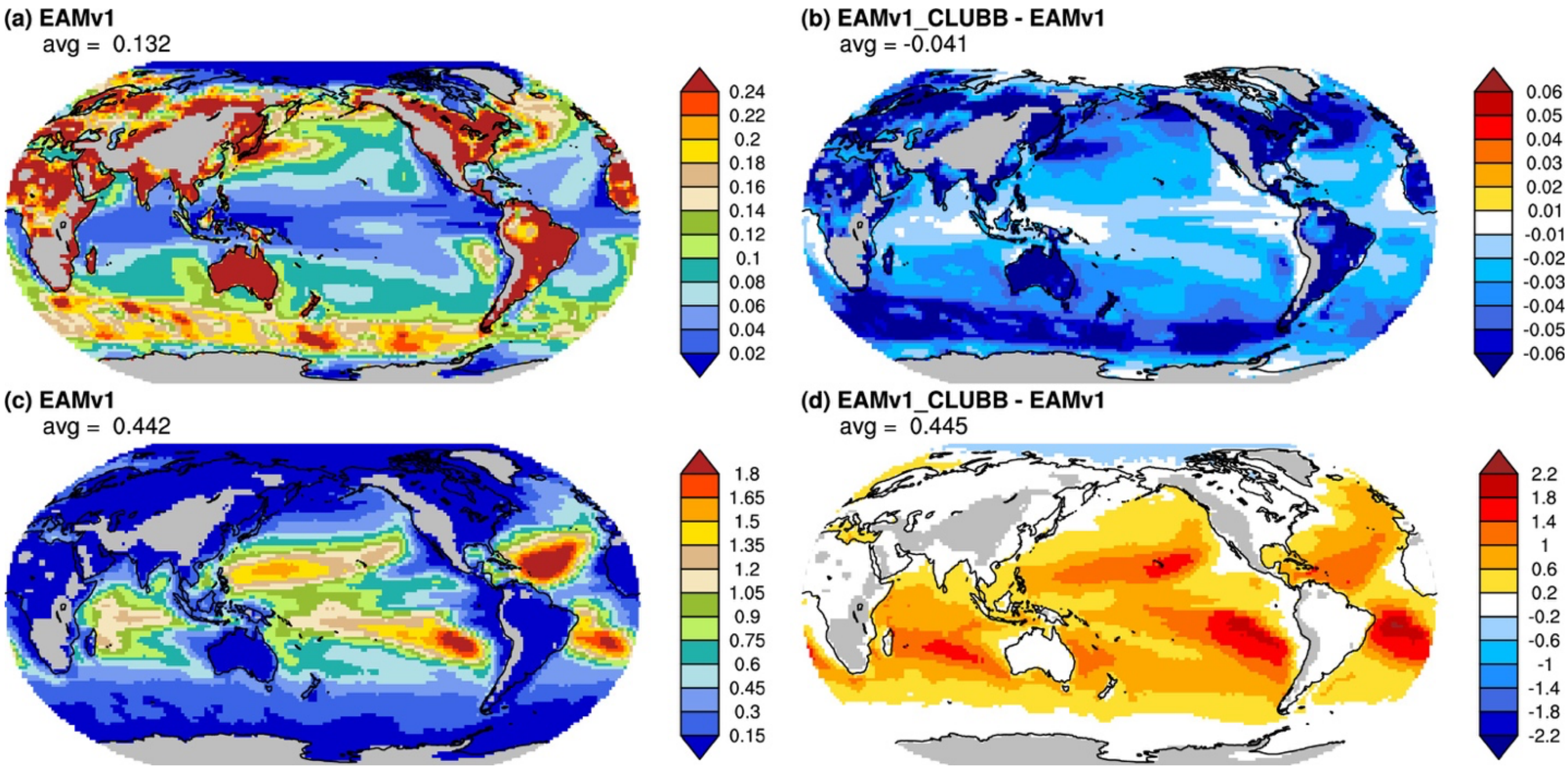

**Figure 2.** Present-day climatology of (a) mean subgrid vertical velocity variance ($\overline{w'^2}$; unit = $m^2s^{-2}$) at 925 hPa in EAMv1; (b) the $\overline{w'^2}$ difference between EAMv1_CLUBB and EAMv1; (c) the skewness of subgrid vertical velocity ($Sk_w \equiv (\overline{w'^3})/(\overline{w'^2}^{3/2})$) in EAMv1; and (d) the $Sk_w$ difference between EAMv1_CLUBB and EAMv1 at 925 hPa.

**Table 2.** Description of tunable parameters and their values in EAMv1 and EAMv1_CLUBB.

| Parameter | Description | EAMv1 | EAMv1_CLUBB |
|---|---|---|---|
| C1 | Coefficient for $\overline{w'^2}$ damping at low $Sk_w$ | 1.335 | 2.4 |
| C1b | Coefficient for $\overline{w'^2}$ damping at high $Sk_w$ | 1.335 | 2.8 |
| C1c | Coefficient for $Sk_w$ dependency of C1 | 1.0 | 0.75 |
| C6rtb | Coefficient for $\overline{w'q_t'}$ damping at high $Sk_w$ | 6.0 | 7.5 |
| C6rtc | Coefficient for $Sk_w$ dependency of C6rt | 1.0 | 0.5 |
| C6thlb | Coefficient for $\overline{w'\theta_l'}$ damping at high $Sk_w$ | 6.0 | 7.5 |
| C6thlc | Coefficient for $Sk_w$ dependency of C6rthl | 1.0 | 0.5 |
| C8 | Coefficient for $\overline{w'^3}$ damping | 4.3 | 5.2 |
| C11 | Coefficient for $\overline{w'^3}$ damping at low $Sk_w$ | 0.80 | 0.7 |
| C11b | Coefficient for $\overline{w'^3}$ damping at high $Sk_w$ | 0.35 | 0.2 |
| C11c | Coefficient for $Sk_w$ dependency of C11 | 0.5 | 0.85 |
| C14 | Coefficient for $\overline{u'^2}$ and $\overline{v'^2}$ damping | 1.06 | 2.0 |
| c_k10 | Ratio of eddy diffusivity of momentum to heat | 0.30 | 0.35 |
| gamma_coef | The width of the Gaussian distribution at low $Sk_w$ | 0.32 | 0.12 |
| gamma_coefb | The width of the Gaussian distribution at high $Sk_w$ | 0.32 | 0.28 |
| gamma_coefc | Coefficient for $Sk_w$ dependency of the Gaussian distribution width | 5.0 | 1.2 |
| mu | Fractional entrainment rate ($m^{-1}$) | 1.e-3 | 5.e-4 |
| wpxp_L_thresh | Eddy length scale threshold for Newtonian and buoyancy damping of $\overline{w'q_t'}$ and $\overline{w'\theta_l'}$ (m) | 60 | 100 |

Uncertainties in cloud microphysical processes affect all non-deep-convective clouds, including subtropical clouds. The tuning of the microphysical processes is justified by fundamental process-level uncertainties as well as simplifying assumptions made in bulk microphysics schemes (including the MG2 scheme used in EAMv1) regarding particle size distributions and the subgrid scale distribution of cloud properties. To increase cloudiness in the Sc regime, we weakened cloud-top entrainment by enhancing droplet sedimentation (Bretherton et al., 2007). Next, we essentially removed the unphysical lower bound of subgrid vertical velocity used for cloud droplet nucleation (wsubmin) by setting the parameter to a very low value that was almost never reached. This improves the coupling between the simulated subgrid updraft velocity and the cloud microphysical properties such as droplet number, size, and condensate amount. We also adjusted the warm rain processes by restoring the heavily tuned prc_exp1, the exponent of droplet number (Nc) in the autoconversion parameterization in EAMv1 (Rasch et al., 2019), to the nominal value based on observations (Wood, 2005). This increases cloudiness in areas where more aerosols are present. The accretion process is also enhanced to compensate for the reduction of precipitation from the change in autoconversion.

It is worth noting that the autoconversion parameterizations in EAMv1 is based on Khairoutdinov and Kogan (2000), which is a function of Qc and Nc. However, the parameter values (i.e., the scale factor and exponents of Qc and Nc) for different cloud regimes are very different (Kogan, 2013), indicating that the autoconversion process is governed by more factors than those considered in the current parameterization. Therefore, there is no one set of parameter values that can optimally represent the autoconversion process for all cloud regimes. Adjusting these parameters to achieve reasonably good representation of cloud and precipitation simulations is possible, but one should use caution when interpreting the results and acknowledge the fundamental deficiency of the underlying process representations in the model. Given the importance of warm rain processes (autoconversion and accretion) in simulating clouds and precipitation and their responses to forcings, developing new parameterizations that can flexibly represent these processes over a broad range of cloud types to address this model deficiency should be included in the roadmap toward next generation ESMs.

### 2.3 Mid- and high-latitude clouds

Another significant cloud bias present in mid- and high latitudes in EAMv1 can be attributed to excessive supercooled liquid clouds due to a suppressed Wegener–Bergeron–Findeisen (WBF) process (Rasch et al., 2019). This reduces conversion from liquid to ice is a consequence of an inherited value of a scaling factor of 0.1 that tuned down the WBF process rate significantly. The WBF rate was tuned down in order to address an underestimate supercooled liquid clouds in CAM5 (Tan et al., 2016;DeMott et al., 2010;Liu et al., 2011). However, EAMv1 eliminated one of the sources of this bias by replacing the Meyers et al. (1992) ice nucleation (IN) scheme from CAM5 with a classical-nucleation-theory (CNT)-based scheme (Hoose et al., 2010;Wang et al., 2014). The CNT scheme addresses the overproduction of ice crystals by Meyers et al. (1992), which scavenges liquid water rapidly. Replacing the Meyers et al. (1992) scheme but maintaining the slow WBF conversion from

liquid to ice produced unrealistically high liquid water path (LWP) in mid- and high latitudes: The LWP poleward of 60N and over Southern Ocean is 15-30% higher than the LWP in the tropics (see discussion in Section 3.1; Figure 3). An unrealistic meridional distribution of LWP can cause significant biases in the radiative energy distribution, atmospheric circulation, and water cycle. The excessive cloud liquid water in mid- and high latitudes can also lead to strong aerosol-cloud interactions and biases in long range transport of aerosols due to strong wet scavenging (Wang et al., 2013). The high-resolution configuration of E3SMv1 reverted the IN scheme to Meyers et al. (1992) to address this bias (Caldwell et al., 2019), but the error compensation from two incorrect cloud processes can potentially produce biases in cloud microphysical properties, adversely impacting the credibility of climate projections.

In this study, we adopted an alternative approach to address this bias. Zhang et al. (2019a) shows that improvements can be made by increasing the WBF process rate. Therefore, we retained the new CNT-based IN scheme that had been shown to perform better than the Meyers et al. (1992) scheme, and significantly increased the scale factor for the WBF process to increase the conversion from liquid to ice. Furthermore, the parameter adjustments in the ZM scheme that improve the upper tropospheric ice clouds in the tropics also increase ice clouds in the mid-latitudes. The model configuration with only MG2 parameter changes added to EAMv1 is labeled as EAMv1_MP (Table 3). The combination of EAMv1_MP and EAMv1_ZM lead to lower LWP and higher ice water path (IWP) in the mid- and high latitudes (see discussion in Section 3.1; Figure 3).

**Table 3.** Description of tunable parameters and their values in EAMv1 and EAMv1_MP.

| Parameter | Description | EAMv1 | EAMv1_MP |
|---|---|---|---|
| cld_sed | Liquid droplet sedimentation adjustment | 1.0 | 1.8 |
| ice_sed_ai | Ice particle fall speed parameter | 500 | 1200 |
| micro_mg_accre_enhan_fac | Liquid cloud accretion adjustment | 1.5 | 1.75 |
| micro_mg_berg_eff_factor | WBF process adjustment | 0.1 | 0.7 |
| prc_exp1 | Exponent of liquid droplet number concentration in autoconversion | -1.2 | -1.4 |
| so4_sz_thresh_icenuc | Aitken model sulfate aerosol size threshold for homogeneous ice nucleation (m) | 0.05e-6 | 0.08e-6 |
| wsubmin | Minimum subgrid vertical velocity used for liquid droplet nucleation (m $s^{-1}$) | 0.2 | 0.001 |

## 2.4 Model simulations

The final revised model (labeled as EAMv1P) includes all changes discussed above and two additional changes to the scale factors for emissions of sea spray and dust aerosols (Table 4) so that the global mean aerosol optical depth ($\tau_{aer}$) is similar between EAMv1 and the recalibrated model.

**Table 4.** Description of emission scale factors and their values in EAMv1 and EAMv1P.

| Parameter | Description | EAMv1 | EAMv1P |
|---|---|---|---|

| | | | |
|---|---|---|---|
| seasalt_emis_scale | Adjustment for sea spray aerosol mobilization | 0.85 | 0.60 |
| dust_emis_fact | Adjustment for dust mobilization | 2.05 | 2.8 |

In this paper, we show model results from grouped parameter adjustments instead of individual parameter changes. Model configurations are listed in Table 5. The effects and the mechanisms of each individual parameter adjustment require further investigation and will be documented in separate manuscripts.

**Table 5.** List of model configurations.

| Configuration | Description |
|---|---|
| EAMv1 | Default EAMv1 configuration |
| EAMv1_CLUBB | EAMv1 with only the CLUBB changes |
| EAMv1_MP | EAMv1 with only the MG2 changes |
| EAMv1_SGV | EAMv1 with only the inclusion of subgrid effects |
| EAMv1_ZM | EAMv1 with only the ZM changes |
| EAMv1P | EAMv1 with all the changes |

Each model configuration was used for global atmospheric simulations where the atmosphere model was coupled with an interactive land model but sea surface temperature (SST) and sea ice cover were prescribed. Emissions of aerosols and their precursors were obtained from CMIP phase 6 (CMIP6) emission datasets (Hoesly et al., 2018;van Marle et al., 2017). We ran the coarse resolution EAM configuration (i.e., ne30np4, which corresponds to approximately 1-degree horizontal grid spacing) with

1) Present-day (Year 2000 here) forcing;
2) Pre-industrial (Year 1850 here) forcing;
3) Present-day forcing, except for pre-industrial aerosol emissions;
4) Pre-industrial forcing with SST elevated by 4 K uniformly;
5) Present-day forcing with SST, sea ice, and solar constant set to pre-industrial conditions.

We compute the effective radiative forcing (ERF) from these prescribed-SST-and-sea-ice experiments (Hansen et al., 2005). Forster et al. (2016) compared different methodologies for computing the ERF and recommend the prescribed-SST-and-sea-ice method. The differences between 1) and 3) provide information on the impacts of anthropogenic aerosols. Contrasting 2) and 4) provides climate feedback estimates. Total anthropogenic ERF ($ERF_{ant}$), also termed total adjusted forcing, is derived by comparing 5) and 2) (Forster et al., 2013). $ERF_{ant}$ includes anthropogenic forcing (greenhouse gas concentrations, aerosols, and land use land cover change) and rapid adjustments in water vapor, clouds, and temperature. All simulations were run for 11 years but the first year was considered as model spin-up and was excluded from the climatology and subsequent analysis.

## 3 Results

### 3.1 Clouds

Table 6 summarizes the global mean present-day climatology of cloud properties using the various model configurations listed in Table 5. The CREs are computed by double radiation calls in the model. Shortwave and longwave CREs contributed from liquid clouds, ice clouds, convective clouds, and snow are independently computed. Rain droplets are not radiatively active in EAMv1. Because radiative transfer is nonlinear, the sum of the CREs from clouds and snow are not equal to the total CRE.

Compared with EAMv1, EAMv1_CLUBB shows lower-magnitude top-of-atmosphere (TOA) net CREs due primarily to a reduction of liquid clouds in the shallow Cu regime. EAMv1_MP also produces lower-magnitude total shortwave and longwave CREs, but it is attributable to the reduction of CREs from both liquid and ice clouds from increasing the WBF process. EAMv1_SGV only marginally increases CREs but EAMv1_ZM significantly enhance the CREs from liquid and ice clouds, though the convective CREs are significantly reduced in EAMv1_ZM because the convective cloud fraction is much lower as a result of reducing dp1. The CRE differences are consistent with the differences in cloud optical depth ($\tau_{cld}$). In contrast, cloud fractions and cloud heights are relatively invariant between different configurations. EAMv1_MP reduces LWP, IWP, Nc, and Ni mostly in mid- and high latitudes, and EAMv1_ZM increases them mostly in the tropics. The EAMv1P configuration combines all of the changes and produces global mean net CRE (-24.28 Wm$^{-2}$) not very different from that in EAMv1 (-24.7 Wm$^{-2}$), but we emphasize that the spatial distribution of clouds is as important as global mean values because different cloud regimes may respond to perturbations differently.

**Table 6.** Global mean 10-year averaged cloud properties of EAMv1, EAMv1_CLUBB, EAMv1_MP, EAMv1_SGV, EAMv1_ZM, and EAMv1P. Relevant cloud properties listed here are TOA shortwave cloud radiative effects (SWCRE; unit = Wm$^{-2}$) and that of liquid clouds (SWCRE$_{liq}$), ice clouds (SWCRE$_{ice}$), snow (SWCRE$_{snow}$), and convective clouds (SWCRE$_{conv}$); TOA longwave cloud radiative effects (LWCRE; unit = Wm$^{-2}$) and that of liquid clouds (LWCRE$_{liq}$), ice clouds (LWCRE$_{ice}$), snow (LWCRE$_{snow}$), and convective clouds (LWCRE$_{conv}$); cloud fraction (unit = %) of the total column (F$_{cld,tot}$), below 700hPa (F$_{cld,low}$), between 400 and 700 hPa (F$_{cld,med}$) and above 400 hPa (F$_{cld,hgh}$); optical depth of all clouds ($\tau_{cld}$) and that of liquid clouds ($\tau_{liq}$), ice clouds ($\tau_{ice}$), snow ($\tau_{snow}$), convective clouds ($\tau_{conv}$), and all clouds below 700 hPa ($\tau_{low}$) and above 400 hPa ($\tau_{hgh}$); column-integrated total LWP (unit = g m$^{-2}$) and IWP (unit = g m$^{-2}$), N$_c$ (unit = 10$^9$ m$^{-2}$) and N$_i$ (unit = 10$^9$ m$^{-2}$); altitude of the top (Z$_{hgh,top}$; unit = km) and base (Z$_{hgh,top}$; unit = km) of clouds above 400 hPa and altitude of the top (Z$_{low,top}$; unit = km) and base (Z$_{low,top}$; unit = km) of clouds below 700 hPa.

| Variable | EAMv1 | EAMv1_CLUBB | EAMv1_MP | EAMv1_SGV | EAMv1_ZM | EAMv1P |
|---|---|---|---|---|---|---|
| SWCRE | -49.31 | -45.37 | -45.11 | -50.30 | -54.15 | -47.27 |
| SWCRE$_{liq}$ | -34.87 | -31.36 | -30.52 | -35.33 | -37.22 | -30.23 |
| SWCRE$_{ice}$ | -10.73 | -10.01 | -8.22 | -10.85 | -16.62 | -13.96 |
| SWCRE$_{snow}$ | -6.22 | -6.03 | -4.72 | -6.22 | -6.83 | -5.32 |
| SWCRE$_{conv}$ | -5.73 | -5.47 | -5.78 | -6.27 | -2.59 | -2.71 |
| LWCRE | 24.61 | 23.55 | 20.98 | 24.66 | 27.11 | 22.99 |
| LWCRE$_{liq}$ | 10.95 | 10.06 | 7.18 | 10.85 | 10.89 | 6.59 |

| | | | | | | |
|---|---|---|---|---|---|---|
| $LWCRE_{ice}$ | 14.19 | 13.56 | 11.03 | 14.27 | 17.49 | 14.47 |
| $LWCRE_{snow}$ | 6.79 | 6.69 | 4.66 | 6.77 | 7.16 | 5.03 |
| $LWCRE_{conv}$ | 1.23 | 1.22 | 1.27 | 1.27 | 0.51 | 0.54 |
| $F_{cld,tot}$ | 67.95 | 65.58 | 65.98 | 69.21 | 69.50 | 66.22 |
| $F_{cld,low}$ | 42.73 | 39.73 | 43.04 | 44.58 | 42.80 | 41.10 |
| $F_{cld,med}$ | 27.18 | 26.92 | 25.99 | 26.75 | 28.99 | 27.29 |
| $F_{cld,hgh}$ | 38.88 | 38.28 | 35.72 | 39.31 | 40.44 | 37.52 |
| $\tau_{cld}$ | 8.25 | 7.65 | 7.27 | 8.19 | 9.61 | 8.00 |
| $\tau_{liq}$ | 5.37 | 4.92 | 4.58 | 5.25 | 4.24 | 4.36 |
| $\tau_{ice}$ | 0.48 | 0.43 | 0.34 | 0.48 | 1.02 | 0.82 |
| $\tau_{snow}$ | 0.50 | 0.49 | 0.47 | 0.50 | 0.58 | 0.52 |
| $\tau_{conv}$ | 1.90 | 1.81 | 1.89 | 1.97 | 2.37 | 2.30 |
| $\tau_{low}$ | 5.61 | 5.09 | 5.38 | 5.59 | 6.10 | 5.26 |
| $\tau_{hgh}$ | 0.62 | 0.60 | 0.53 | 0.63 | 0.87 | 0.69 |
| LWP | 53.71 | 51.11 | 47.02 | 52.98 | 58.79 | 49.77 |
| IWP | 11.07 | 10.49 | 9.72 | 11.11 | 20.35 | 17.98 |
| $N_c$ | 14.35 | 13.22 | 12.83 | 14.16 | 15.53 | 11.91 |
| $N_i$ | 0.29 | 0.25 | 0.17 | 0.29 | 0.57 | 0.43 |
| $Z_{HCT}$ | 11.90 | 11.83 | 11.70 | 11.94 | 11.92 | 11.74 |
| $Z_{HCB}$ | 8.87 | 8.82 | 8.75 | 8.89 | 8.84 | 8.72 |
| $Z_{LCT}$ | 2.04 | 2.01 | 2.03 | 1.98 | 2.11 | 2.03 |
| $Z_{LCB}$ | 0.61 | 0.57 | 0.60 | 0.59 | 0.60 | 0.55 |

Figure 3 shows that the changes made in EAMv1_ZM increase the IWP significantly at most latitudes except the polar regions. This is likely due to the combination of reducing the convective autoconversion efficiency (by reducing c0_lnd and c0_ocn) and decreasing the ice particle size detrained from deep convection (by reducing ice_deep), which increases the ice crystal number and prolongs the lifetime of ice clouds. EAMv1_MP shows slight reduction of IWP in the Southern Ocean, while significantly reducing LWP in mid- and high latitudes. This remedies the unrealistically high LWP in those regions in EAMv1 due to its weak WBF process.

Because of the general IWP increase in EAMv1_ZM, the meridional distribution of liquid condensate fraction (LCF) is reduced as a result of changes made in ZM (Figure 3c). Interestingly, the global mean atmospheric temperature where ice and liquid each contribute to 50% of total condensate, T5050 (McCoy et al., 2015;McCoy et al., 2016), in EAMv1 is about 240 K, which is significantly lower than observational estimates of 254-258 K (McCoy et al., 2016). While the CMIP5 models tend to freeze liquid condensates at higher temperatures (Cesana et al., 2015;Tan et al., 2016;McCoy et al., 2016), EAMv1 appears to have overcorrected this bias and produced excessive supercooled liquid at low temperatures. Consistent with Zhang et al. (2019a), EAMv1_MP increases the T5050. Combining with changes introduced in EAMv1_ZM, EAMv1P produces a much more reasonable T5050 of 254K, which is at the lower bound of the observational estimates. We note that even though Hu et al. (2010) provided an observationally derived LCF-T relationship based on the Cloud-Aerosol Lidar and Infrared Pathfinder Satellite Observation (CALIPSO) measurements (Winker et al., 2007), EAMv1 does not have the CALIPSO cloud phase simulator (Cesana and Chepfer, 2013) so that a fair comparison is not possible. Evaluating the model LCF-T relationship against satellite observations in a consistent way will be very useful and requires further investigation.

Differences in the simulated cloud phase can have important implications for aerosol-cloud interactions (ACI) because the physical processes regulating the interactions between aerosols and warm cloud and between aerosols and cold clouds are very different. The simulated cloud phase can also affect cloud feedbacks to warming (Tan et al., 2016). The ACI and cloud feedbacks will be discussed in Section 3.4 and 3.5.

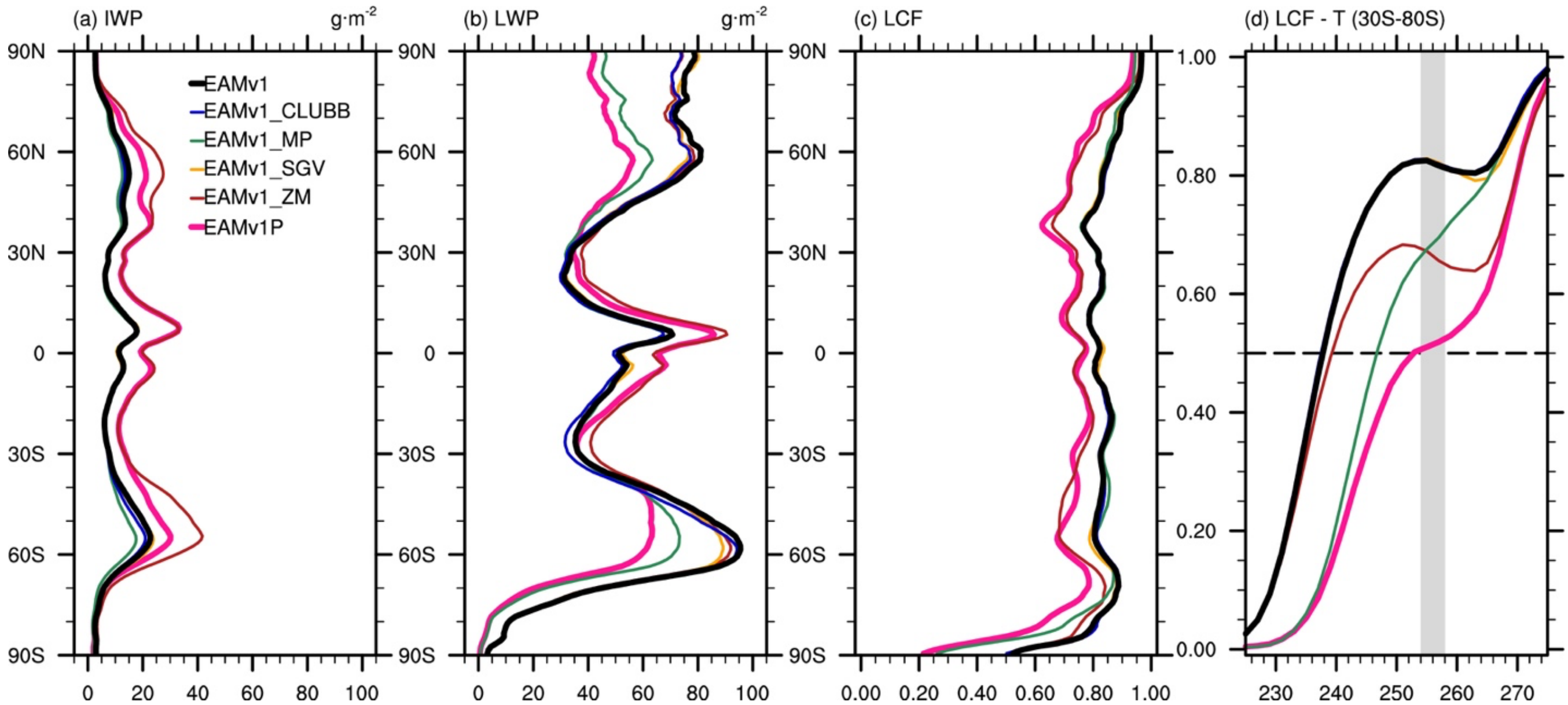

**Figure 3.** Zonal mean of (a) ice water path (IWP); (b) liquid water path (LWP); and (c) liquid condensate fraction (LCF), defined as the ratio of liquid to total cloud condensate amount; and (d) LCF as a function of temperature (unit = K) between 30°S and 80°S. The horizontal dashed line in (d) denotes T5050 (McCoy et al., 2016;McCoy et al., 2015) where ice and liquid each contributes to 50% of the total condensate. The observational estimate of T5050 range (McCoy et al., 2016) are shown in the gray shaded area.

Figure 4 shows the impacts of the grouped parameter changes on TOA SWCRE, illustrating how EAMv1 regional cloud biases are addressed by the various model configurations. By enabling the variable skewness treatment in CLUBB and the adjustments that follow, the overly bright shallow Cu and the significant lack of Sc in EAMv1 are greatly improved in EAMv1_CLUBB. SWCRE associated with coastal Sc is increased by about 10-20 Wm$^{-2}$ off the coast of California and by about 30-40 Wm$^{-2}$ off the coast of Peru and Chile, while over shallow Cu regime SWCRE is reduced by 20-30 Wm$^{-2}$. The elevated $Sk_w$ in the shallow Cu regime also reduces the cloud water removal time scale (not shown). In EAMv1_MP, tuning up the WBF process corrects the SWCRE bias at mid- and high latitudes. With increasing fraction of ice condensate, the cloud water removal timescale is reduced (not shown) because warm rain processes are less efficient in removing condensate than ice precipitation processes (Mülmenstädt et al., 2021). Adjustments to cloud droplet sedimentation and warm rain processes

make moderate improvements to Sc. Changes to ice crystal sedimentation and sulfate aerosol size result in significant reduction in tropical SWCRE, as upper tropospheric clouds respond to these adjustments the most. Furthermore, EAMv1_SGV increases clouds in areas where large-scale winds are weak and convection occurs frequently, including the TWP and Amazonia. Some effects on the eastern Pacific are also observed. EAMv1_ZM further increases cloudiness in the ITCZ, especially in the western and eastern Pacific. Cloudiness in the Southern Pacific Convergence Zone (SPCZ) is also improved. Setting c0_lnd and c0_ocn to lower values essentially slows down the convective autoconversion process, leading to longer water removal timescale in the tropics. Combining all the changes, EAMv1P shows improved cloud distribution with reduced biases in the tropics, subtropics, and mid- and high latitudes, indicating that the changes discussed in Section 2 are appropriate.

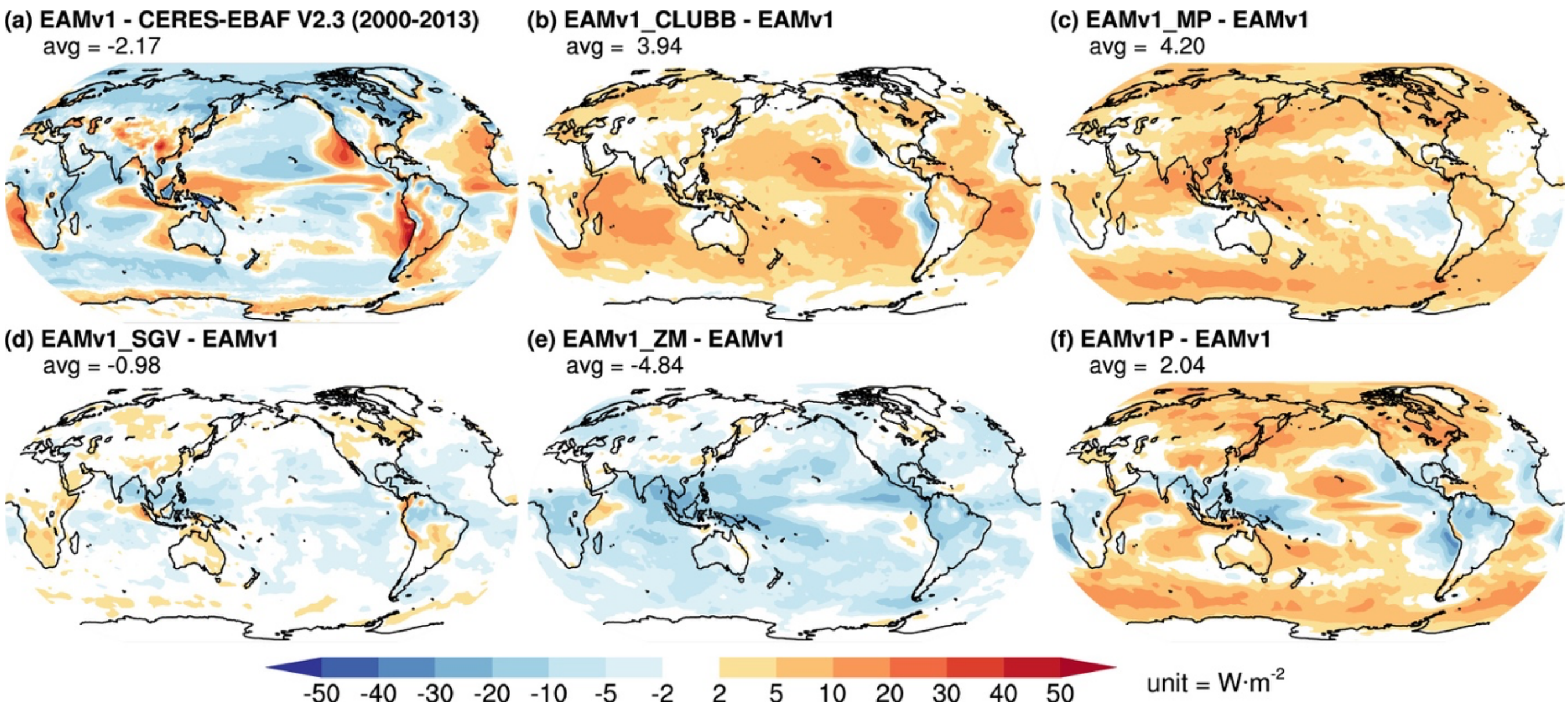

**Figure 4.** Difference of present-day TOA SWCRE (W m$^{-2}$) climatology between (a) EAMv1 and Clouds and Earth's Radiant Energy System (CERES) (Wielicki et al., 1996) Energy Balance and Filled (EBAF) (Loeb et al., 2012;Loeb et al., 2003) v2.3 averaged over 2000-2013; (b) EAMv1_CLUBB and EAMv1; (c) EAMv1_MP and EAMv1; (d) EAMv1_SGV and EAMv1; (e) EAMv1_ZM and EAMv1; and (f) EAMv1P and EAMv1. Model TOA SWCRE are 10-year averages.

Further evaluation using the Cloud-Aerosol Lidar and Infrared Pathfinder Satellite Observation (CALIPSO) cloud simulator (Chepfer et al., 2008), as part of the Cloud Feedback Model Intercomparison Project (CFMIP) Observation Simulator Package (COSP) (Bodas-Salcedo et al., 2011), that samples the model at 1:30 PM local time for comparisons of total cloud fraction with version 3.1.2 of the General Circulation Model-Oriented CALIPSO Cloud Product (GOCCP) (Chepfer et al., 2010)) shows similar results. Figure 5 shows cloud bias reductions in the Northern Hemisphere high latitudes, the stratocumulus regions, TWP, and over tropical lands. These improvements match our expectation, increasing our confidence in the model clouds. However, there are some differences between the comparisons in Figure 4 and 5. In Figure 4, EAMv1

Open Access EGU

shows overly bright clouds in trade cumulus regions and over the Southern Ocean and EAMv1P reduces these biases. Figure 5 shows that EAMv1 produces less clouds in trade cumulus regions and more clouds over then Southern Ocean than GOCCP, and EAMv1P increases the bias in trade cumulus regions and does not change the Southern Ocean bias. This could indicate that the improvements to the TOA SWCRE over these regions are achieved by compensating errors between cloud fraction and cloud optical depth. This also demonstrates the challenge of optimizing an ESM when there are only limited observables to evaluate the model.

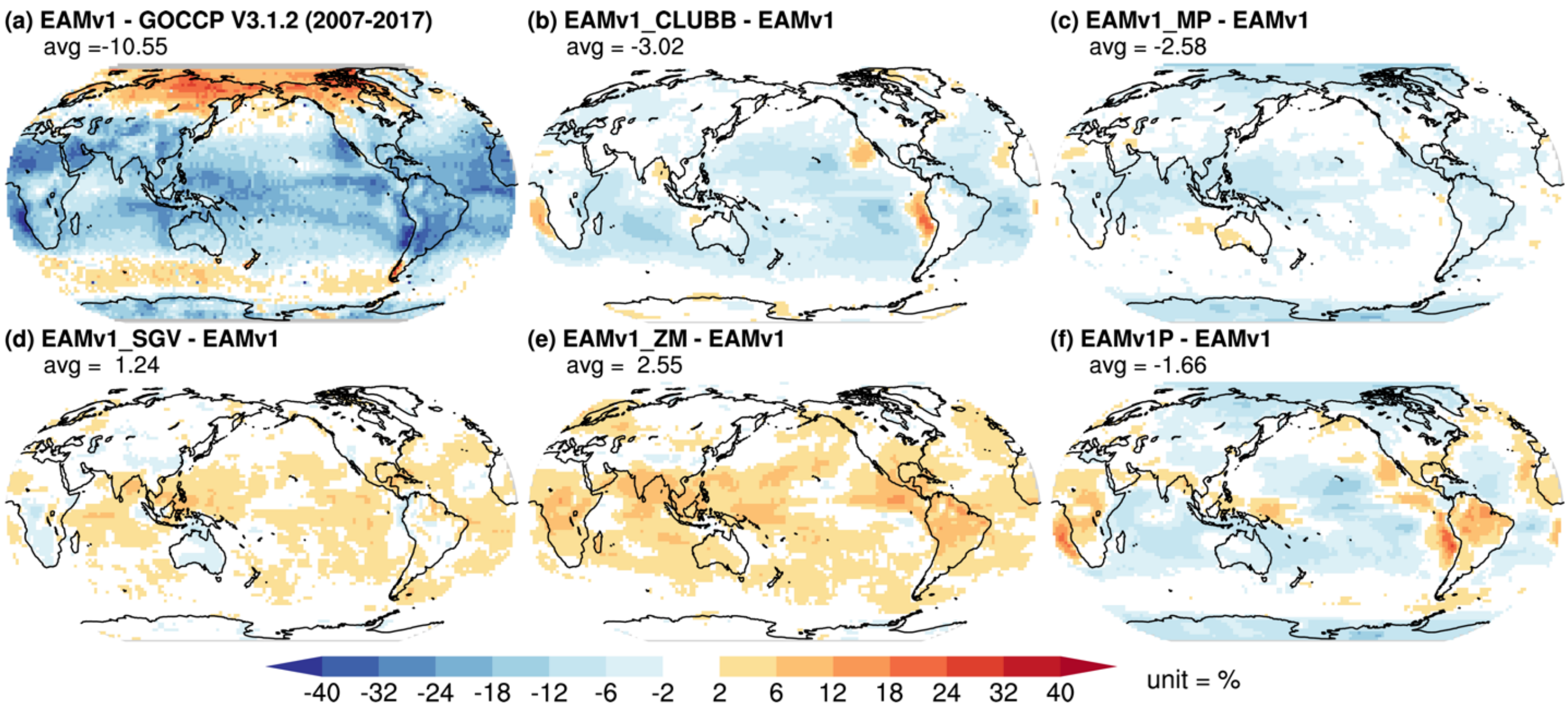

**Figure 5.** Total cloud fraction between (a) EAMv1 and the version 3.1.2 of the GOCCP total cloud fraction (Chepfer et al., 2010) averaged over 2007-2017; (b) EAMv1_CLUBB and EAMv1; (c) EAMv1_MP and EAMv1; (d) EAMv1_SGV and EAMv1; (e) EAMv1_ZM and EAMv1; and (f) EAMv1P and EAMv1. Model cloud fractions are derived from the CALIPSO cloud simulator (Chepfer et al., 2008) sampled at 1:30 PM local time.

Given the importance of low clouds in Earth's radiation budget, we investigate the planetary boundary layer (PBL) properties in different model configurations to gain insights into the physical mechanisms associated with the parameter adjustments. Table 7 shows that the adjustments to CLUBB parameters affect the simulated PBL properties significantly, as expected. The adjustments to CLUBB parameters directly reduce the $\overline{w'^2}_{925}$ and increase $Sk_{w925}$, but they also govern the turbulent mixing and cloud processes in the PBL, producing different impacts on the PBL. The weaker $\overline{w'^2}_{925}$ indicates a shallower PBL, reducing both the PBL decoupling strength ($PBL_{dcp}$, defined as the difference between cloud base height and lifting condensation level (LCL) (Jones et al., 2011) as well as the frequency of occurrence of decoupled PBL. The cloud-top entrainment rate for PBL clouds ($w_e$) is reduced as a result. Note that the changes to the ZM deep convection scheme also

reduce cloud-top entrainment likely through strengthening of the large-scale subsidence in the sub-tropics. On the other hand, higher $Sk_{w925}$ indicates that the model produces more asymmetric mixing and shallow Cu-like clouds. The increase of Cu-like clouds and decrease of Sc-like clouds can lead to weaker low cloud feedback (Cesana et al., 2019) and results will be discussed further in Section 3.5. Finally, we find that the inverse relative variance of cloud water, which affects the enhancement factors of autoconversion, accretion, and immersion freezing (Morrison and Gettelman, 2008), are not sensitive to the parameter changes. Thus, there is a limited impact on these three processes from changes in subgrid in-cloud water variance.

**Table 7.** Global mean 10-year averaged PBL properties of EAMv1, EAMv1_CLUBB, EAMv1_MP, EAMv1_SGV, EAMv1_ZM, and EAMv1P. Relevant PBL properties listed here are subgrid vertical velocity variance ($\overline{w'^2}$, unit = m$^2$ s$^{-2}$) and the subgrid vertical velocity skewness ($Sk_w$) at 925 hPa; PBL decoupling strength ($PBL_{dcp}$; unit = km); PBL decoupling frequency ($FREQ_{dcp}$; unit = %); cloud-top entrainment rate ($w_e$; unit = m day$^{-1}$); and inverse relative variance of cloud water at 925 hPa ($\nu_{925}$). Only columns with clouds in PBL are sampled.

| Variable | EAMv1 | EAMv1_CLUBB | EAMv1_MP | EAMv1_SGV | EAMv1_ZM | EAMv1P |
|---|---|---|---|---|---|---|
| $\overline{w'^2}_{925}$ | 0.13 | 0.09 | 0.13 | 0.12 | 0.13 | 0.09 |
| $Sk_{w925}$ | 0.45 | 0.90 | 0.47 | 0.41 | 0.50 | 1.02 |
| $PBL_{dcp}$ | 0.32 | 0.28 | 0.32 | 0.30 | 0.31 | 0.26 |
| $FREQ_{dcp}$ | 26.20 | 22.90 | 26.38 | 25.67 | 25.90 | 22.05 |
| $w_e$ | 171.9 | 148.8 | 171.7 | 164.8 | 143.4 | 121.5 |
| $\nu_{925}$ | 2.91 | 2.83 | 2.85 | 3.05 | 2.92 | 2.87 |

Next, we compare the estimated inversion strength (EIS) (Wood and Bretherton, 2006) between model simulations and a reanalysis dataset. EIS has traditionally been considered as an important cloud controlling factor affecting low cloud feedback (Klein et al., 2017). Figure 6 shows that EAMv1 generally underestimates EIS, except in the tropics. The revised model EAMv1P alleviates many of the biases, but some biases remain. EAMv1_CLUBB reduces the bias over land in general (except for north Africa) as well as the mid- and high-latitude ocean. EAMv1_MP shows significant difference in the polar regions, indicating that reducing supercooled liquid in the mixed phase cloud regime can change polar PBL properties. EAMv1_SGV enhances the EIS as a result of convection invigoration. Similarly, EAMv1_ZM directly reduces the bias in the tropics and produces enhanced EIS in mid- and high latitudes through large-scale circulation responses.

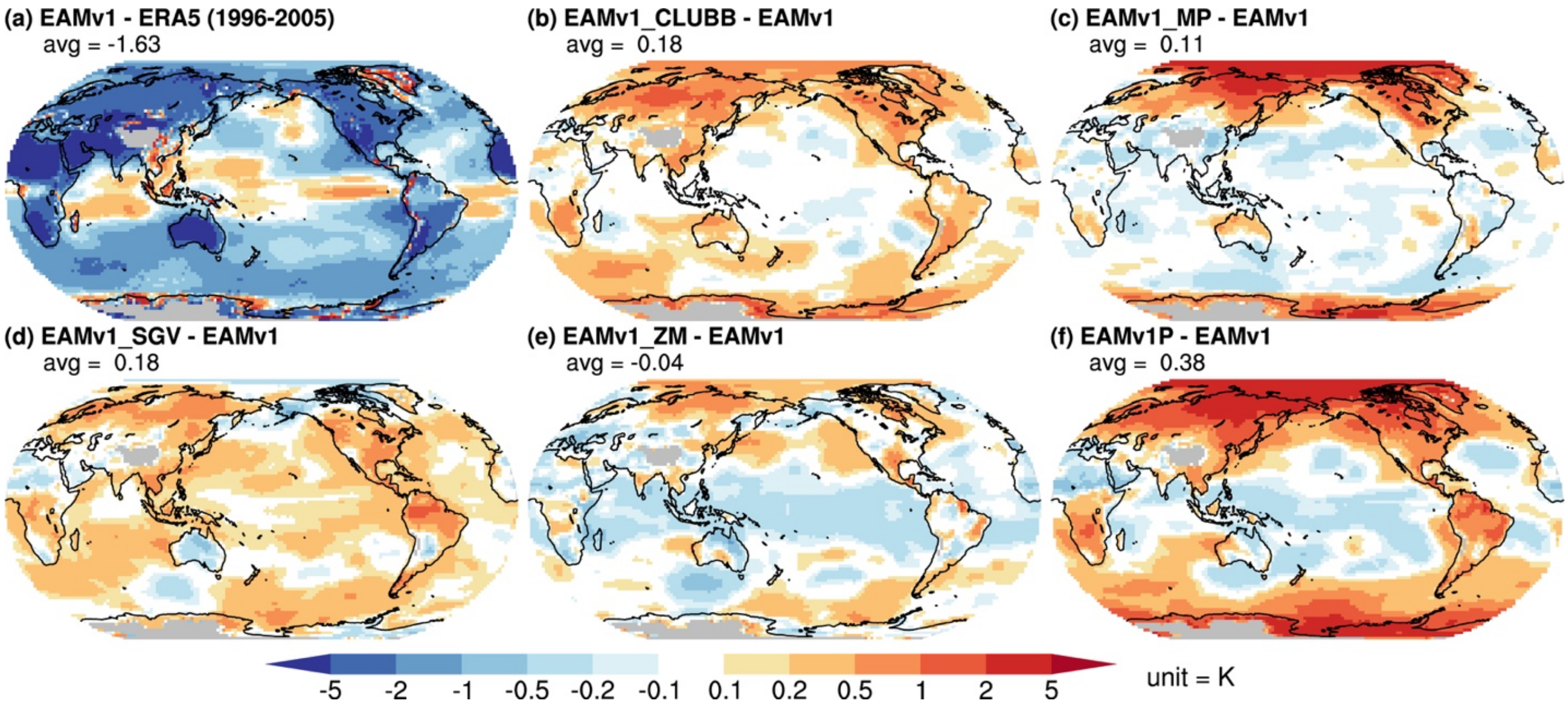

**Figure 6.** Estimated inversion strength (EIS; unit = K). The EIS computed from European Centre for Medium-Range Weather Forecasts' (ECMWF's) fifth generation global meteorological reanalysis (ERA5) (Hersbach et al., 2019) is used for the comparison with EAMv1.

Figure 7 shows that changes in EAMv1_CLUBB also significantly reduce the PBL decoupling strength (Jones et al., 2011). The decoupled PBL is often a sign that the PBL grows too deep, so that the negative buoyancy at the top of the PBL is insufficient to mix through the sub-cloud layer (Wood, 2012). These conditions favor the transition from Sc to shallow Cu (Wood, 2012;Xiao et al., 2011), reducing the overall cloudiness and contributing to the lack of Sc in EAMv1. This longstanding regional cloud bias is primarily alleviated by adjustments to CLUBB parameters, particularly the increases of C1 and C1b that reduces $\overline{w'^2}$. Furthermore, EAMv1_SGV also reduces the PBL decoupling strength over tropical land as well as subtropical and mid-latitude ocean, likely due to the enhanced surface flux which moistens the PBL. The recalibrated model EAMv1P shows the collective effect of significant reduction in decoupling strength (Figure 7) and frequency (not shown).

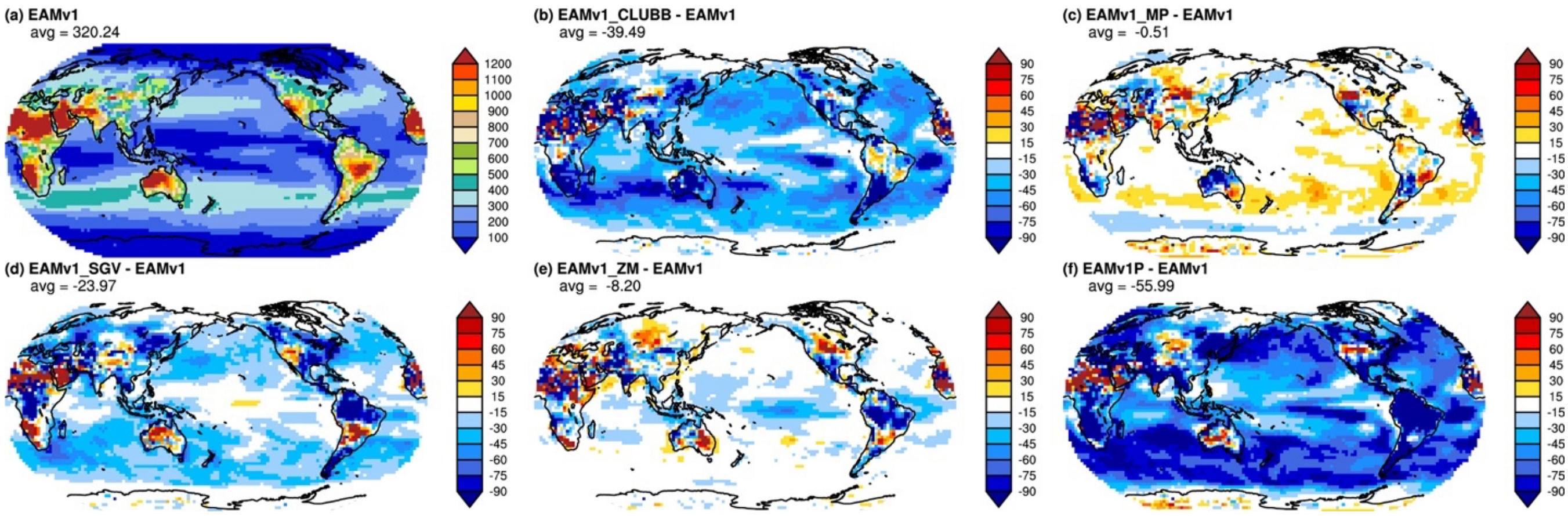

**Figure 7.** PBL decoupling strength (unit = m) defined as the difference between LCL and the altitude of cloud base (Jones et al., 2011). The PBL decoupling strength is computed at every cloud physics timestep (dt = 5 minutes), averaged over samples where decoupled PBL is detected.

We also diagnose the cloud-top entrainment efficiency (Bretherton et al., 2007) in different model configurations to further clarify the physical mechanisms associated with the parameter adjustments. Cloud-top entrainment efficiency is defined as $A = w_e \Delta b\, z_i / w_*^3$, where $w_e$ is the entrainment rate computed by differencing the resolved vertical motion and change of inversion height ($z_i$), $\Delta b$ is the virtual potential temperature jump scaled into buoyancy jump ($\Delta b = g\frac{\Delta \theta_v}{\theta_{ref}}$) where the reference virtual potential temperature $\theta_{ref}$ is 300 K, and $w_*$ is the convective velocity ($w_* = (2.5\int_0^{z_i} \overline{w'b'} dz)^{1/3}$) that measures the buoyancy integrated over the boundary layer where b' is the buoyancy perturbation and $\overline{w'b'}$ is the buoyancy flux. Figure 8 shows that the largest differences are again a result of changes made in CLUBB. As $\overline{w'^2}$ is reduced, EAMv1_CLUBB produces shallower PBL which also reduces cloud-top entrainment efficiency. In EAMv1_MP, the enhancement of liquid and ice sedimentation also reduces entrainment efficiency (Bretherton et al., 2007). EAMv1_SGV generally enhances the surface fluxes and produces deeper and relatively less stable PBL, leading to enhanced mixing between the PBL and the free troposphere.

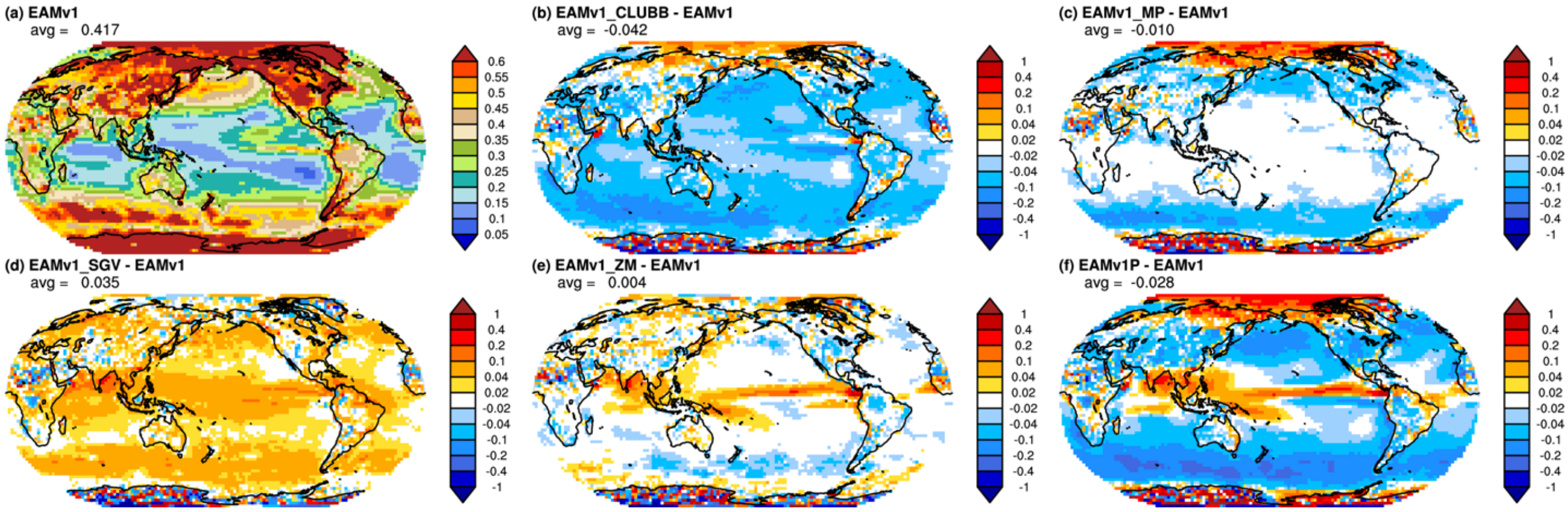

**Figure 8.** Cloud-top entrainment efficiency (Bretherton et al., 2007). The cloud-top entrainment efficiency is computed at every cloud physics timestep (dt = 5 minutes) of the model.

The changes in PBL decoupling strength and cloud-top entrainment efficiency shown in Figs 6 and 7 are consistent with expectations and understanding of the physical mechanisms connecting the parameter adjustments, CREs, and PBL properties, even though they are not directly controlled by any tunable parameters. However, currently there is no global observational estimate for decoupling frequency and cloud-top entrainment efficiency so we cannot assert that the recalibration improves these physical mechanisms affecting clouds. Future studies that derive decoupling frequency and cloud-top entrainment efficiency, as well as other important cloud controlling factors, from field campaign measurements for evaluating models in particular regions and time periods would be highly valuable.

### 3.2 Precipitation

Table 8 shows the global mean precipitation characteristics. We find that adjustments to the ZM scheme (e.g., reducing the convective autoconversion efficiency and convective cloud fraction) lead to a reduction of convective precipitation (PRECC) and an increase of large-scale precipitation (PRECL). Here convective precipitation refers to the precipitation produced by the ZM deep convection parameterization and large-scale precipitation refers to the precipitation produced by the MG2 cloud microphysics parameterization. While EAMv1 produces more convective precipitation than large-scale precipitation, the revised model EAMv1P corrects this bias so that the model is in better agreement with observational estimates (Yang et al., 2013). The shift from convective to large-scale precipitation is expected to improve precipitation characteristics (Yang et al., 2013) because more detailed cloud microphysics processes are considered for large-scale clouds.

Nevertheless, the common bias in ESMs of producing frequent drizzle and light precipitation is pronounced in EAMv1 and adjustments of parameters have only a marginal impact. This suggests that the precipitation PDF bias is not related to parametric uncertainty and perhaps is attributed to model's structural deficiency such as issues with the trigger and closure in

its deep convection scheme, as well as the coarse resolution insufficient to simulate strong moisture convergence or dependency of precipitation formation on unresolved mesoscale forcing. Recent studies indicated that using an improved convective trigger (Xie et al., 2019) or incorporating a stochastic convection scheme (Wang et al., 2021) into ZM could help address the "too-frequent-too-weak" precipitation biase in EAMv1. Lastly, we show that the fraction of large-scale precipitation produced by autoconversion ($R_{auto}$ in Table 8) in EAMv1 is already much lower than its predecessor model CAM5 even at 0.25-degree horizonal grid spacing (Ma et al., 2015), and the changes in EAMv1_MP further reduce the autoconversion fraction. This change will affect the model estimate of aerosol indirect effects (Posselt and Lohmann, 2009;Wang et al., 2012;Gettelman et al., 2013).

**Table 8.** Global mean 10-year averaged precipitation fields of EAMv1, EAMv1_CLUBB, EAMv1_MP, EAMv1_SGV, EAMv1_ZM, and EAMv1P. Relevant precipitation variables listed here are total, convective, and large-scale precipitation rates (PRECT, PRECC, and PRECL; unit = mm day$^{-1}$); ratio of deep convective precipitation to total precipitation ($R_{conv}$); frequency of occurrence (unit = %) of no precipitation ($FREQ_{dry}$), drizzle with precipitation rates less than 0.5 mm day$^{-1}$ ($FREQ_{drizzle}$), light precipitation with precipitation rates between 0.5 and 8 mm day$^{-1}$ ($FREQ_{light}$), moderate precipitation with precipitation rates between 8 and 80 mm day$^{-1}$ ($FREQ_{moderate}$), and heavy precipitation with precipitation rates exceeding 80 mm day$^{-1}$ ($FREQ_{heavy}$); and ratio of autoconversion to total precipitation ($R_{auto}$).

| Variable | EAMv1 | EAMv1_CLUBB | EAMv1_MP | EAMv1_SGV | EAMv1_ZM | EAMv1P |
|---|---|---|---|---|---|---|
| PRECT | 3.07 | 3.02 | 3.10 | 3.09 | 3.02 | 3.01 |
| PRECC | 1.76 | 1.74 | 1.79 | 1.82 | 1.32 | 1.38 |
| PRECL | 1.32 | 1.29 | 1.32 | 1.27 | 1.70 | 1.63 |
| $R_{conv}$ | 0.57 | 0.58 | 0.58 | 0.59 | 0.44 | 0.46 |
| $FREQ_{dry}$ | 6.38 | 6.30 | 6.41 | 6.50 | 5.98 | 6.13 |
| $FREQ_{drizzle}$ | 50.94 | 51.08 | 50.59 | 50.16 | 50.02 | 49.07 |
| $FREQ_{light}$ | 34.23 | 34.33 | 34.44 | 34.84 | 36.35 | 37.17 |
| $FREQ_{moderate}$ | 8.41 | 8.26 | 8.52 | 8.46 | 7.57 | 7.56 |
| $FREQ_{heavy}$ | 0.05 | 0.05 | 0.05 | 0.05 | 0.10 | 0.09 |
| $R_{auto}$ | 0.20 | 0.21 | 0.12 | 0.20 | 0.16 | 0.11 |

As discussed in Rasch et al. (2019) and shown in Figure 9, EAMv1 produces high annual mean precipitation over the globe, over high elevation, over the Maritime Continent, and in the central Pacific, but low annual mean precipitation over Amazonia and oceanic TWP. With an improved cloud distribution, we find the precipitation simulation improves as well. Figure 9 shows that tropical precipitation is greatly improved. EAMv1_SGV enhances precipitation in TWP, eastern Pacific, and Amazonia, while EAMv1_CLUBB and EAMv1_ZM reduce precipitation in the central Pacific. This suggests that the displaced Walker circulation in EAMv1 is significantly improved in the recalibrated model. EAMv1_SGV also reduces precipitation bias over high elevation regions such as the Andes and Himalayas likely through non-local circulation response. We also find an unexpected improvement from the ZM changes by reducing the double ITCZ bias. While the physical mechanism remains unclear and requires further investigation, our results corroborate the finding of Song and Zhang (2018) that the double ITCZ bias is sensitive to the adjustments in the deep convection parameterization, which affects the tropical clouds (and energy budget) and precipitation directly and the large-scale circulations indirectly.

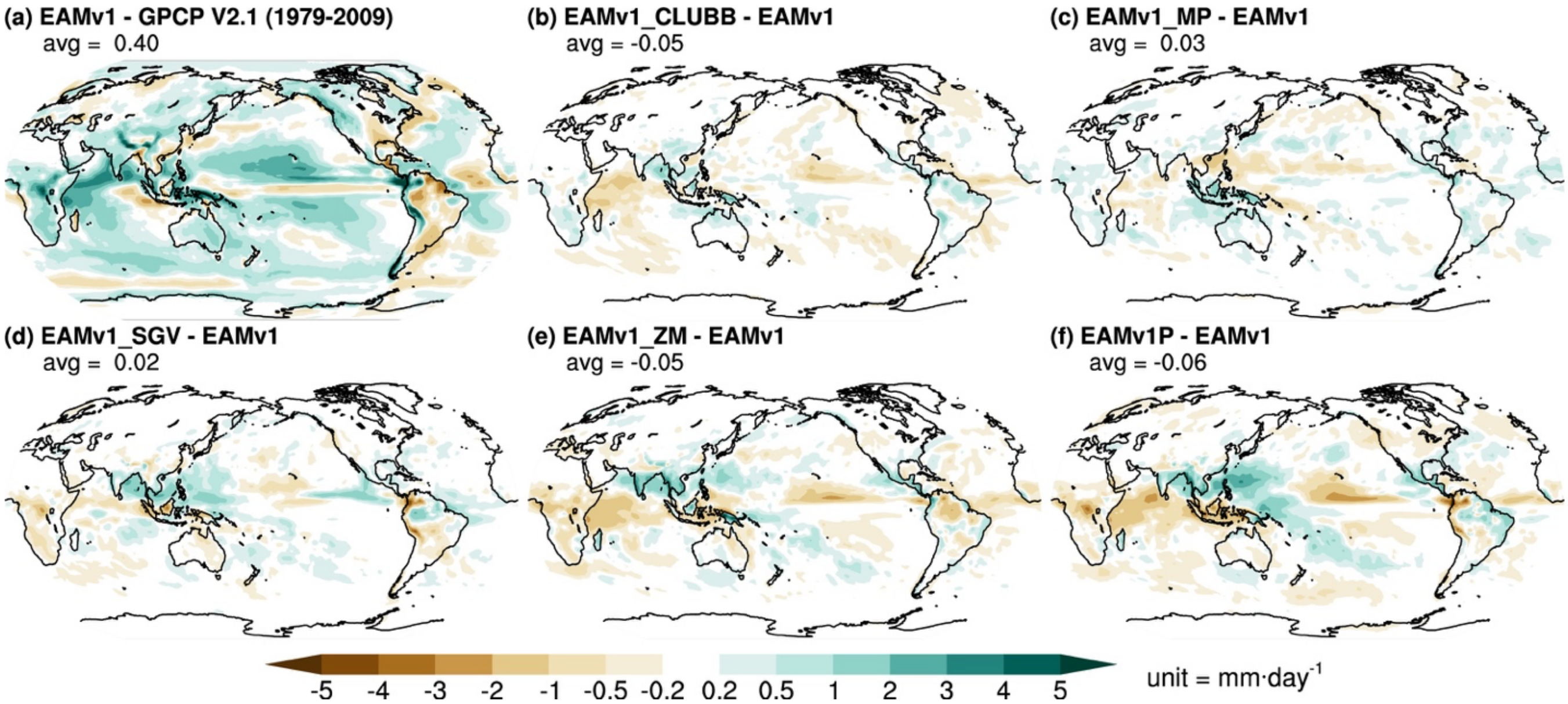

**Figure 9.** Total precipitation rate differences (unit = mm day$^{-1}$). The Global Precipitation Climatology Project (GPCP) version 2.1 dataset (Huffman et al., 2001) is used for the comparison with EAMv1.

In summary, the recalibrated model with improved clouds also produces more realistic present-day precipitation climatology. Pronounced precipitation biases in the tropics, over land, and over high elevation are significantly reduced. The improved realism of the precipitation distribution is consistent with the improved cloud distribution. These improvements lead to a more realistic atmospheric circulation and positive impacts on other aspects of the simulated atmosphere. The remaining biases in tropical clouds and precipitation could be related to coarse model resolution which fails to resolve islands, mesoscale convection, and small-scale meteorological fields (Wang et al., 2018), or the deficiency in representing the triggering of deep convection (Xie et al., 2019). The lack of representing ice clouds in CLUBB can also contribute to remaining biases in mid- and high latitudes (Zhang et al., 2020).

### 3.3 Other aspects of the present-day climate

Our recalibration is governed by an understanding of the physical mechanisms present in the atmosphere and their representation in parameterizations. Our effort has focused on improving the CREs across cloud regimes. Improvements to clouds and precipitation have been accomplished that are consistent with our expectations, but evaluation of other aspects of the simulated present-day climate is essential. If many other aspects are also improved, our confidence in the underlying physics in the model will be increased. Otherwise, we are forced to suspect that the model achieves its behavior through compensating biases.

Near-surface air temperature is an important state variable for validating the fidelity of the ESMs. Both dynamical and physical processes affect the temperature field, so an appropriate balance between these processes is essential for producing a realistic simulation of present-day conditions. Therefore, the near-surface air temperature can also be viewed as a minimum requirement for providing some confidence in projections of future climate. However, like many weather and climate models (Morcrette et al., 2018), EAMv1 produces significant near-surface air temperature biases. The Northern Hemisphere (NH) high latitudes exhibit a 1-5 K warm bias and there are cold biases in other places (Figure 10). The warm high latitude bias and the cold tropical bias produce a weaker equator-to-pole temperature gradient, which can cause errors in mid-latitude baroclinicity, storm tracks, and large-scale circulations. It can also lead to excessive melting of sea ice and land ice, which has adverse impacts on ocean circulation. Figure 10 shows that the parameter adjustments that aim to improve CREs generally improve the near-surface temperature, and the changes in EAMv1_MP lead to the largest improvements. This suggests that the liquid cloud bias in EAMv1 due to the underactive WBF process coupled with the CNT-based IN parameterization may be responsible for the near-surface temperature bias. Strong liquid-to-ice conversion improves the CREs and subsequently affects the near-surface temperature, which will further impact circulations and affect other aspects of the Earth's climate.

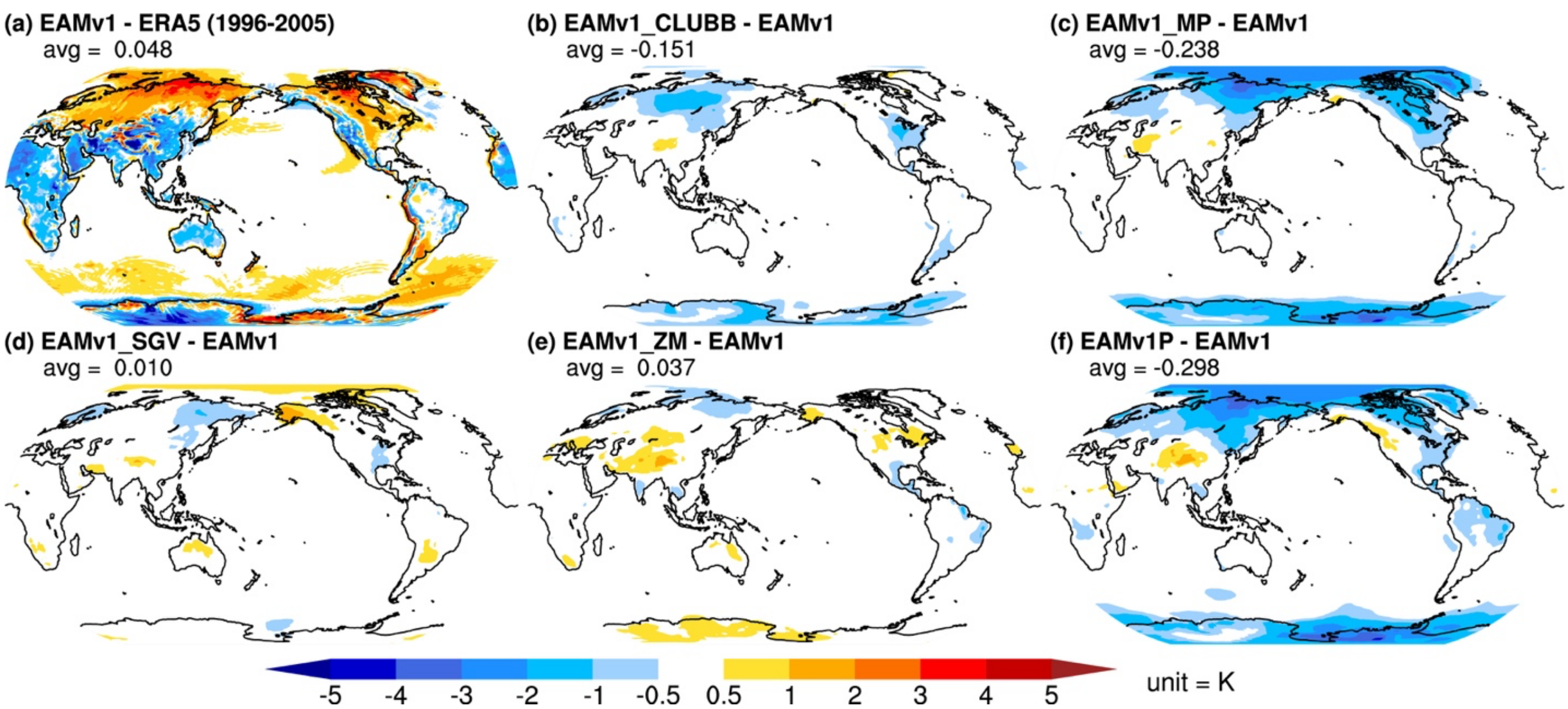

**Figure 10.** 2-meter height air temperature (unit = K). The ERA5 reanalysis is used for the comparison with EAMv1.

Surface winds affect the physical climate and the biogeochemical cycle in a variety of ways. In EAMv1, surface winds affect surface flux of heat, moisture, and momentum, which influence the thermodynamic properties in the PBL but also more generally the atmospheric energy and water cycles. The emissions of sea spray aerosols and mineral dust are a function of

surface winds. Over the ocean, surface winds drive the ocean surface currents and influence the mixed layer depth, heat budget, and carbon uptake in the ocean. Figure 11 shows that surface winds in EAMv1 are significantly stronger than those in the MERRA-2 reanalysis, especially in the Southern Ocean and North Atlantic. In the tropical Pacific Ocean, the trade easterlies are too strong, which pushes the cold tongue into the Indo-Pacific warm pool. The wind direction biases are reduced in EAMv1_SGV when the gustiness parameterization is enabled such that the subgrid winds are accounted for in surface flux calculations. EAMv1_ZM also shows some minor improvements in TWP. Combining all the model changes, the revised model EAMv1P shows significant improvements in surface winds in many parts of the tropics, North Atlantic, and Southern Ocean. In the fully coupled E3SM, these improvements may lead to more realistic ocean circulations as well as ocean-atmosphere exchange of heat, moisture, momentum, trace gases, and aerosols.

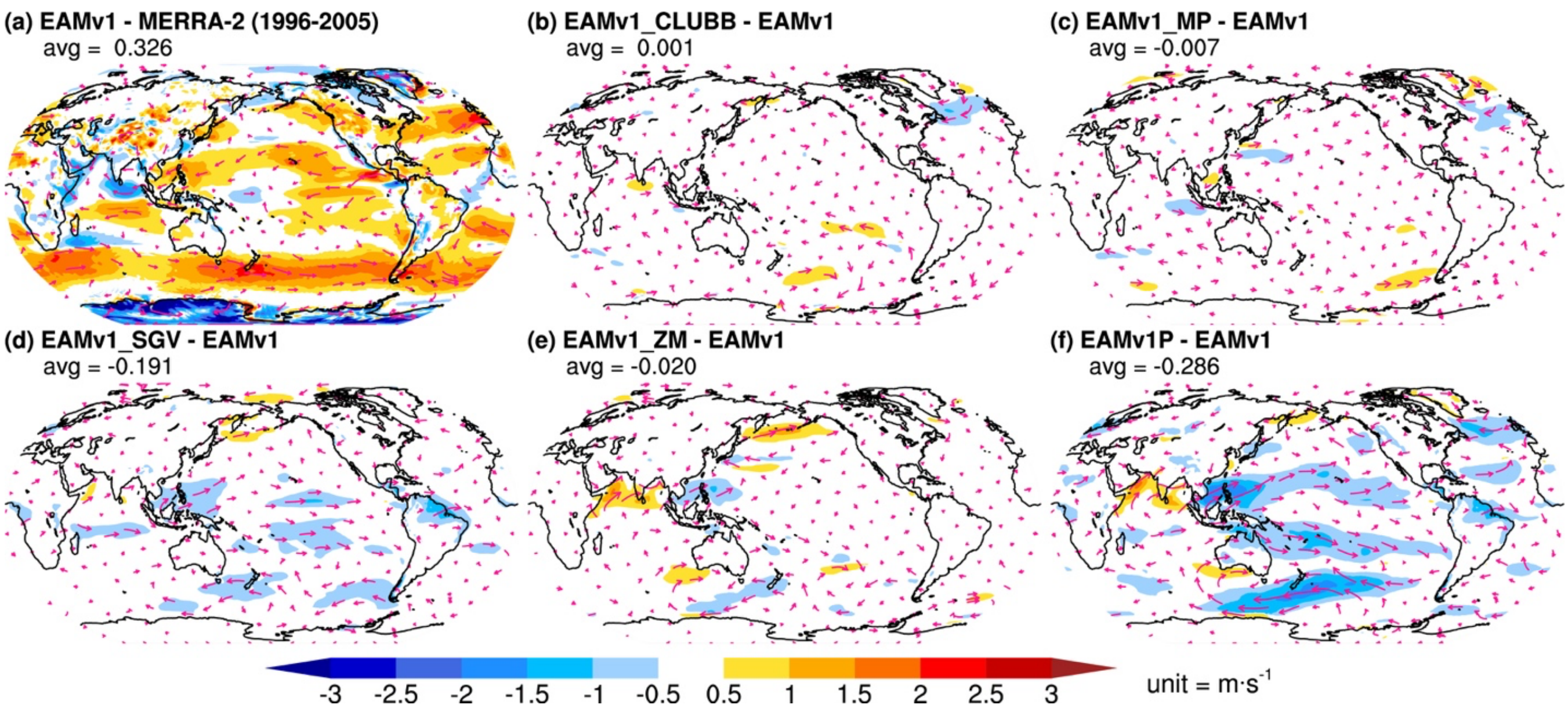

**Figure 11.** Winds at the lowest model level. The surface winds from the Modern-Era Retrospective Analysis for Research and Applications Version 2 (MERRA-2) (Gelaro et al., 2017) is used for the comparison with EAMv1.

Although our recalibration is only targeted to improve CRE features (Figure 4), those changes can affect aerosols as well because cloud processing is an important sink in the aerosol lifecycle. Figure 12 shows that the changes in EAMv1_MP and EAMv1_ZM increase the aerosol loading, while EAMv1_SGV produces lower aerosol loading. The changes in aerosol loading are partially due to the changes in wet scavenging. In EAMv1_MP, the reduction of supercooled liquid water path increases aerosol loading in mid- and high latitudes. EAMv1_SGV enhances the surface moisture flux which also increases wet scavenging, and the weakened convective autoconversion in EAMv1_ZM reduces the wet removal of aerosols. We also find that the revisions have reduced dust emissions over the Sahara because of the weakened turbulence in EAMv1_CLUBB.

Collectively, the recalibrated model EAMv1P reduces the aerosol optical depth ($\tau_{aer}$) biases in the NH mid- and high- latitudes, in the tropics, and over land in general. There are, however, remaining $\tau_{aer}$ biases in the subtopics, eastern Pacific, eastern Atlantic, and Southern Ocean.

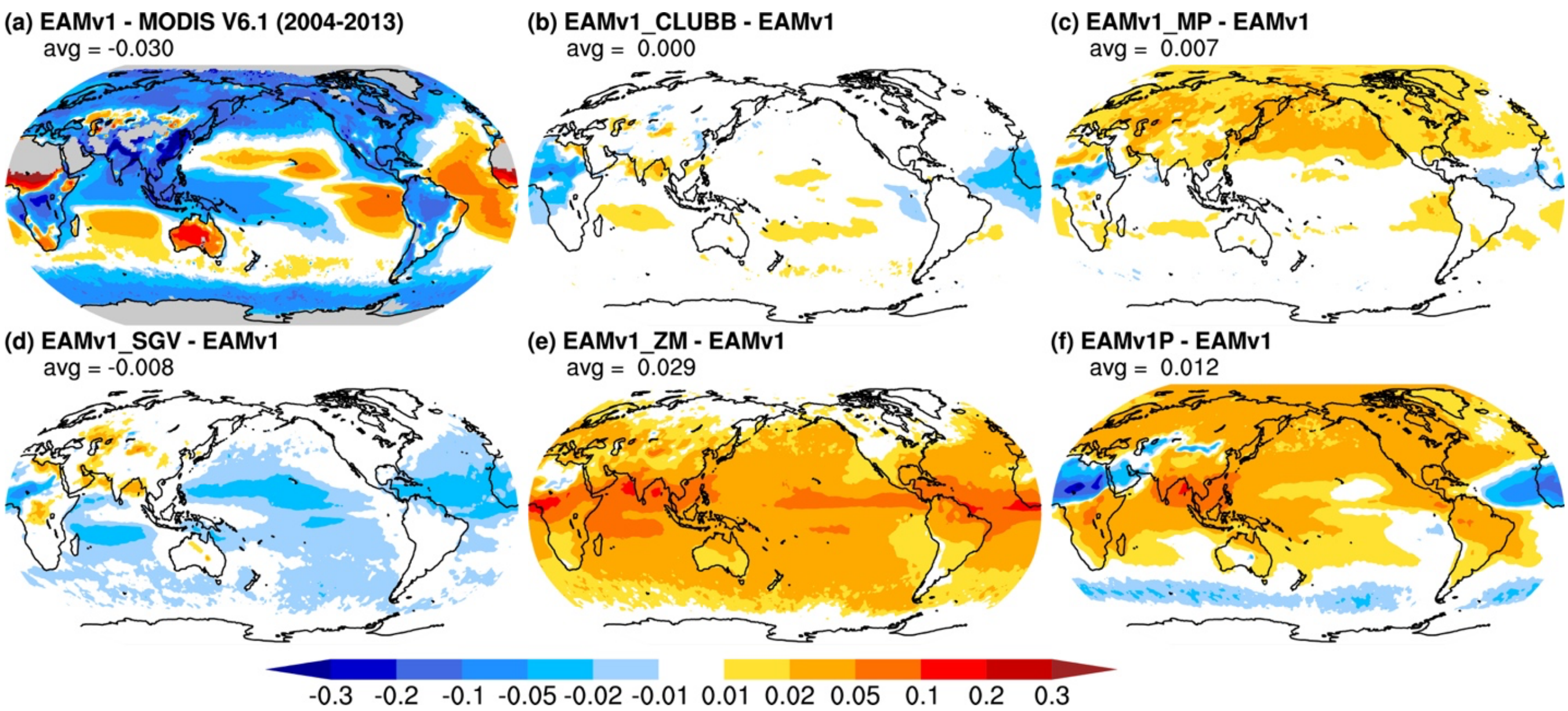

**Figure 12.** Clear-sky aerosol optical depth ($\tau_{aer}$). The MODIS onboard Aqua $\tau_{aer}$ data product (Levy et al., 2013) is used for comparison with EAMv1. Model clear-sky $\tau_{aer}$ is sampled at 1:30 PM local time.

In addition to improvements in near-surface temperature, surface winds, and column-integrated aerosols, we observe improvements to sea level pressure (SLP) as well as temperature and wind fields in the recalibrated model EAMv1P (Figure 13). Many other aspects of the climate are carefully evaluated using E3SM standard diagnostics (https://portal.nersc.gov/project/e3sm/beharrop/EAMv1P/). We find that the recalibrated model shows improvements in most aspects of the simulated present-day climate (which are not tuning targets), and small or no degradation in others. We conclude that when improvements in simulating clouds across regimes are achieved by applying adjustments based on an understanding of the physical mechanisms, those changes are manifested by more realistic simulation of many features of the global atmosphere. Because the correct response of the nonlinear climate system depends on both realistic base state and realistic process representations, the improved realism in the recalibrated model EAMv1P provides greater confidence in estimating the responses of the climate system to anthropogenic forcings and, ultimately, the ECS.

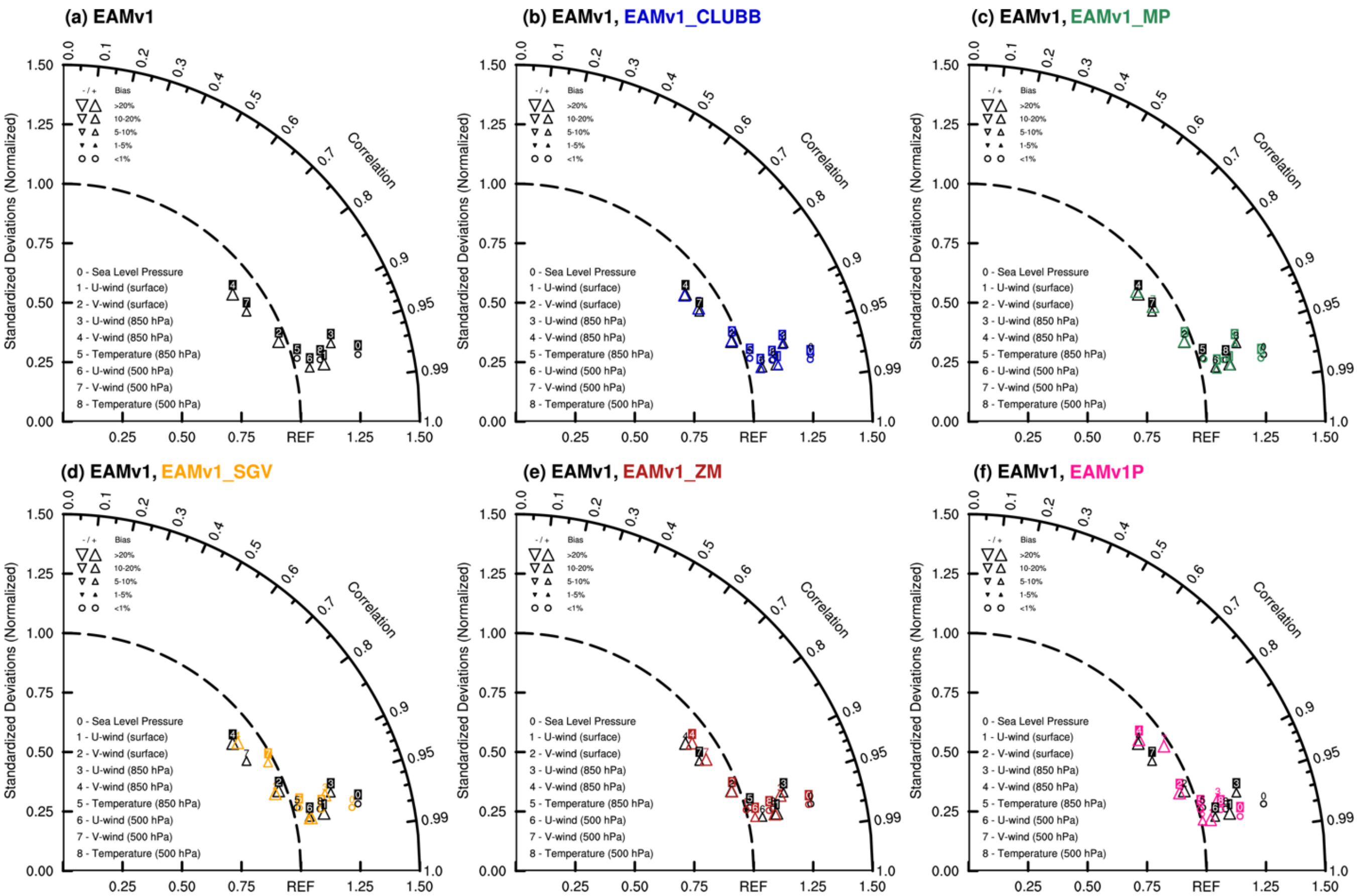

**Figure 13.** Taylor diagram comparing sea level pressure, temperature, and winds in EAMv1, EAMv1_CLUBB, EAMv1_MP, EAMv1_SGV, EAMv1_ZM, and EAMv1P with the ERA5 reanalysis.

### 3.4 Responses to anthropogenic aerosols

The role of aerosols in the climate system is a major uncertainty in projections of Earth's future climate as well as in interpreting how the climate has been forced over recent decades. The uncertainty has been attributed to both a lack of understanding of aerosol emissions in pre-industrial times (Carslaw et al., 2013) as well as uncertainties associated with modeling aerosol and cloud processes (Regayre et al., 2018;Yoshioka et al., 2019). E3SMv1 produces notable biases in the historical evolution of surface temperature due to a combination of high ECS (from cloud feedback) and strong aerosol forcing, both of which are likely to be too large (Golaz et al., 2019). In this section, we assess the cloud and precipitation responses to anthropogenic aerosols in the recalibrated model where processes influencing aerosols and clouds operate differently from EAMv1 and the simulated present-day atmosphere is more realistic than that in EAMv1. Our goal is to understand the impacts and the physical mechanisms of the parameter adjustments on cloud and precipitation responses to aerosols. The effects of anthropogenic
aerosols are assessed by differencing paired simulations where one uses the present-day aerosol emissions and the other uses the pre-industrial aerosol emissions (see Section 2 for the experiment design).

Table 9 shows the global mean net total $ERF_{ant}$ in EAMv1 is quite low compared to CMIP5 (Forster et al., 2013) and other previous generation models (Kiehl, 2007). This is mostly attributed to the aerosol ERF ($ERF_{aer}$) (Golaz et al., 2019). EAMv1_MP increases $ERF_{ant}$, but other parameter adjustments lower $ERF_{ant}$ so that the recalibrated model EAMv1P produces about the same $ERF_{ant}$. $ERF_{aer}$ comprises the ERF associated with aerosol-radiation interactions ($ERF_{ari}$) and aerosol-cloud interaction ($ERF_{aci}$), and aerosol-induced surface albedo changes. The $ERF_{aer}$ is computed by differencing all-sky TOA radiative flux between paired fixed SST simulations with present-day and pre-industrial aerosol emissions (Hansen et al., 2005), which is referred to as ERF_fSST in Forster et al. (2016). $ERF_{aci}$ is defined as the clean-sky TOA CRE difference (Ghan, 2013). Note that the Ghan (2013) method removes the direct radiative effect from the anthropogenic aerosols on CREs, producing stronger $ERF_{aci}$ (-1.48 W $m^{-2}$) compared to the Boucher et al. (2013) method (~-1 W $m^{-2}$) used in Wang et al. (2020), which assumes that $ERF_{aci}$ is the residual between $ERF_{ari+aci}$ and $ERF_{ari}$. EAMv1 produces slightly weaker net $ERF_{aer}$ (-1.42 W $m^{-2}$) and $ERF_{aci}$ (-1.48 W $m^{-2}$) than its predecessor CAM5's -1.47 and -1.53 W $m^{-2}$, respectively (Ghan et al., 2012). EAMv1's $ERF_{aer}$ falls within the 68% confidence range of -1.6 to -0.6 W $m^{-2}$ (where the 90% confidence range is between -2.0 and -0.4 W $m^{-2}$) estimated recently by considering various lines of evidence including models, observations, theories, energy balance requirements, and observed temperature constraints (Bellouin et al., 2020).

In EAMv1_MP, $ERF_{aci}$ is significantly weakened primarily due to 1) the reduction of supercooled liquid clouds in the NH storm track from tuning up the WBF process, because the $ERF_{aci}$ due to aerosol effects on liquid clouds is reduced; and 2) the reduction of sulfate aerosols participating in homogeneous ice nucleation by increasing the size threshold of sulfate aerosols. Since $ERF_{aci}$ is mostly attributed to aerosols effects on liquid clouds in EAMv1, reducing liquid clouds reduces $ERF_{aci}$. Conversely, changes introduced in EAMv1_ZM enhance $ERF_{aci}$. Since the ZM scheme does not consider detailed cloud microphysical processes, this enhancement is likely due to the overall increase of cloudiness as shown in Figure 1. Collectively, the net $ERF_{aci}$ and $ERF_{aer}$ in EAMv1P remain about the same as EAMv1, but EAMv1P produces significantly weaker $ERF_{aci,sw}$ and $ERF_{aci,lw}$.

The first-order response of the hydrological cycle is associated with changes in shortwave radiation and that of surface temperature is associated with changes in longwave radiation (Dhara, 2020). Because $ERF_{aer}$ is reduced for both shortwave and longwave in EAMv1P, the recalibrated model shows reduced aerosol-induced response in precipitation (Table 9) and land surface temperature (Table 9), even though the net ERFaer is about the same. Furthermore, the $\tau_{aer}$ difference between the paired simulations with present-day and pre-industrial aerosol emissions ($\Delta\tau_{aer}$) in EAMv1P agrees much better with estimates from model ensembles (Watson-Parris et al., 2020) and from an estimate based on a combination of models and observations

(Kinne et al., 2006) than that in EAMv1. Because $\Delta\tau_{aer}$ is significantly larger in EAMv1P than EAMv1 while $ERF_{aci}$ in the two model configurations are similar, the sensitivity of CREs to aerosol perturbations (i.e., the change of CRE per unit aerosol perturbation) is lower in EAMv1P.

**Table 9.** Global mean 10-year averaged total $ERF_{ant}$ derived from paired simulations with present-day and pre-industrial forcings; and shortwave, longwave, and net $ERF_{aer}$ and shortwave, longwave, net $ERF_{aci}$ (unit = W $m^{-2}$), and the difference in total precipitation rate (PRECT, unit = mm $day^{-1}$), land surface temperature (Ts; unit = K), and aerosol optical depth ($\tau_{aer}$) difference between paired simulations with present-day and pre-industrial aerosol emissions.

| Variable | EAMv1 | EAMv1_CLUBB | EAMv1_MP | EAMv1_SGV | EAMv1_ZM | EAMv1P |
|---|---|---|---|---|---|---|
| $ERF_{ant}$ | 1.19 | 1.19 | 1.48 | 1.05 | 0.97 | 1.24 |
| $ERF_{aer}$ | -1.42 | -1.46 | -1.09 | -1.55 | -1.72 | -1.46 |
| $ERF_{aer,sw}$ | -2.19 | -2.24 | -1.55 | -2.28 | -2.36 | -1.72 |
| $ERF_{aer,lw}$ | 0.76 | 0.78 | 0.46 | 0.73 | 0.64 | 0.26 |
| $ERF_{aci}$ | -1.48 | -1.53 | -1.25 | -1.53 | -1.79 | -1.46 |
| $ERF_{aci,sw}$ | -2.02 | -2.11 | -1.48 | -2.11 | -2.24 | -1.55 |
| $ERF_{aci,lw}$ | 0.54 | 0.58 | 0.23 | 0.58 | 0.45 | 0.08 |
| ΔPRECT | -0.028 | -0.024 | -0.024 | -0.026 | -0.025 | -0.021 |
| ΔTs | -0.20 | -0.01 | -0.05 | -0.09 | -0.04 | -0.10 |
| $\Delta\tau_{aer}$ | 0.024 | 0.023 | 0.026 | 0.023 | 0.029 | 0.033 |

Figure 14 shows that the recalibration leads to smaller magnitude of both positive and negative $ERF_{aci}$ in most places. The aerosol-induced strong warming in the Arctic and strong cooling in the NH storm track, East Asia, and North America are reduced, indicating a weaker local CRE response to aerosols in EAMv1P. EAMv1_MP again produces the most significant reduction, most likely due to the more effective WBF process that reduces the supercooled liquid clouds. Other changes introduced in EAMv1_MP may also contribute to the weaker $ERF_{aci}$ in East Asia, Northeast Pacific, and North America, including 1) enhancing the sedimentation of ice and liquid cloud droplets; 2) reducing the sulfate aerosols available for homogeneous ice nucleation; and 3) reducing the minimum subgrid vertical velocity used for liquid droplet nucleation. In Figure 15, we show that aerosol-induced regional land surface temperature changes in EAMv1 are reduced in all other model configurations, even though the $ERF_{aer}$ remains the same. With the reduced sensitivity of surface temperature to aerosol perturbations, the concerning signature in the historical temperature changes of significant cooling in the 1950s and the fast warming in the 1990s in E3SMv1 (Golaz et al., 2019) are likely to reduce in the recalibrated model if the cause of the biases is indeed due to the overly strong aerosol forcing.

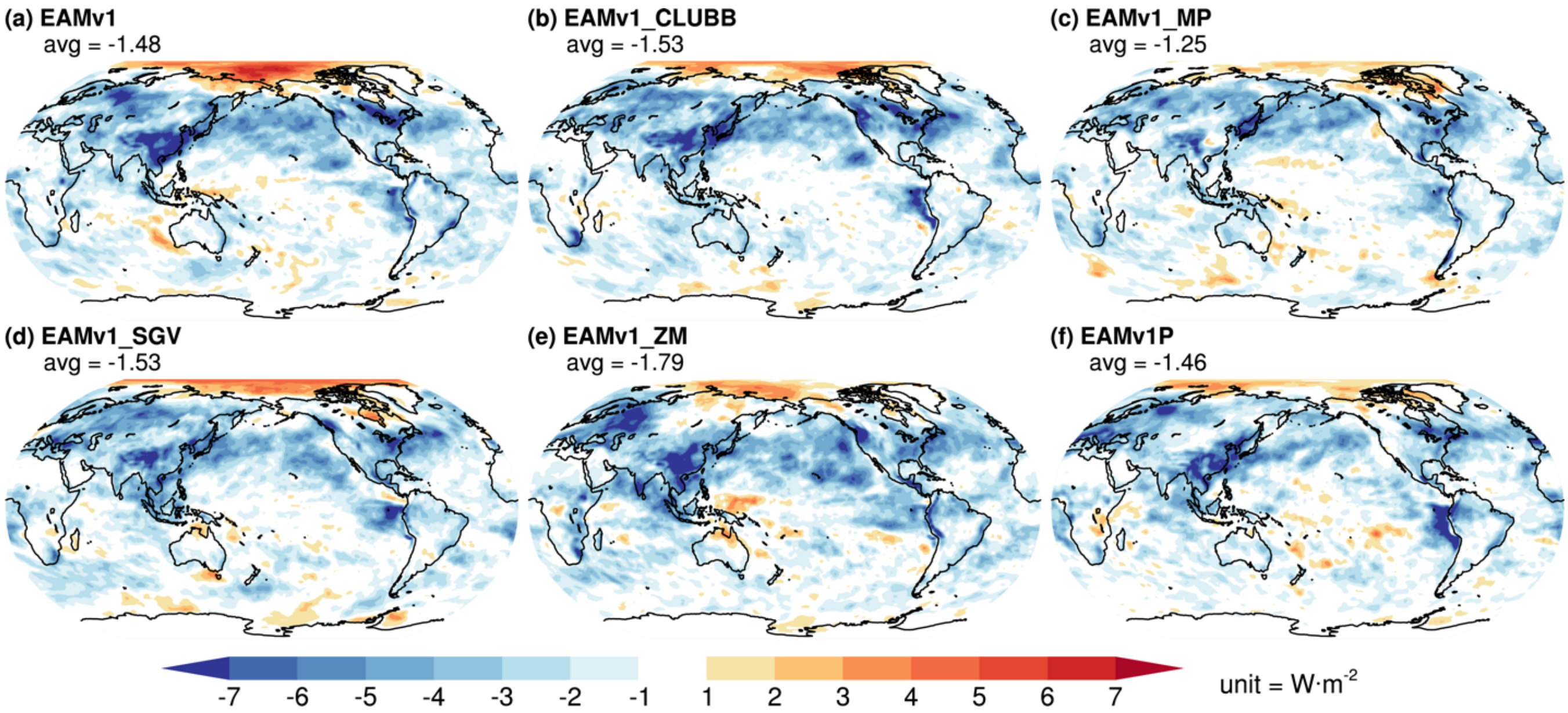

**Figure 14.** $ERF_{aci}$ estimated using the Ghan (2013) method in (a) EAMv1, (b) EAMv1_CLUBB, (c) EAMv1_MP, (d) EAMv1_SGC, (e) EAMv1_ZM, and (f) EAMv1P.

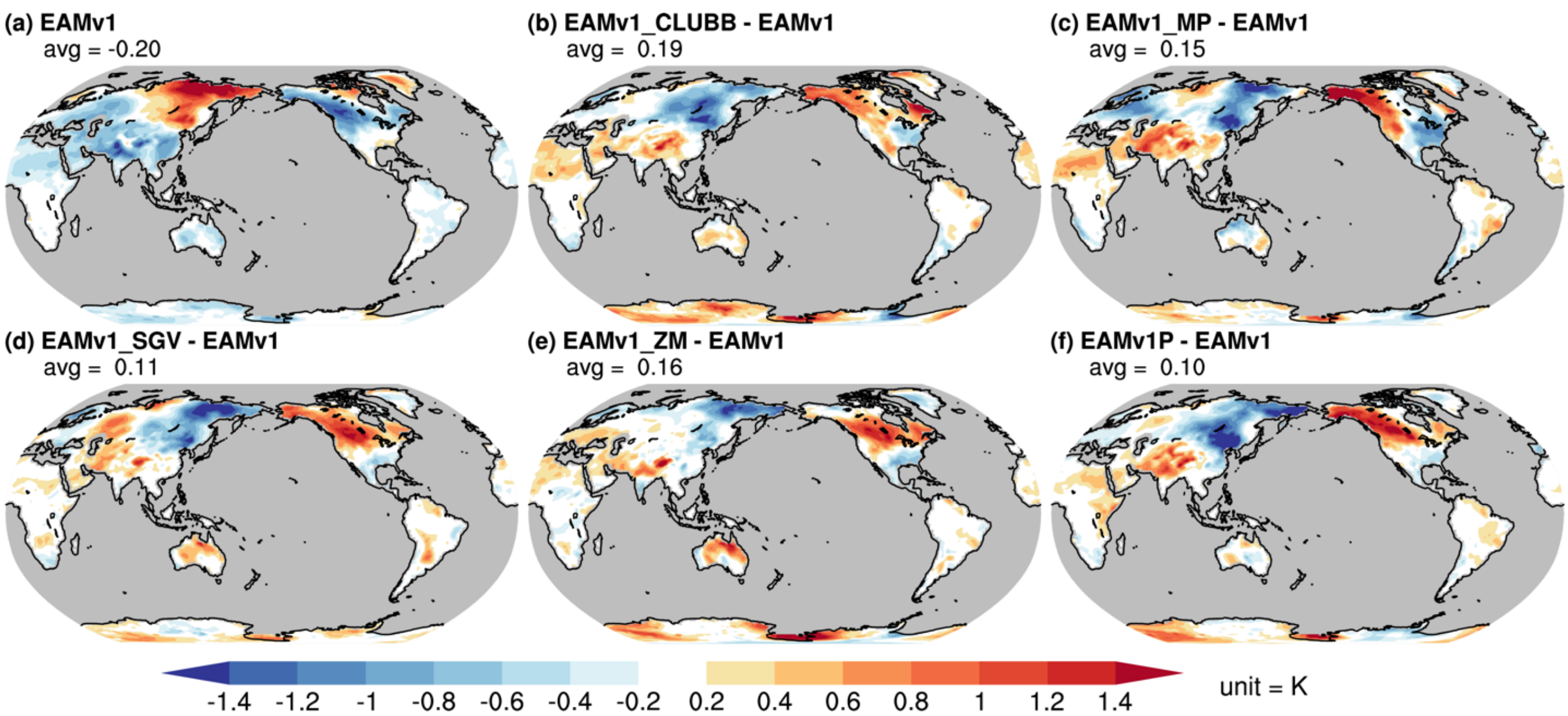

**Figure 15.** Aerosol-induced changes in land surface temperature.

Table 10 shows that the aerosol-induced change in cloud fraction remains small in all model configurations. For column-integrated condensate amount, EAMv1_MP significantly reduces the sensitivity of LWP and IWP to aerosols. EAMv1_ZM also reduces the IWP sensitivity. The droplet and ice number concentrations are highly sensitive to anthropogenic aerosols as expected, but EAMv1_MP significantly reduces the sensitivity of both $N_c$ and $N_i$ to aerosols, while EAMv1_ZM reduces only the sensitivity of $N_i$ to aerosol perturbations. Changes in cloud macro- and micro-physical properties drive cloud optical property and radiative effect changes as well. EAMv1_MP reduces the sensitivity of $\tau_{liq}$, $\tau_{ice}$, and $\tau_{snow}$ to aerosols, leading to lower sensitivity of CRE for corresponding hydrometeors to aerosol perturbations. EAMv1_ZM also reduces the sensitivity of $\tau_{ice}$ and $\tau_{snow}$ to aerosols as well as the corresponding CRE sensitivities. This is likely due to the reduction of the ice particle size detrained from deep convection, which increases $N_i$ in the unperturbed pre-industrial environment so that the ice clouds are less susceptible to aerosols. Finally, the revised model EAMv1P shows decreases in shortwave and longwave CRE responses.

**Table 10.** Same as Table 6, except that the change of cloud properties induced by anthropogenic aerosols relative to their pre-industrial values (unit = %) are shown. Variables are defined in Table 6.

| Variable | EAMv1 | EAMv1_CLUBB | EAMv1_MP | EAMv1_SGV | EAMv1_ZM | EAMv1P |
|---|---|---|---|---|---|---|
| $\Delta_R F_{cld,tot}$ | 0.37 | 0.56 | 0.20 | 0.50 | 0.48 | 0.38 |
| $\Delta_R F_{cld,low}$ | 0.42 | 0.89 | 0.38 | 0.43 | 0.71 | 0.88 |
| $\Delta_R F_{cld,med}$ | 0.93 | 1.05 | 0.74 | 0.77 | 1.01 | 0.65 |
| $\Delta_R F_{cld,hgh}$ | 0.34 | 0.15 | -0.22 | 0.56 | 0.27 | -0.17 |
| $\Delta_R LWP$ | 4.16 | 4.88 | 3.32 | 4.08 | 4.34 | 3.50 |
| $\Delta_R IWP$ | 2.33 | 2.31 | 0.78 | 2.18 | 0.85 | -0.26 |
| $\Delta_R N_c$ | 46.91 | 47.31 | 41.72 | 45.04 | 44.59 | 41.43 |
| $\Delta_R N_i$ | 15.15 | 15.02 | 9.65 | 15.23 | 7.33 | 1.03 |
| $\Delta_R \tau_{cld}$ | 11.08 | 11.51 | 9.13 | 10.67 | 10.11 | 8.13 |
| $\Delta_R \tau_{liq}$ | 16.57 | 17.62 | 14.13 | 15.92 | 16.47 | 15.15 |
| $\Delta_R \tau_{ice}$ | 6.43 | 6.06 | 3.30 | 6.08 | 2.51 | 0.47 |
| $\Delta_R \tau_{snow}$ | 0.93 | 0.71 | 0.11 | 0.84 | 0.32 | -0.31 |
| $\Delta_R \tau_{conv}$ | 1.38 | 1.33 | 1.65 | 1.93 | 2.48 | 1.15 |
| $\Delta_R \tau_{low}$ | 11.07 | 11.82 | 9.66 | 10.57 | 10.70 | 9.83 |
| $\Delta_R \tau_{hgh}$ | 8.83 | 8.80 | 3.58 | 8.98 | 5.26 | 0.87 |
| $\Delta_R SWCRE$ | 3.21 | 3.75 | 2.17 | 3.35 | 3.10 | 1.97 |
| $\Delta_R SWCRE_{liq}$ | 5.32 | 6.11 | 3.97 | 5.43 | 5.97 | 4.93 |
| $\Delta_R SWCRE_{ice}$ | 3.67 | 3.74 | 1.27 | 3.76 | 0.92 | -0.99 |
| $\Delta_R SWCRE_{snw}$ | 0.12 | -0.04 | -1.01 | 0.21 | -0.30 | -1.50 |
| $\Delta_R SWCRE_{conv}$ | -1.14 | -0.69 | -0.69 | -0.62 | -1.11 | -1.68 |
| $\Delta_R LWCRE$ | 2.17 | 2.47 | 1.05 | 2.33 | 1.60 | 0.27 |
| $\Delta_R LWCRE_{liq}$ | 3.87 | 4.28 | 2.53 | 3.90 | 4.26 | 3.47 |
| $\Delta_R LWCRE_{ice}$ | 3.76 | 3.92 | 1.78 | 3.83 | 1.94 | -0.20 |
| $\Delta_R LWCRE_{snw}$ | 0.89 | 0.72 | -0.34 | 0.94 | 0.62 | -0.71 |
| $\Delta_R LWCRE_{conv}$ | -1.88 | -0.96 | -1.16 | -1.29 | -1.66 | -1.77 |

In EAMv1, anthropogenic aerosols reduce the frequency of occurrence of light precipitation (< 2 mm day$^{-1}$) across all large-scale dynamical regimes based on large scale vertical velocity at 500 hPa, reduce light-to-moderate precipitation (< 80

mm day$^{-1}$) in strong ascending regions (< -20 hPa day$^{-1}$), and increase precipitation between 2.5 and 20 mm day$^{-1}$ in general (Figure 16). The parameter adjustments in EAMv1_MP, EAMv1_SGV, and EAMv1_ZM all lead to weakened precipitation response compared to EAMv1. In consequence, cloud and precipitation processes become less sensitive to aerosol perturbations in the recalibrated model.

In summary, the recalibration reduces the overall responses of CREs, surface temperature, and hydrological cycle to aerosols. Evaluation of the hydrological cycle response to aerosols indicates total precipitation rate is influenced globally (Table 9), regionally (not shown), and in terms of large-scale precipitation frequency of occurrence (using a joint PDF; Figure 16). However, the global mean $ERF_{ant}$, $ERF_{aer}$ and $ERF_{aci}$ remain about the same between the default model EAMv1 and the recalibrated model EAMv1P due to invariant effects of changes in $N_c$, and due to compensations in shortwave and longwave effects that vary in opposite directions. These analyses demonstrate that the global mean ERFs are insufficient for understanding or constraining the response of hydrological cycle and surface temperature to aerosols. The shortwave and longwave contribution to the total aerosol ERF as well as the spatial distribution of aerosol ERF need to be considered to understand how aerosols affect the Earth system. Furthermore, the unperturbed base state climate can play a role as well. As shown in Figure 10, the recalibrated model reduces the surface temperature bias significantly, which can lead to a more realistic response of surface temperature to forcings.

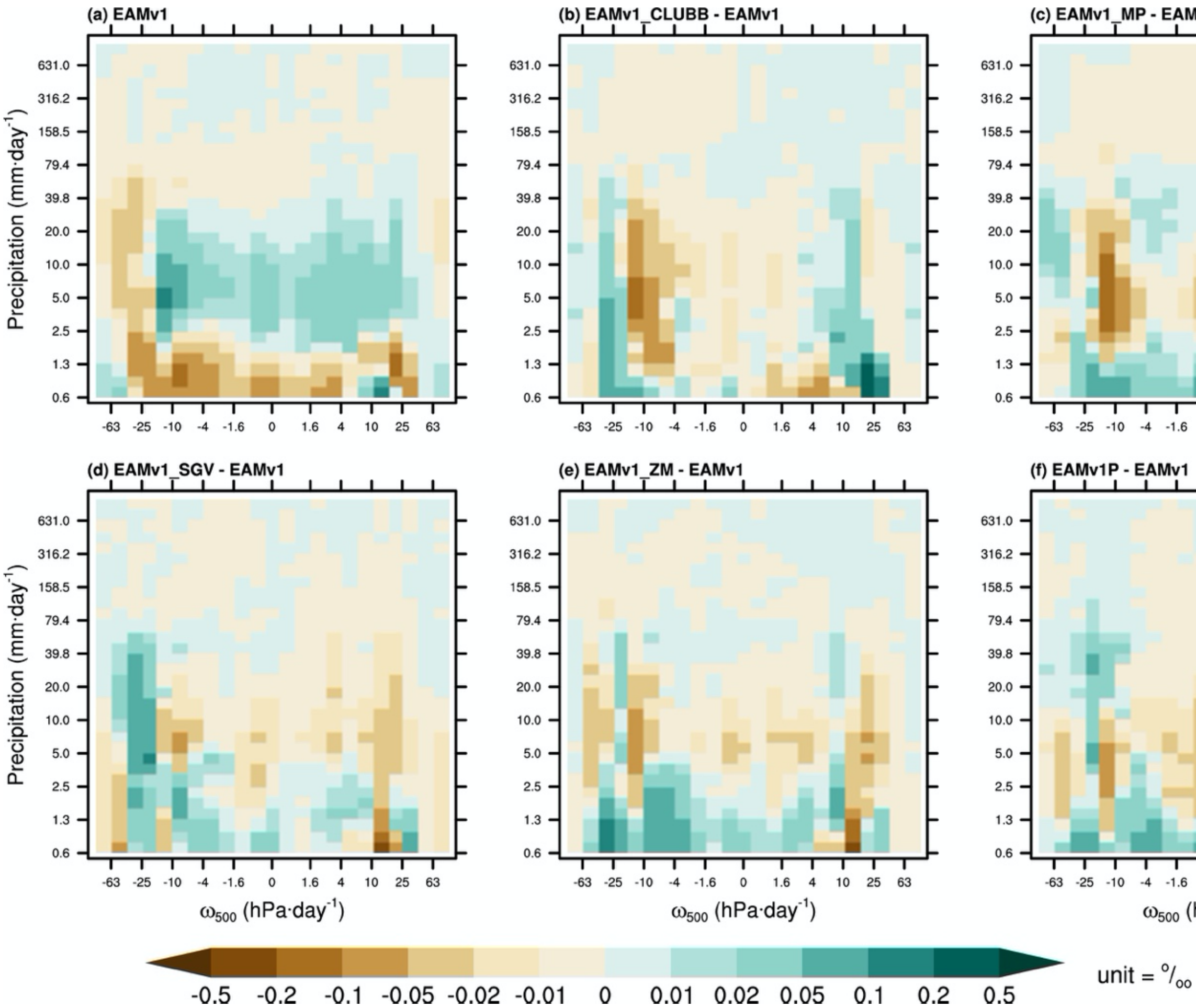

**Figure 16.** Anthropogenic aerosol-induced change in frequency of occurrence of resolved-scale precipitation as a function of vertical velocity at 500 hPa (unit = hPa day$^{-1}$) in (a) EAMv1; and differences between (b) EAMv1_CLUBB and EAMv1; (c) EAMv1_MP and EAMv1; (d) EAMv1_SGV and EAMv1; (e) EAMv1_ZM and EAMv1; and (f) EAMv1P and EAMv1. Model precipitation rates are sampled at every model time step (dt = 30 minutes).

### 3.5 Response to surface warming

The response of the Earth system to surface warming is of great scientific and societal importance. ECS values in CMIP6 span a significantly wider range (1.8 to 5.6 K) than in CMIP5, and their substantially higher multi-model mean value has been attributed to the same causes identified in E3SMv1: strong positive cloud feedbacks (Zelinka et al., 2020). In this section, we discuss the impacts of parameter adjustments on cloud and other climate feedbacks. The feedbacks are assessed using the Cess methodology (Cess et al., 1989) by contrasting the difference between a control pre-industrial simulation and a perturbed simulation with SST elevated by 4 K globally (See Section 2 for the experiment design).

Figure 17 shows that EAMv1's total climate feedback of -1.51 W m$^{-2}$ K$^{-1}$ is weaker than the CMIP5 multi-model mean (-1.6 W m$^{-2}$ K$^{-1}$), but within the inter-model spread of -1.05 to -1.95 W m$^{-2}$ K$^{-1}$ (Ringer et al., 2014). The less negative feedback suggests a faster warming in the late 20$^{th}$ century and a higher ECS, consistent with the findings in Golaz et al. (2019). EAMv1_CLUBB and EAMv1_MP produce stronger global mean feedback which will lead to lower ECS and weaker warming in the 20$^{th}$ Century, while EAMv1_ZM produces positive feedback in the tropics. The recalibrated model EAMv1P produces a stronger climate feedback of -1.74 W m$^{-2}$ K$^{-1}$, a 15% increase from EAMv1.

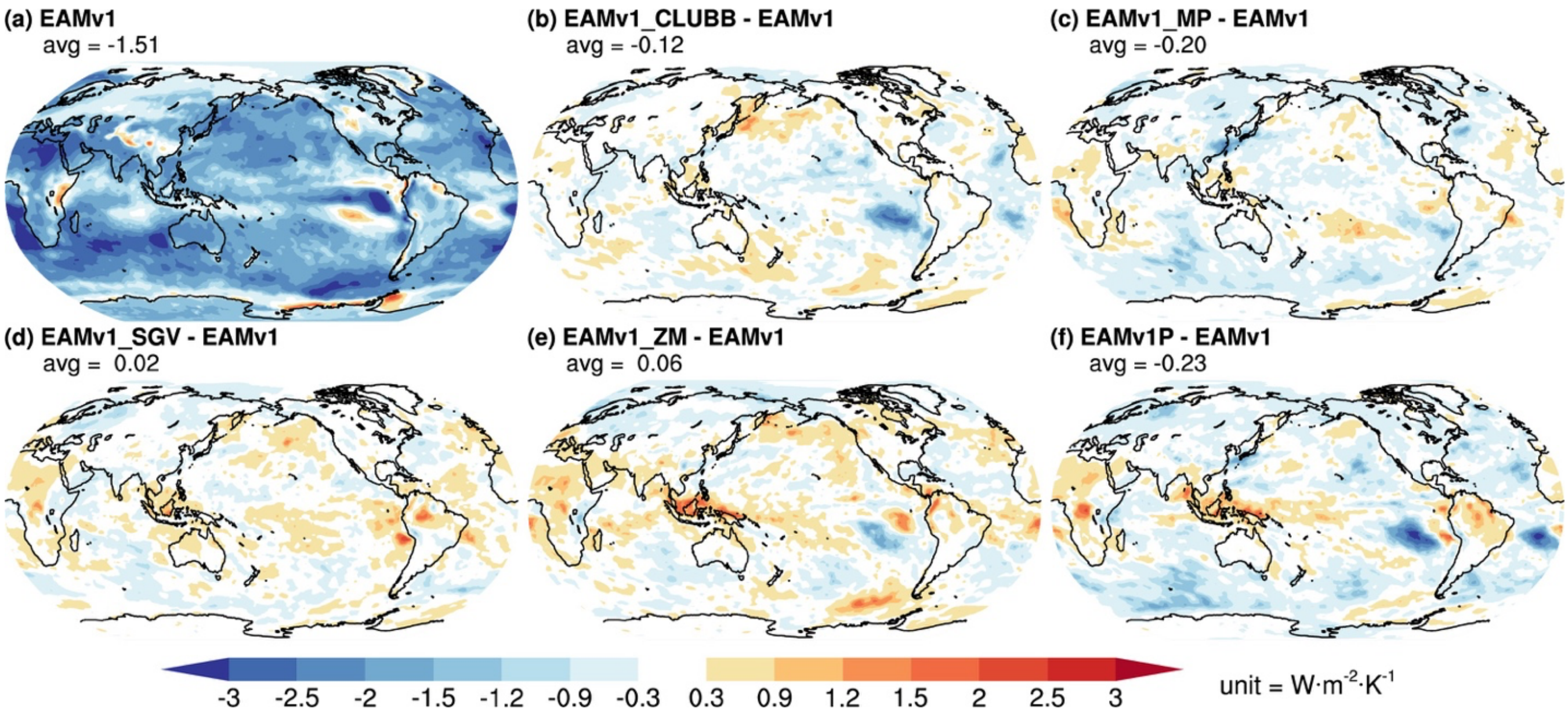

**Figure 17.** Climate feedback parameter (Cess et al., 1989).

In Figure 18, the climate and cloud feedback decomposition analysis using radiative kernels (Zelinka et al., 2012;Pendergrass et al., 2018) reveals that the non-cloud feedbacks are invariant across different model configurations and that the variation in total climate feedback is due solely to the spread in cloud feedbacks as a result of our parameter and subgrid adjustments. The total, shortwave, and longwave cloud feedbacks are weakened from 0.77, 0.35, and 0.42 W m$^{-2}$ K$^{-1}$ in EAMv1 to 0.47 (-39%), 0.20 (-43%), and 0.27 W m$^{-2}$ K$^{-1}$ (-35%) in EAMv1P. The stronger negative total climate feedback from the weakened positive cloud feedback suggest that the recalibration will produce a slower warming in the late 20$^{th}$ Century and lower ECS.

Figure 18b shows that EAMv1_CLUBB and EAMv1_MP both reduce the magnitude of shortwave cloud feedback. EAMv1_MP strengthens the negative shortwave cloud optical depth feedback likely due to the reduction of mean-state supercooled liquid in mixed phase clouds (by strengthening the WBF process). The weaker cloud feedback in

EAMv1_CLUBB comes from the reduction of cloud amount feedback. This is likely due to the fact that EAMv1_CLUBB improves the simulation of shallow Cu. Because Sc cloud amount decreases more with warming than shallow Cu (Cesana et al., 2019), producing shallow Cu rather than Sc reduces cloud amount feedback. In other words, EAMv1_CLUBB simulates a control-state climate with more Cu and less Sc than the default EAMv1, so the positive feedback from warming-induced reductions of low cloud cover is weakened because Cu are more resilient to warming than Sc. In the meantime, EAMv1_CLUBB reduces the decoupling strength and cloud-top entrainment in the Sc regime, which can also reduce the cloud amount feedback.

Contrary to the effects introduced by EAMv1_CLUBB and EAMV1_MP, EAMv1_ZM enhances total cloud feedback. Figure 18b shows that EAMv1_ZM significantly reduces both shortwave and longwave cloud optical depth feedbacks and diminishes longwave cloud amount feedback. The large reduction of the negative shortwave cloud optical depth feedback results in a stronger positive total cloud feedback. This indicates that changes made in EAMv1_ZM, particularly 1) reducing ice particle radius detrained from deep convection (ice_deep) and 2) reducing convective autoconversion (c0_ocn and c0_lnd), which make convective clouds and their anvils opaque in the present-day climate, result in a weaker sensitivity of CRE to surface warming. However, the physical mechanism relating those tuning choices to cloud feedbacks remain unclear and requires further investigation.

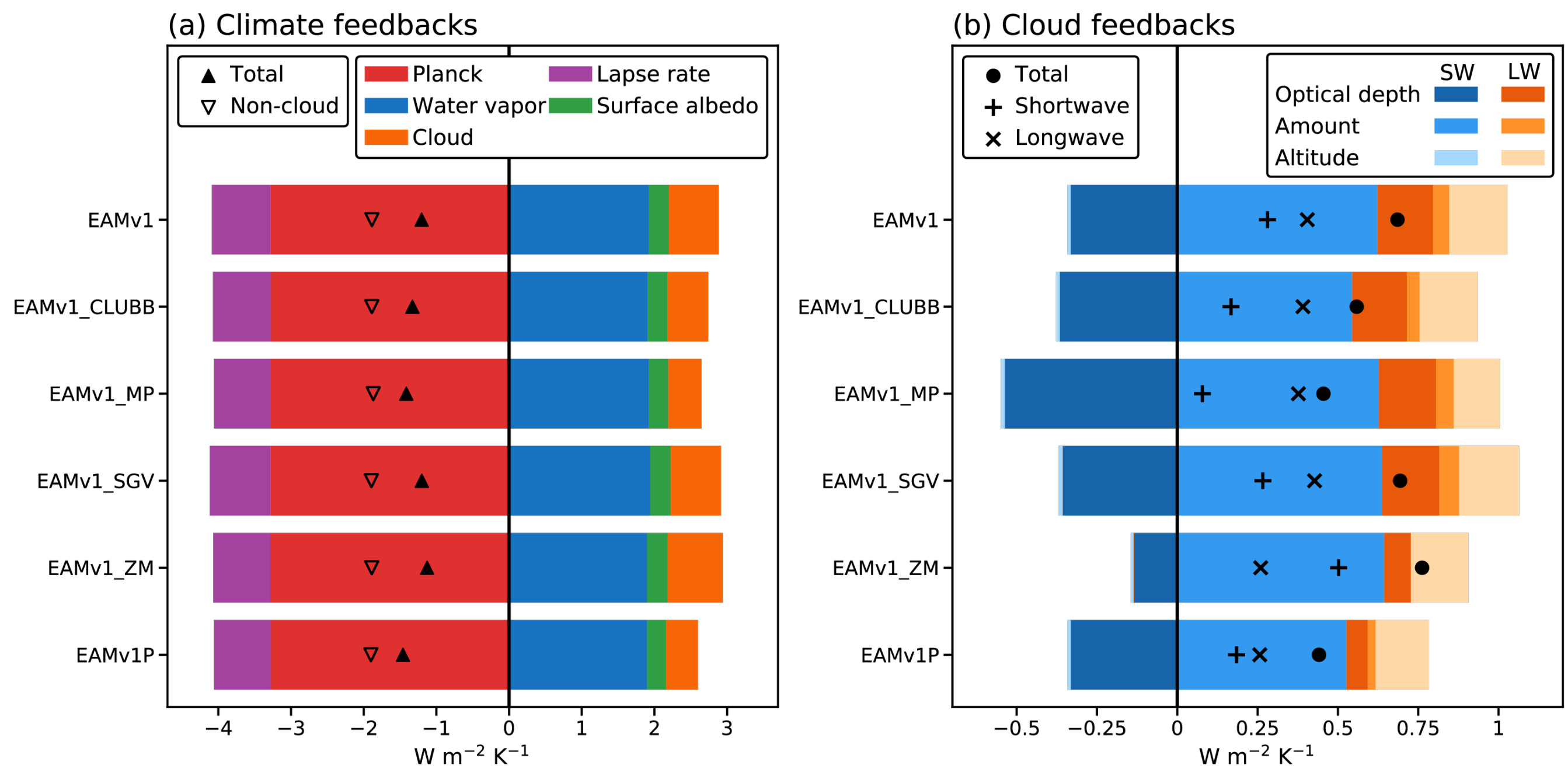

**Figure 18.** (a) climate feedbacks and (b) cloud feedbacks decomposed using radiative kernels (Zelinka et al., 2012;Pendergrass et al., 2018).

Figure 19 shows that parameter adjustments affect cloud feedbacks in different geographical regions. The total cloud feedback appears to be a balance between cloud optical depth feedback and cloud amount feedback, as the cloud altitude feedback is insensitive to our adjustments in parameters and subgrid effects.

In the tropics, the recalibrated model EAMv1P shows stronger positive total cloud feedback (Figure 19a), which can be attributed to the enhanced cloud optical depth feedback introduced by EAMv1_SGV and EAMv1_ZM. This highlights the importance of realistic representation of cloud properties associated with deep convection, including both the deep convective clouds as well as the anvil detrained from deep convection. In the subtropics, EAMv1P produces weaker positive total cloud feedback due to the reduction of cloud amount feedback in EAMv1_CLUBB and EAMv1_ZM. EAMv1_CLUBB weakens turbulent mixing and increases the skewness $Sk_w$ in the shallow Cu regions to facilitate asymmetric vertical mixing that enhances shallow Cu rather than the symmetric vertical mixing that enhances Sc. For this reason, a weaker positive cloud feedback is expected since Sc cloud amount decreases more with warming than shallow Cu (Cesana et al., 2019). EAMv1_ZM also reduces subtropical cloud amount feedback likely through its impacts on circulation which affects subtropical subsidence and clouds. In mid- and high latitudes, EAMv1_MP makes the largest contribution to modifying cloud feedbacks. Making the WBF process more efficient reduces supercooled liquid clouds in the mean state, which strengthens the negative cloud optical depth feedback through enhancing the negative cloud phase feedback (Tan et al., 2016). We note that the high latitude cloud optical depth feedback is highly uncertain. Sherwood et al. (2020) estimated the feedback to be near zero based on two studies, Ceppi et al. (2016) and Terai et al. (2016), which reported feedback estimates of similar magnitude but opposite signs. Hence, it remains unclear if the stronger negative cloud optical depth feedback in the Southern Ocean produced by EAMv1_MP and EAMv1P is closer to reality, but this essentially reduces the global total cloud feedback due to the sign reversal of the total cloud feedback in the Southern Ocean.

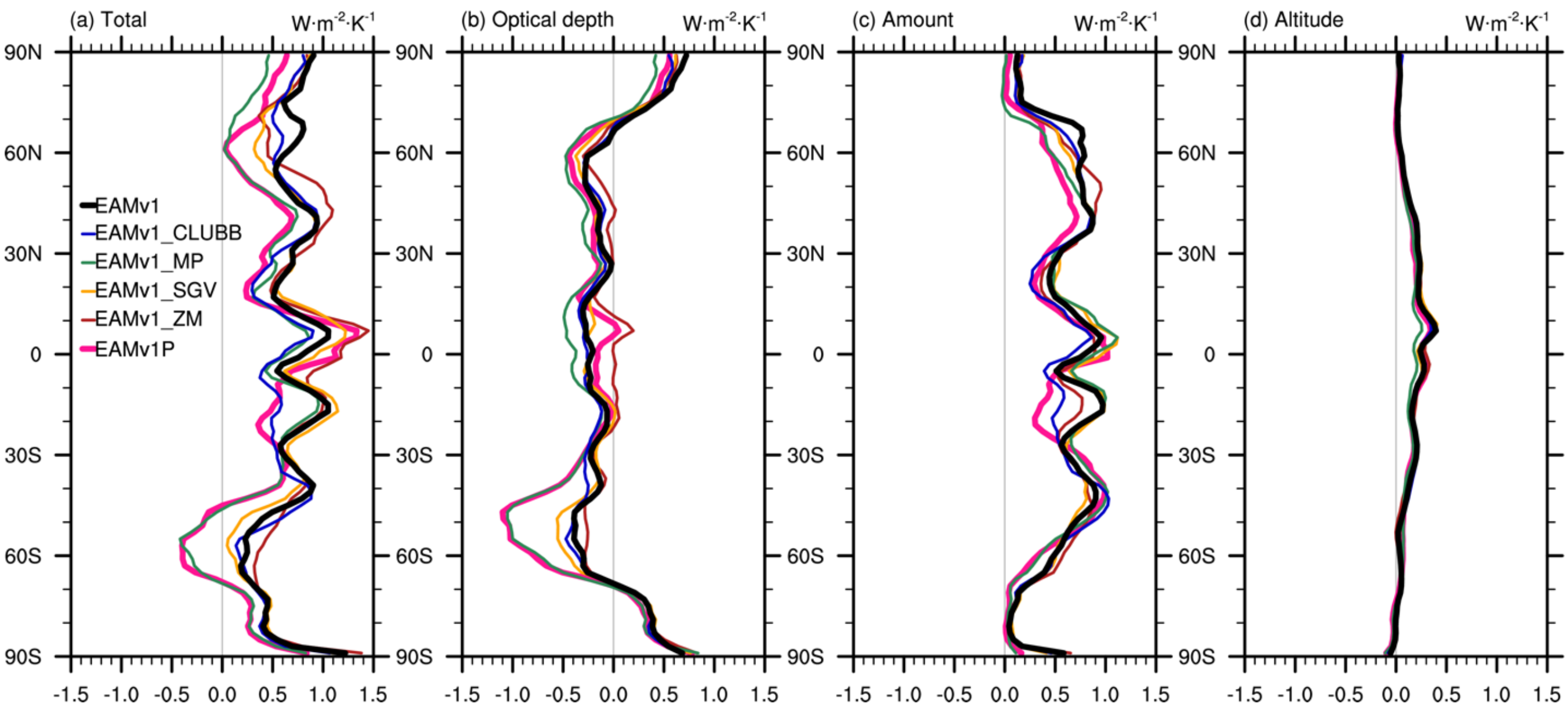

**Figure 19.** Zonal mean of (a) total cloud feedback; (b) cloud optical depth feedback; (c) cloud amount feedback; and (d) cloud altitude feedback.

In Table 11, we find that cloud fraction changes induced by surface warming are insensitive to the recalibration. LWP increases as the surface warms. By making the WBF process more efficient, EAMv1_MP shows a greater LWP response to surface warming, which weakens the positive cloud feedback as discussed previously. Liquid and ice particle numbers $N_c$ and $N_i$ are both reduced with surface warming, and parameter adjustments in EAMv1_MP and EAMv1_ZM affect the sensitivity. In terms of radiative properties, we find that the recalibration reverses the sign of the response of $\tau_{liq}$ to surface warming largely due to the changes made in EAMv1_MP, leading to cloud thickening instead of thinning in the lower troposphere (i.e., increasing $\tau_{low}$ as surface warms). In the upper troposphere, EAMv1_ZM reduces the $\tau_{hgh}$ sensitivity to surface warming, which weakens the positive high cloud feedback. The modifications in EAMv1_ZM have the largest impact on the changes in CRE response changes associated with ice clouds. Combining all the changes, the revised model EAMv1P reverses the sign of the liquid CREs, likely due to the cloud phase response to warming caused by increased IWP in the model.

**Table 11.** Same as Table 6, except that the change of cloud properties induced by surface warming relative to their pre-industrial values (unit = % $K^{-1}$) are shown. Variables are defined in Table 6.

| Variable | EAMv1 | EAMv1_CLUBB | EAMv1_MP | EAMv1_SGV | EAMv1_ZM | EAMv1P |
|---|---|---|---|---|---|---|
| $\Delta_R F_{cld,tot}$ | -0.61 | -0.55 | -0.52 | -0.62 | -0.70 | -0.56 |
| $\Delta_R F_{cld,low}$ | -1.37 | -1.36 | -1.26 | -1.41 | -1.38 | -1.17 |
| $\Delta_R F_{cld,med}$ | -2.81 | -2.79 | -2.40 | -2.76 | -3.04 | -2.78 |
| $\Delta_R F_{cld,hgh}$ | 0.36 | 0.32 | 0.45 | 0.31 | -0.03 | 0.11 |
| $\Delta_R LWP$ | 1.75 | 1.82 | 2.73 | 1.92 | 2.11 | 3.12 |
| $\Delta_R IWP$ | -3.65 | -3.52 | -3.79 | -3.74 | -4.19 | -3.74 |

| | | | | | | |
|---|---|---|---|---|---|---|
| $\Delta_R N_c$ | -2.09 | -2.02 | -1.39 | -1.84 | -1.69 | -0.64 |
| $\Delta_R N_i$ | -2.13 | -2.59 | -3.54 | -2.01 | -4.42 | -4.35 |
| $\Delta_R \tau_{cld}$ | 0.26 | 0.37 | 0.98 | 0.40 | 0.14 | 1.00 |
| $\Delta_R \tau_{liq}$ | -0.28 | -0.20 | 0.69 | 0.04 | 0.16 | 1.44 |
| $\Delta_R \tau_{ice}$ | -2.99 | -3.02 | -3.53 | -3.10 | -4.21 | -3.80 |
| $\Delta_R \tau_{snow}$ | -0.25 | -0.23 | -0.02 | -0.27 | -0.97 | -0.62 |
| $\Delta_R \tau_{conv}$ | 2.50 | 2.66 | 2.70 | 2.26 | 2.30 | 2.48 |
| $\Delta_R \tau_{low}$ | -0.54 | -0.42 | 0.23 | -0.39 | -0.30 | 0.68 |
| $\Delta_R \tau_{hgh}$ | 7.04 | 6.81 | 6.78 | 7.04 | 4.71 | 4.62 |
| $\Delta_R$SWCRE | -0.72 | -0.50 | -0.41 | -0.70 | -1.07 | -0.60 |
| $\Delta_R SWCRE_{liq}$ | -0.48 | -0.13 | 0.10 | -0.43 | -0.27 | 0.63 |
| $\Delta_R SWCRE_{ice}$ | -2.42 | -2.46 | -2.89 | -2.50 | -3.71 | -3.55 |
| $\Delta_R SWCRE_{snw}$ | -0.62 | -0.50 | -0.51 | -0.61 | -1.33 | -1.10 |
| $\Delta_R SWCRE_{conv}$ | -0.59 | -0.56 | -0.42 | -0.67 | -0.53 | -0.51 |
| $\Delta_R$LWCRE | -0.58 | -0.50 | -0.79 | -0.53 | -1.18 | -1.21 |
| $\Delta_R LWCRE_{liq}$ | -0.08 | 0.29 | -0.16 | 0.02 | 0.06 | 0.50 |
| $\Delta_R LWCRE_{ice}$ | -1.18 | -1.27 | -1.80 | -1.20 | -2.27 | -2.47 |
| $\Delta_R LWCRE_{snw}$ | 0.36 | 0.40 | 0.31 | 0.42 | -0.24 | -0.11 |
| $\Delta_R LWCRE_{conv}$ | -2.56 | -2.47 | -2.39 | -2.63 | -2.56 | -2.48 |

In assessing the impact of parameter changes to ECS, we also computed the lower tropospheric mixing index (LTMI) (Sherwood et al., 2014) and found that the recalibration leads to a 10% reduction in LTMI (not shown), which corresponds to about 1 K decrease in ECS based on the LTMI-ECS relationship from CMIP5. Most parameter adjustments do not alter LTMI. EAMv1_ZM produces lower LTMI because it reduces convective activity by weakening the convective autoconversion process to increase cirrus cloud opacity that stabilizes the troposphere. However, because the statistical significance of the relationship between LTMI and ECS has decreased in CMIP6 compared to CMIP5 (Schlund et al., 2020), LTMI might not be a good predictor for ECS in E3SM.

In Section 3.2 we show that the parameter changes in EAMv1_ZM significantly reduce the ratio of convective precipitation rate to total precipitation rate in the present-day climatology. This change can lead to different precipitation response to surface warming because different precipitation mechanisms are employed between the convection and the microphysics parameterizations. Figure 20 shows enhanced convective precipitation with warming in the tropics, the SPCZ, and storm tracks in EAMv1. EAMv1_ZM significantly reduces the response, likely due to the reduced convective autoconversion efficiency. Other parameter adjustments also affect the response in the Indo-Pacific warm pool, but the parameter changes do not have direct impact on convective precipitation so the change in response might be caused by circulation feedbacks. In the recalibrated model EAMv1P, the convective precipitation response to surface warming is reduced mostly in the tropics. The global mean convective precipitation response is reduced by 0.013 mm day$^{-1}$ K$^{-1}$ (-24%) compared to the response in EAMv1. The relative increase of convective precipitation due to surface warming, however, is only slightly reduced from 3.07% K$^{-1}$ in EAMv1 to 2.97% K$^{-1}$ in EAMv1P.

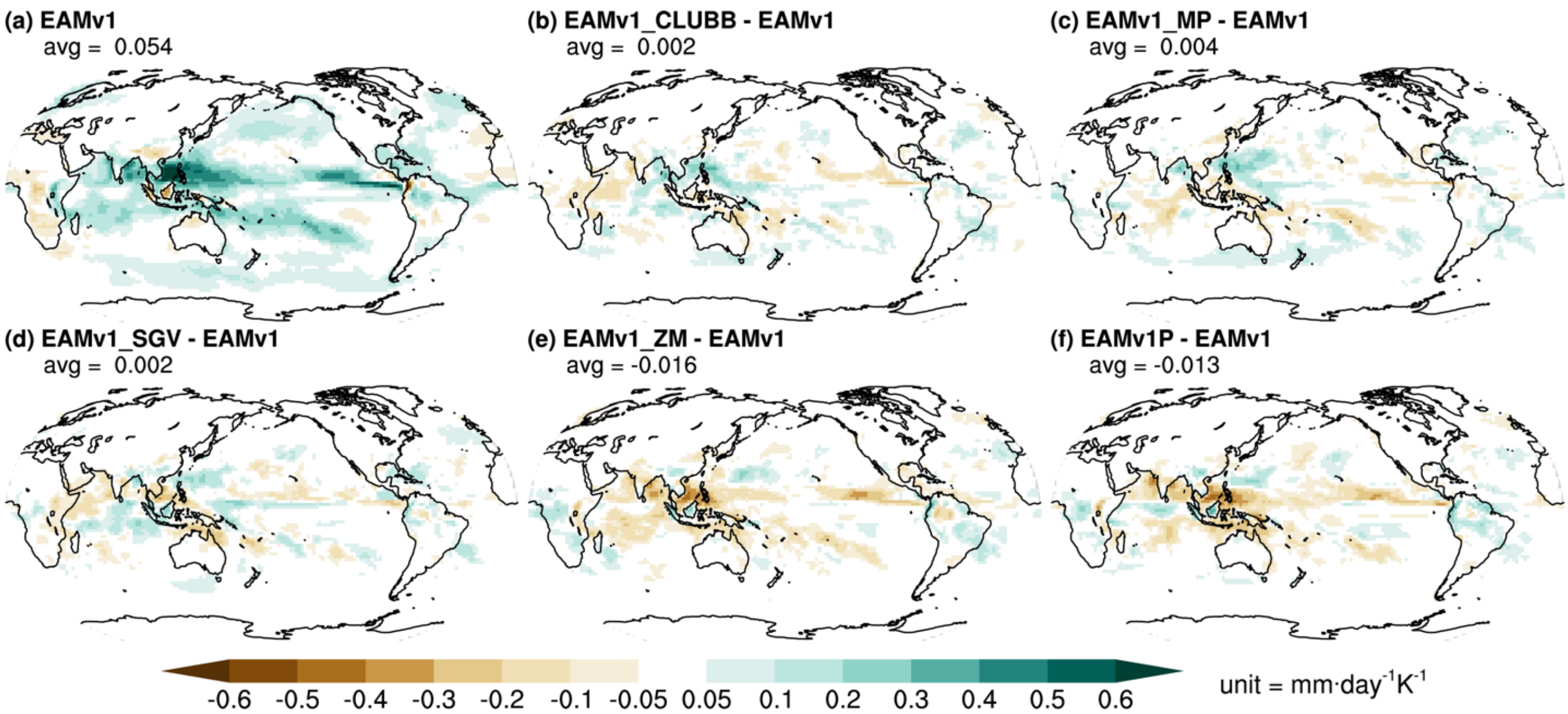

**Figure 20.** The change of convective precipitation rate induced by surface warming.

925

The large-scale precipitation response in EAMv1 has a similar magnitude as the convective precipitation response, but the response is larger in the storm tracks and not as strong in the tropics (Figure 21). EAMv1_ZM significantly enhances the response in the TWP because the parameter changes in EAMv1_ZM shift the precipitation from convective to large-scale so that the response comes from the large-scale precipitation. The recalibrated model EAMv1P enhances the large-scale precipitation response by 0.018 mm day$^{-1}$ K$^{-1}$ (+37%), compared to EAMv1. The relative increase of large-scale precipitation due to surface warming is also increased from 3. 17% K$^{-1}$ in EAMv1 to 4.11% K$^{-1}$ in EAMv1P.

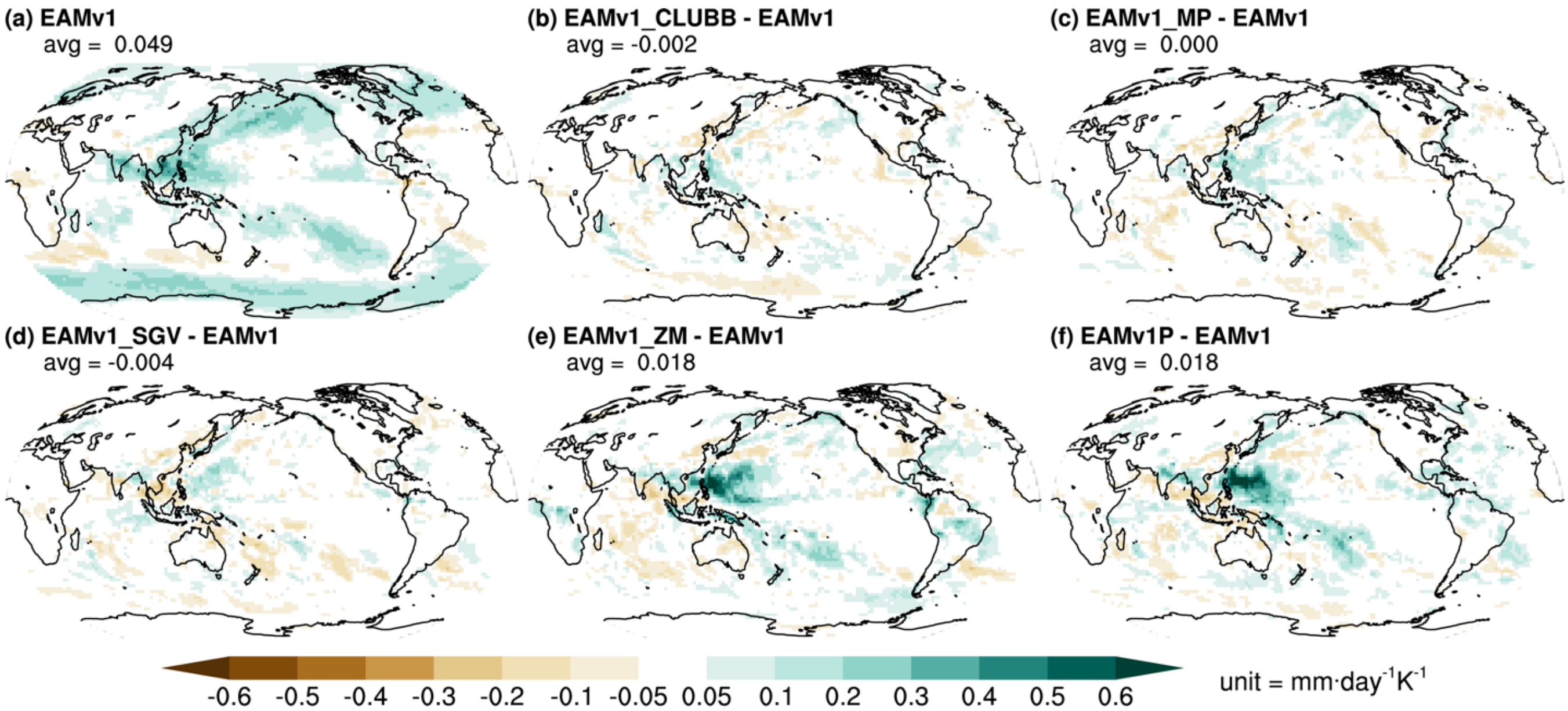

**Figure 21.** The change of large-scale precipitation rate induced by surface warming.

In summary, the recalibration enhances the negative climate feedback by reducing the positive cloud feedback. The storm track, shallow Cu regions, and the Indo-Pacific warm pool are the regions where the cloud feedback is most sensitive to the parameter adjustments. The largest precipitation response is seen in the tropics, SPCZ, and storm tracks. The parameter adjustments in the ZM deep convection parameterization produce the largest changes in the response. Because the default model EAMv1 and the recalibrated model EAMv1P produce different climate and cloud feedbacks, the two models are expected to produce different estimates of ECS, even though their $ERF_{ant}$ are about the same. Our results are consistent with the findings of Smith et al. (2020) that the statistical relationship between the $ERF_{aer}$ and ECS established in Kiehl (2007) and Forster et al. (2013) is challenged by the current generation ESMs. Fully coupled model simulations are needed to test this hypothesis.

## 4 Summary and Discussion

In this study, we have developed a new model configuration of EAMv1, namely the EAMv1P, using a model calibration strategy that focuses on calibrating CREs that can be reliably observed across cloud regimes and geographical regions. The recalibration was guided by our understanding of the physical mechanisms which relate biases to uncertain process assumptions. The recalibrated model produces improved present-day cloud and precipitation climatology and reduced sensitivity to aerosol perturbation and surface warming. Below we summarize the changes and behavior of the intermediate model configurations:

- Accounting for the subgrid effects (EAMv1_SGV) was intended to increase cloudiness where large-scale winds are weak and convection occurs frequently (e.g., TWP and Amazon) by enhancing local surface fluxes of heat, moisture, and momentum in those regions. Compared to other intermediate model configurations, EAMv1_SGV produces the largest impact in reducing the tropical surface wind direction bias, which will likely reduce the cold tongue bias in the fully coupled E3SM. Introducing the subgrid effects also reduces precipitation bias over TWP, Amazon, and high-elevation regions (e.g., Himalayas and Andes). EAMv1_SGV produces moderately weaker surface temperature response and weaker precipitation response to aerosol forcing compared to the default model EAMv1.

- Parameter adjustments in the ZM deep convection parameterization (EAMv1_ZM) were intended to improve tropical clouds by weakening the convective autoconversion and reducing detrained ice crystal radius. We find that these changes increase IWP globally. Furthermore, we find that EAMv1_ZM is the only model configuration that produces a stronger $ERF_{aci}$ and a stronger positive cloud feedback. The enhanced $ERF_{aci}$ is seen in East Asia, Europe, and Sc and shallow Cu regions. The increased cloud feedback is primarily due to the significant reduction of negative cloud optical depth feedback in the tropics.

- Parameter adjustments in the CLUBB parameterization (EAMv1_CLUBB) were introduced to improve the subtropical Sc, shallow Cu, and the Sc-to-Cu transition by making parameters a function of the skewness of subgrid vertical velocity $Sk_w$. We find that the changes also significantly reduce the precipitation bias over the central Pacific Ocean. The changes introduced in EAMv1_CLUBB do not affect $ERF_{aci}$, but they lead to the largest reduction of the positive cloud amount feedback in the subtropics, compared to other intermediate model configurations.

- Parameter adjustments in the MG2 parameterization (EAMv1_MP) were intended to 1) address the excessive supercooled cloud liquid in the mid- and high latitudes by enhancing the WBF process; 2) reduce ice particle number by reducing the sulfate aerosol available for homogeneous ice nucleation; and 3) improve Sc by enhancing the droplet sedimentation rate. We find that these changes give the largest reduction in $ERF_{aci}$ in the mid- and high latitudes, in areas under great anthropogenic influence (e.g., East Asia, North America), and in the subtropics. EAMv1_MP also produces the weakest total cloud feedback due to the stronger negative cloud optical depth feedback in the tropics, and mid- and high latitudes. The significant enhancement of negative cloud optical depth feedback results in a reversal of the sign of the total cloud feedback in the Southern Ocean.

The revised model EAMv1P includes all the changes discussed above. We find that EAMv1P produces a much more realistic CRE distribution than EAMv1 by addressing cloud biases in the tropics, subtropics, and mid- and high latitudes. This is achieved through modest adjustments in the ZM deep convection scheme and subgrid effects, CLUBB turbulence, and MG2

microphysics. The improved CRE distribution leads to better radiative energy distribution, which is essential for setting up a realistic atmospheric circulation that further improves the overall fidelity of the model atmospheric state. We have also compared results from grouped parameter changes to understand how process assumptions affect CRE as well as other aspects of the simulated atmosphere. We show that the recalibrated model produces more improvements than the sum of the improvements from individual intermediate configuration, demonstrating the nonlinearity in the climate system and the necessity of combining all of the improvements that target different biases in different regimes.

Cloud, precipitation, and surface temperature responses to anthropogenic aerosols and greenhouse gases are major sources of uncertainty in the simulated climate of the past, present, and future. Since the climate system is nonlinear, realistic estimates of the system's response depend on a realistic base state. EAMv1's deficiencies in base state fidelity likely contribute to its biases in the historical surface temperature evolution as well as its high ECS. In contrast, the recalibrated model EAMv1P produces a much more realistic present-day base climate state, due to a better calibration of cloud properties and subgrid effects that improve the representation of physical mechanisms compared to EAMv1. Hence, the revised model EAMv1P is more likely to produce credible estimates of the climate system's response to external forcings and climate projections when running as part of the fully coupled E3SM.

We show that the sensitivity of clouds, precipitation, and surface temperature to anthropogenic aerosols is significantly lower in the recalibrated model than in the default model, suggesting the potential to improve the historical surface temperature evolution over E3SMv1, such as the potential to reduce the cold bias between the 1960s and 1980s. We find that the responses to anthropogenic aerosols are mostly affected by parameter adjustments in EAMv1_MP and EAMv1_ZM. To simulate historical surface temperature evolution accurately, future model development efforts should target these two parameterizations so that processes of cloud microphysical and deep convective processes are better constrained to represent real-world processes.

The recalibrated model EAMv1P also produces smaller cloud feedback compared to the default model EAMv1, suggesting potential improvements to the surface temperature evolution, like slower warming after the 1980s and a lower ECS. Parameter adjustments in EAMv1_CLUBB, EAMv1_MP, and EAMv1_ZM significantly affect cloud feedbacks. Hence, to reduce the uncertainty in the predictions of future climate, subgrid cloud properties and process representations including turbulent mixing, cloud macro- and micro-physics, and deep convection need to be better constrained.

Lastly, EAMv1 and EAMv1P produce different surface temperature responses to anthropogenic aerosols and different cloud feedbacks (and, consequently, ECS) even though they produce the same global mean ERF. This suggests that the statistical relationships between the global mean ERF, cloud feedback, and ECS established in Kiehl (2007) and Forster et al. (2013) do not apply to current generation ESMs, as documented in Smith et al. (2020). This indicates that global mean ERF is

not a good indicator of the historical and future climate change. Other factors such as the spectral composition (i.e., shortwave vs. longwave) and spatial distribution of the ERF and cloud feedback, as well as the realism of the unperturbed base climate state need to be considered. Identifying the process representations that affect only ERF, those that affect only cloud feedback, and those that affect both is an important step toward better understanding of the evolution of the climate system.

*Code and data availability.* The E3SM model code and input data is available at [https://doi.org/10.11578/E3SM/dc.20180418.36]. The model simulation data used in this study is available at [https://portal.nersc.gov/archive/home/p/plma/www/eamv1_tunings]. The ERA5 data is obtained from Copernicus Climate Change Service Climate Data Store (CDS), accessed on December 27, 2019, at [https://cds.climate.copernicus.eu/cdsapp#!/home]. The CERES-EBAF cloud radiative effect data and the GPCP precipitation climatology data are obtained from the AMWG diagnostics package, available at [http://www.cesm.ucar.edu/working_groups/Atmosphere/amwg-diagnostics-package/]. The MODIS data is obtained from NASA GIOVANNI at [https://giovanni.gsfc.nasa.gov/giovanni/]. The MERRA-2 dataset is obtained from NASA Goddard Earth Sciences (GES) Data and Information Services Center (DISC) at [https://disc.gsfc.nasa.gov/]. The GOCCP data is obtained from [https://climserv.ipsl.polytechnique.fr/cfmip-obs/]. The E3SM diagnostics package used in this study was archived at [https://doi.org/10.5281/zenodo.5555094].

*Author contributions.* PM designed the study, performed simulations and analyses, and prepared the first draft of the manuscript. BEH, VEL, RN, AG, HM, HW, KZ, PAB, ZZ, HS, XL, JW, PMC, and PJR contributed to developing the tuning strategy and analysis. SAK, MDZ, and YZ performed the cloud feedback decomposition analysis and contributed to cloud evaluation. YQ, JY, CRJ, MH, ST, XZ, WL, JQ, HC, MAB, JM, SH, QT, and JF contributed to the evaluation and analysis of results. BS implemented the code in E3SMv1. XZ, LKB, and JDF contributed to assessment of subgrid effects. HW and MAT contributed to the testing of model sensitivity to time-stepping and process coupling. JG, SX, and LRL contributed to interpreting results and comparison with other modeling studies. All authors contributed to the writing of the manuscript.

*Competing interests.* Po-Lun Ma is a Topical Editor and Richard Neale is an Executive Editor and Topical Editor of Geoscientific Model Development. Other authors declare that they have no conflict of interest.

*Acknowledgements.* The model tuning was supported as part of the Energy Exascale Earth System Model (E3SM) project [https://dx.doi.org/10.11578/E3SM/dc.20180418.36], funded by the U.S. Department of Energy, Office of Science, Office of Biological and Environmental Research, Earth System Model Development (ESMD) program area. The analyses of effective radiative forcing and aerosol-cloud interactions were supported by the Enabling Aerosol-cloud interactions at GLobal convection-permitting scalES (EAGLES) project, funded by the U.S. Department of Energy, Office of Science, Office of Biological and Environmental Research, Earth System Model Development program area. The development and evaluation of gustiness effects over land were supported as part of the Integrated Cloud, Land-Surface, and Aerosol System Study (ICLASS)

science focus area, funded by the U.S. Department of Energy, Office of Science, Office of Biological and Environmental Research, Atmospheric System Research (ASR) program. The development of cloud and boundary layer diagnostics was funded by the U.S. Department of Energy, Office of Science, Office of Biological and Environmental Research, Regional and Global Model Analysis (RGMA) program area. This research used resources of the National Energy Research Scientific Computing Center (NERSC), a U.S. Department of Energy Office of Science User Facility operated under Contract No. DE-AC02-05CH11231. This research also used a high-performance computing cluster provided by the Office of Biological and Environmental Research Earth System Model Development program area and operated by the Laboratory Computing Resource Center at Argonne National Laboratory. The Pacific Northwest National Laboratory is operated for the U.S. Department of Energy by Battelle Memorial Institute under contract DE-AC05-76RL01830. Work at Lawrence Livermore National Laboratory was performed under the auspices of the U. S. DOE by Lawrence Livermore National Laboratory under contract No. DE-AC52-07NA27344. Sandia National Laboratories is a multi-mission laboratory managed and operated by National Technology & Engineering Solutions of Sandia, LLC, a wholly owned subsidiary of Honeywell International Inc., for the U.S. Department of Energy's National Nuclear Security Administration under contract DE-NA0003525. This paper describes objective technical results and analysis. Any subjective views or opinions that might be expressed in the paper do not necessarily represent the views of the U.S. Department of Energy or the United States Government.

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
