# Peer review of "Better calibration of cloud parameterizations and subgrid effects increases the fidelity of E3SM Atmosphere Model version 1"

_Geoscientific Model Development, 2021_

## Author Comment (AC1)

**Reply to Reviewer 1**

This summarizes (at some length) efforts to improve the calibration and evaluation of the atmospheric component of the E3SM coupled model. The procedure described seems extremely labor intensive and (frankly) somewhat arbitrary. Nonetheless, the results do show significant improvements and curiously, a reduction in the implied climate sensitivity (assessed via Cess-type perturbations). This is publishable with only minor revisions (as outlined below) and perhaps some condensing to reduce length and repetition.

Reply: We thank the reviewer for the comment. We have considered the reviewer's request to reduce length but respectfully decided to stand by our instinct that to be sufficiently comprehensive about this challenging topic it is important to show all of these diagnostics. We also note that there is precedent in GMD for such technical details for model development and documentation. This paper attempts to carefully document the approach, strategy, and results of model calibration, including the strengths and weaknesses of the approach and results. Furthermore, since most of the recalibration is part of the E3SMv2, this paper serves as a documentation of improvements to E3SMv1 that lead to the new model.

I have two questions that might add to some of the discussion. What is the prospect for automating some of these tests, using ML/AI for instance to reduce the burden and increase the area of phase space tested? I don't have huge confidence that the current procedure will lead to true (local) minima in errors, but I'd like to see this discussed here.

Reply: We have added a paragraph at the end of the paper to discuss the potential use of AI/ML in model calibration:

"It is natural to wonder if an equivalent or superior ESM calibration might have been achievable with less human effort or fewer computational resources via semi-automated machine learning (ML) methods that emulate or expand the workflow outlined in this paper. Indeed, emulating a complex model's parameter sensitivities following human constructed trial simulations to aid model calibration and uncertainty quantification would be an intriguing possibility. Several recent studies have shown successful application of ML methods in model calibration (Cleary et al., 2021;Dunbar et al., 2021;Couvreux et al., 2021;Hourdin et al., 2021). In theory, reinforcement learning (RL) with an appropriately formulated agent-based optimization system could be guided via its loss function formulation with skill metrics that optimize for the same patterns and mean state climate metrics that we prioritized in this study. In practice, however, this ML task faces a fundamental challenge that the cost of an individual agent-reward sample is performing multi-year climate simulations. The workflow outlined in this paper has the considerable advantage that experienced human experts make educated parameter interventions based on assessment of the simulation that discriminates desired effects in a nuanced way and tolerates certain unintended consequences. It is not clear how available ML methods could be infused with analogous physical foresight to make similar decisions, and thus logical to expect they would require more evaluation samples to succeed via brute force. Therefore, experimenting with clever strategies to increase reward density and to integrate physical knowledge from experts in the ML workflow would be a highly worthy long-term challenge."

Secondly, there is a preprint related to ECS in CESM2 (a related model), that has pointed out some odd (possibly erroneous) coding related to the ice-nucleation in that model (CAM6). Does this have any relevance here? https://www.essoar.org/pdfjs/10.1002/essoar.10507790.1

Reply: In E3SM, we do use the cloud ice number limiter (Nimax), but it is set to the in-cloud ice nucleation number. In Zhu et al (2021), they indicated that just removing nimax is not sufficient to lower ECS (and on its own it actually leads to unrealistic features). The key seems to be removing nimax as well as increasing the number of microphysics sub timesteps (which would add computational cost). Given these constraints, we decided not to pursue this but we agree that this could be part of a future sensitivity study.

Minor points:
line 40. "...precise knowledge of ... ERF is not enough". This is a strawman argument. Who has ever said that it was?

Reply: In this study, we find that even though the recalibration does not change the global mean ERFs, the sensitivity of clouds, precipitation, and surface temperature to aerosol perturbations is significantly reduced. This suggests that global mean ERFs are not enough to constrain historical or future climate change. We think this statement is consistent with our findings. We also added a paragraph describing the empirical relationship between ERF and ECS in CMIP models and why it is scientifically interesting to point out that the global mean ERFs are insufficient: "Furthermore, an empirical relation has been shown to exist between the global mean ERFant and ECS in climate models from both the CMIP3 and CMIP5 collections (Kiehl, 2007;Forster et al., 2013). The relationship between ERFant and ECS exists because both values in models are sensitive to simulated clouds. Our tuning strategy specifically targets improving the representation of clouds, and it is worth asking whether these improvements uphold or alter the ERFant -ECS relation. The small difference in ERFant between the EAMv1 and EAMv1P configurations suggests the possibility of a similar small difference in ECS between these two configurations, and yet we find this is not the case."

line 84. The comparison to other ESMs is irrelevant. It the comparison to the constrained range from observations that matters (Sherwood et al, 2020).

Reply: We have replaced "compared to other ESMs" with "compared to estimates based on multiple lines of evidence including process understanding, historical climate record, and paleoclimate record (Sherwood et al., 2020)".

line 114-115. Is there any evidence that the skill scores dervied from a 5 day simulation are correlated to skill scores from a year or 10 year run? Presumably they are not being tested against the same observations?

Reply: We agree with the reviewer that calibration based on short simulations might reach a different configuration than that based on long simulations, because short simulations are intended for understanding the fast physics (e.g., clouds, convections, turbulence) and their local effects, while multi-year simulations include the large-scale feedback. In line 121-124, we stated that "Another limitation is that the short simulations focus on fast physical processes and rapid adjustments. By design, important factors such as slow internal variability of the atmosphere and circulation feedbacks are not considered, so any conclusion drawn from the short simulation ensemble might not be applicable to the calibration of the ESM for climate simulations."

line 120: "in hindsight"? is this referring to the 5-day simulations with EAMv1, or the previous one-at-a-time approach.

Reply: This refers to the 5-day simulations. In Line 120, we stated that "In hindsight, the parameter set selected for the short simulation ensemble during the EAMv1 development was insufficient because parameters not included in the original ensemble were later found to be important."

line 125. There is a big gap between 5 days and 10 years. Is there any assessment of how useful different lengths of simulation might be? For instance 1 year might be a good compromise?

Reply: The length of the simulations depends on the scales that model developers intend to consider in the model calibration. Short simulations with simulation length of a few days are useful for understanding fast physics and adjustments (Ma et al., 2021;Ma et al., 2014;Xie et al., 2012). In this study, we intended to

account for slow internal variability (e.g., inter-annual variability), so multi-year simulations are needed. In the development stage of EAMv1, both 5-yr and 10-yr simulations were performed.

line 128. use the actual times (10 years and 5 days) rather than 'short' or 'long' - relative measures are not very specific.

Reply: Both Wan et al. (2014) and Qian et al. (2018) use "short ensemble" to describe their methods because short simulations are not necessarily 5 days long. Therefore, we believe "short ensemble" is the appropriate term here. For clarification, we revised the text: "The one-at-a-time calibration approach using multi-year simulations and the short simulation ensemble approach using multi-day simulations…"

line 145. "perfect" is too much to ask. But the point about non-uniqueness is important.

Reply: We agree that we do not intend to achieve a perfect model configuration in this study. We stated that "out calibration does not lead to a unique and perfect configuration".

line 160. Why? The authors just spent two pages saying why this was not a good approach!

Reply: We compared the two approaches and discussed their strengths and weaknesses. The purpose of this study is to improve the model fidelity through physics guided tuning and to understand the impacts of parameter changes on the simulated clouds, precipitation, and climate. Therefore, the one-at-a-time approach is appropriate for this study.

line 224. "might"? --> "will"

Reply: We changed the text to "It is logical to expect that increasing model spatial resolution will reduce the impacts of these subgrid effects. Thus, a retuning of these subgrid effects would likely be needed when the model is run at a different horizontal resolution."

line 245. Has the length of these simulations been mentioned?

Reply: The design of the simulations, including the length, is described in Section 2.4.

line 385. How long are these simulations? (line 400 suggests 11 years, but is that just for simulation #5?) In any case, move this up in the text.

Reply: Done.

line 420/Table 6. Add observed values (where available) for comparison (i.e. from CERES, or CALIPSO).

Reply: We have added the satellite observations in Table 6 and added the text "Satellite observations summarized in Stubenrauch et al. (2013) and Neubauer et al. (2019) are also provided but we note that it is dangerous, and can be misleading, to compare model state variables with satellite retrievals without using a simulator since large retrieval and sampling uncertainties exist."

line 438-449. please compare with Cesana et al (2021, doi:10.1029/2021GL094876). The implementation of a CALIPSO simulator should indeed be a high priority. Without a realistic target for LCF this tuning will inevitably be haphazard, but I think it likely that the EAMv1P is more realistic.

Reply: We agree that the implementation of the CALIPSO simulation with cloud phase diagnostics will be very important. Unfortunately, EAMv1 does not have such capability. We have already mentioned this

issue in the manuscript: "While the CMIP5 models tend to freeze liquid condensates at higher temperatures (Cesana et al., 2015;Tan et al., 2016;McCoy et al., 2016), EAMv1 appears to have overcorrected this bias and produced excessive supercooled liquid at low temperatures. Consistent with Zhang et al. (2019), EAMv1_MP increases the T5050. Combining with changes introduced in EAMv1_ZM, EAMv1P produces a much more reasonable T5050 of 254K, which is at the lower bound of the observational estimates. We note that even though Hu et al. (2010) provided an observationally derived LCF-T relationship based on the Cloud-Aerosol Lidar and Infrared Pathfinder Satellite Observation (CALIPSO) measurements (Winker et al., 2007), EAMv1 does not have the CALIPSO cloud phase simulator (Cesana and Chepfer, 2013) so that a fair comparison is not possible. Evaluating the model LCF-T relationship against satellite observations in a consistent way will be very useful and requires further investigation."

Cesana et al. (2021) pointed out the need for accounting for snow in the radiation calculation. Since E3SM already account for snow radiative effects, we do not think this is relevant to our study.

figure 4. The authors should add the EAMv1P-CERES map as well, so that the improemved version can be compared directly with panel a. (also in Figures 5, 6, 9, 10, 11, and 12)

Reply: A primary goal of this study is to assess the impacts of parameter changes on the simulated climate. Comparing different model configurations and the observations does not achieve that goal. Therefore, we believe that showing the differences between the default and the recalibrated model is the best way to present the results.

line 497. It is of course challenging, but I don't think the biggest challenge is the lack of observational data.

Reply: We have removed the sentence.

line 635. This will always be true - it is not a binary situation.

Reply: We changed the text to "While the possibility of compensating biases always exists, our confidence in the underlying physics in the model will be increased if many other aspects are also improved. Otherwise, we are forced to suspect that the model achieves its behavior primarily through compensating biases."

line 802. This is an odd argument. Who has ever claimed that ERF is sufficient to determine responses? It is precisley the opposite - the major uncertainty (since the Charney report!) has always been in the sensitivity.

Reply: In this paragraph, we summarize our findings by stating that the default model EAMv1 and the recalibrated model EAMv1P produce the same global mean ERFs but their responses of hydrological and surface temperature to aerosols are different. Hence, we find that global mean ERFs are insufficient to understand the response of hydrological cycle and surface temperature to aerosols. The shortwave and longwave contribution to the total aerosol ERF as well as the spatial distribution of aerosol ERF need to be considered. We think this statement is consistent with our findings. We also added a paragraph describing the empirical relationship between ERF and ECS in CMIP models and why it is scientifically interesting to point out that the global mean ERFs are insufficient: "Furthermore, an empirical relation has been shown to exist between the global mean ERFant and ECS in climate models from both the CMIP3 and CMIP5 collections (Kiehl, 2007;Forster et al., 2013). The relationship between ERFant and ECS exists because both values in models are sensitive to simulated clouds. Our tuning strategy specifically targets improving the representation of clouds, and it is worth asking whether these improvements uphold or alter the ERFant -ECS relation. The small difference in ERFant between the EAMv1 and EAMv1P configurations suggests the possibility of a similar small difference in ECS between these two configurations, and yet we find this is not the case."

line 818+. The comparison to the other models is fine, but the comparison should be with observationally constrained estimates - ie. Sherwood et al (2020), IPCC AR6 Chp. 7 etc.

Reply: We have added Sherwood et al (2020). We did not cite IPCC AR6 because the webpage stated that the documents carry the note from the Final Government Distribution "Do Not Cite, Quote or Distribute".

line 1019. This has never been claimed.

Reply: In this paragraph, we stated that even though the default model EAMv1 and the recalibrated model EAMv1P have the same global mean ERF, they produce different surface temperature response to aerosols and different cloud feedback. Hence, global mean ERF is not a good indicator for historical or future climate change. We think this statement is correct and consistent with our findings. We also added a paragraph describing the empirical relationship between ERF and ECS in CMIP models and why it is scientifically interesting to point out that the global mean ERFs are insufficient: "Furthermore, an empirical relation has been shown to exist between the global mean ERFant and ECS in climate models from both the CMIP3 and CMIP5 collections (Kiehl, 2007;Forster et al., 2013). The relationship between ERFant and ECS exists because both values in models are sensitive to simulated clouds. Our tuning strategy specifically targets improving the representation of clouds, and it is worth asking whether these improvements uphold or alter the ERFant -ECS relation. The small difference in ERFant between the EAMv1 and EAMv1P configurations suggests the possibility of a similar small difference in ECS between these two configurations, and yet we find this is not the case."

**Reference**

[revised manuscript text omitted]

---

## Author Comment (AC2)

**Reply to Reviewer 2**

This manuscript describes the process of retuning of version 1 of the atmospheric component EAM of E3SM climate model, which focused on parameters related to various cloud processes, and how the retuning impacted a range of quantities beyond the tuning targets, ranging from surface temperature in the present-day to aerosol forcing (via cloud adjustments) to cloud feedbacks.

General comments

The manuscript does a good job of explaining the reasoning that led to the strategy used in retuning, which relies on the bet that improving the representation of clouds will lead to improvements across the board. It's nice to see that the bet pays off.

Reply: We thank the reviewer for the positive comment.

There's rather a lot of detail in section 2, describing groups of parameters were tuned. There's really a lot of detail in section 3, which explains how the returned model behaves with respect to a wide range of emergent phenomena. There is so much detail, in fact, that the manuscript works much better as a description of what was done than it does as an explanation of what was learned. If the authors' goal is to document the strategy and its impacts they have succeeded, but if they aim to influence the ways in which readers undertake or understand model tuning they would be well advised to bring their ideas into sharper focus. Sharpening the manuscript will almost certainly involve relegating material to appendices or supplemental material.

Reply: We have considered the reviewer's request to reduce length but respectfully decided to stand by our instinct that to be sufficiently comprehensive about this challenging topic it is important to show all of these diagnostics. We also note that there is precedent in GMD for such technical details for model development and documentation. This paper attempts to carefully document the approach, strategy, and results of model calibration, including the strengths and weaknesses of the approach and results. Furthermore, since most of the recalibration is part of the E3SMv2, this paper serves as a documentation of improvements to E3SMv1 that lead to the new model.

A large proportion of the very many figures are of the form of Figure 4: six (well-constructed) maps showing the difference of v1 against observations, four maps showing the change induced by the four sets of parameter changes, then a map showing the aggregate change of the final retuning relative to the original. Readers are left to judge improvement by mentally subtracting the bias in the upper left plot from the change in the lower right plot. Could the lower plot be revised to show, for example, the improvement or degradation in the original bias as a result of the tuning?

Reply: A primary goal of this study is to assess the impacts of parameter changes on the simulated climate. We have considered the reviewer's request and experimented the suggested way to present results, but we found that this does not show the impacts of the parameter changes clearly. Therefore, we believe that showing the differences between the default and the recalibrated model is the best way to present the results.

Versions of the six-ma figure (e.g. Fig 7) with no observational constraint are harder for readers to assess.

Reply: Indeed there is no observational constraint for the global climatology of PBL decoupling strength (Fig 7) and cloud-top entrainment efficiency (Fig 8). However, they provide physical insights into the PBL characteristics and low clouds. By comparing them between different model configurations, we assess how the parameter changes affect PBL and low-level clouds, which is a primary goal of this study.

Tuning relies on the ability to measure improvements in simulations, normally relative to observations. It's remarkable that the authors spend essentially no time discussing the sources of their observational constraints, or how uncertainty in these constraint is or isn't considered as part of the tuning strategy.

Reply: We acknowledge that the uncertainty associated with observational datasets varies. We have tried to minimize the impact of observational uncertainties by using cloud simulators (Fig 5) and by accounting for the satellite sampling strategy (Fig 12). We have also added observational constraints, where available, in Table 6, and the text "Satellite observations summarized in Stubenrauch et al. (2013) and Neubauer et al. (2019) are also provided but we note that it is dangerous, and can be misleading, to compare model state variables with satellite retrievals without using a simulator since large retrieval and sampling uncertainties exist.".

The tuning strategy used by the authors is somewhat traditional. Comparisons to other approaches (e.g. the automated calibration to process-scale constraints used by HiTune, doi:10.1029/2020MS002217 or the formal inference discussed in the Clima project, doi:10.1016/j.jcp.2020.109716) would no doubt be welcome.

Reply: We have added a paragraph to discuss the potential of using automated machine learning approaches for model calibration in the end of the paper:

"It is natural to wonder if an equivalent or superior ESM calibration might have been achievable with less human effort or fewer computational resources via semi-automated machine learning (ML) methods that emulate or expand the workflow outlined in this paper. Indeed, emulating a complex model's parameter sensitivities following human constructed trial simulations to aid model calibration and uncertainty quantification would be an intriguing possibility. Several recent studies have shown successful application of ML methods in model calibration (Cleary et al., 2021;Dunbar et al., 2021;Couvreux et al., 2021;Hourdin et al., 2021). In theory, reinforcement learning (RL) with an appropriately formulated agent-based optimization system could be guided via its loss function formulation with skill metrics that optimize for the same patterns and mean state climate metrics that we prioritized in this study. In practice, however, this ML task faces a fundamental challenge that the cost of an individual agent-reward sample is performing multi-year climate simulations. The workflow outlined in this paper has the considerable advantage that experienced human experts make educated parameter interventions based on assessment of the simulation that discriminates desired effects in a nuanced way and tolerates certain unintended consequences. It is not clear how available ML methods could be infused with analogous physical foresight to make similar decisions, and thus logical to expect they would require more evaluation samples to succeed via brute force. Therefore, experimenting with clever strategies to increase reward density and to integrate physical knowledge from experts in the ML workflow would be a highly worthy long-term challenge."

More specific comments

Tuning of course involves the changing of specific parameters. The variable names in the specific computer code are perhaps too specific to be in the main text. This information, and indeed probably the original and changed values, could be summarized in one or more tables in an appendix.

Reply: A primary purpose of this paper is to document the approach, strategy, and results of model calibration, including the strengths and weaknesses of the approach and results. Furthermore, since most of the recalibration is part of the E3SMv2, this paper serves as a documentation of improvements to E3SMv1 that lead to the new model. Therefore, we place the tables in Section 2.

Line 113: the current term of art is "perturbed parameter ensemble".

Reply: We have changed "perturbed physics ensemble" to "perturbed parameter ensemble".

The subsections of section 2 are labeled as tropical clouds, low clouds, etc. In practice each section might also be categorized according the scheme whose parameters are being tuned. Indicating this (e.g. "Tropical clouds and the deep convection scheme") might guide readers' attention.

Reply: We label the subsections by the cloud regime/type because there are multiple parameterizations that affect one cloud regime/type. While the reviewer's comment is well-received, we choose not to introduce another layer in the sub-sections but we have revised the text to better introduce the recalibration process.

Section 2 describes the re-tuning in detail, including which parameters are re-tuned and why. General material (e.g. line 286-294) should be deferred or removed.

Reply: Line 286-294 describes how the skewness in CLUBB is formulated and re-tuned. We think this is a very important piece of information because retuning the skewness significantly reduces the outstanding bias of thin marine stratocumulus and overly bright shallow cumulus, as described in Section 2.2.

Line 220: The authors scale the temperature variance provided by one scheme by a factor of 2 before introducing it in another scheme. It's not clear whether this is a reasonable physical assumption. It if is the choice should be justified; if it's not the choice should be explained.

Reply: We have added the sentence to provide an explanation: "Based on sensitivity tests, a scaling factor of 2.0 was introduced to enhance the effect so that the simulated tropical clouds are in better agreement with observations (as discussed in Section 3)"

Section 2.2: how are the different cloud regimes identified in practice, during a simulation?

Reply: We used the vertical velocity at 500hPa and the lower tropospheric stability (Medeiros and Stevens, 2011). The geographical distribution is also very useful. We have added the text in Section 2.2.

One reason for showing six panels in Figure 4 and its many analogs is to highlight the geographic distribution of the impacts of parameter changes. The authors might ask themselves if maps are the best way to show these differences in all cases.

Reply: We do want to reveal the impacts of parameter changes on the simulated climate around the world, including both the positive and negative impacts. Therefore, we chose to show the global maps.

Line 395-400 could be edited for clarity and to remove general material, as could lines 486-490.

Reply: Because the readers might not be familiar with the approach, we think it is necessary to provide the description of the experiments.

Line 528: EIS is thought to control low cloud properties, not their feedback (sensitivity to surface temperature change).

Reply: We have changed the text to "EIS has traditionally been considered as an important cloud controlling factor affecting low clouds and low cloud feedback (Klein et al., 2017;Myers et al., 2021)." For instance, Myers et al. (2021) state that "The strongest individual components of the [low-cloud] feedback in almost all regions globally are those due to SST and EIS (Supplementary Fig. 6), owing to the large forced responses of these cloud-controlling factors".

The central point of lines 715-728 could no doubt be made more compactly and directly.

Because the readers might not be familiar with the approach, we think it is necessary to provide the description of how the ERFs are computed from the model.

Line 739-740 are bewildering.

Reply: We have changed the text to "Both longwave and shortwave radiation affect surface temperature and atmospheric cooling rates, which govern the hydrological cycle."

Figure 13 is hard for readers to interpret. Coding bias with shapes and variables with numbers is really quite unfriendly - bias would be better coded with size or shading, leaving shape to stand for quantities. But readers will also appreciate guidance in interpretation, since all the panels look very much the same to an unpracticed reader.

Reply: The Taylor diagram (Taylor, 2001) summarizes multiple aspects of climate simulations in a single diagram. It has long been widely used in atmospheric science, and its format is familiar and fairly standard. We have added the reference and also the text "While EAMv1_CLUBB and EAMv1_MP do not produce different results from EAMv1, we find that the meridional wind at 850 and 500 hPa (coded as number 4 and 7) in EAMv1_SGV and EAMv1_ZM are in better agreement with ERA-5 the normalized standard deviation reduces."

Line 831: Does the present analysis use the specific kernels of Zelinka 2012 and Pendergrass 2018? If so this should be made explicit. If the text refers to the ideas the original papers (e.g. doi:10.1175/2007JCLI2044.1) should be referenced.

Reply: We used the kernels of Zelinka et al. (2012a, b), Zelinka et al. (2013), and Pendergrass et al. (2018) for our analysis. We have revised the text to cite the papers: "In Figure 18, climate feedbacks diagnosed using the Pendergrass et al. (2018) radiative kernel reveal that the non-cloud feedbacks are invariant across different model configurations and that the variation in total climate feedback is due solely to the spread in cloud feedbacks as a result of our parameter and subgrid adjustments. Further decomposing the cloud feedback into its total, shortwave, and longwave components via cloud radiative kernels (Zelinka et al., 2012a, b;Zelinka et al., 2013) indicates that cloud feedbacks are weakened from 0.77, 0.35, and 0.42 W m$^{-2}$ K$^{-1}$ in EAMv1 to 0.47 (-39%), 0.20 (-43%), and 0.27 W m$^{-2}$ K$^{-1}$ (-35%) in EAMv1P."

Figure 18, especially panel a, is not particularly informative, since readers are asked to compare small changes in large numbers introduced by tuning.

Reply: Fig 18(a) shows that "the non-cloud feedbacks are invariant across different model configurations and that the variation in total climate feedback is due solely to the spread in cloud feedbacks as a result of our parameter and subgrid adjustments." Fig 18(b) then shows the decomposition of cloud feedback to identify which specific cloud feedbacks (i.e., shortwave or longwave; amount, optical depth, or altitude) are affected by the parameter changes. We find this analysis very insightful.

**Reference**

Cleary, E., Garbuno-Inigo, A., Lan, S. W., Schneider, T., and Stuart, A. M.: Calibrate, emulate, sample, J Comput Phys, 424, ARTN 109716, 10.1016/j.jcp.2020.109716, 2021.

Couvreux, F., Hourdin, F., Williamson, D., Roehrig, R., Volodina, V., Villefranque, N., Rio, C., Audouin, O., Salter, J., Bazile, E., Brient, F., Favot, F., Honnert, R., Lefebvre, M. P., Madeleine, J. B., Rodier, Q., and Xu, W. Z.: Process-Based Climate Model Development Harnessing Machine Learning: I. A Calibration Tool for Parameterization Improvement, J Adv Model Earth Sy, 13, ARTN e2020MS002217, 10.1029/2020MS002217, 2021.

Dunbar, O. R. A., Garbuno-Inigo, A., Schneider, T., and Stuart, A. M.: Calibration and Uncertainty Quantification of Convective Parameters in an Idealized GCM, J Adv Model Earth Sy, 13, ARTN e2020MS002454, 10.1029/2020MS002454, 2021.

Hourdin, F., Williamson, D., Rio, C., Couvreux, F., Roehrig, R., Villefranque, N., Musat, I., Fairhead, L., Diallo, F. B., and Volodina, V.: Process-Based Climate Model Development Harnessing Machine Learning: II. Model Calibration From Single Column to Global, J Adv Model Earth Sy, 13, ARTN e2020MS002225, 10.1029/2020MS002225, 2021.

Klein, S. A., Hall, A., Norris, J. R., and Pincus, R.: Low-Cloud Feedbacks from Cloud-Controlling Factors: A Review, Surv Geophys, 38, 1307-1329, 10.1007/s10712-017-9433-3, 2017.

Medeiros, B., and Stevens, B.: Revealing differences in GCM representations of low clouds, Clim Dynam, 36, 385-399, 10.1007/s00382-009-0694-5, 2011.

Myers, T. A., Scott, R. C., Zelinka, M. D., Klein, S. A., Norris, J. R., and Caldwell, P. M.: Observational constraints on low cloud feedback reduce uncertainty of climate sensitivity, Nature Climate Change, 11, 501-+, 10.1038/s41558-021-01039-0, 2021.

Neubauer, D., Ferrachat, S., Siegenthaler-Le Drian, C., Stier, P., Partridge, D. G., Tegen, I., Bey, I., Stanelle, T., Kokkola, H., and Lohmann, U.: The global aerosol-climate model ECHAM6.3-HAM2.3-Part 2: Cloud evaluation, aerosol radiative forcing, and climate sensitivity, Geosci Model Dev, 12, 3609-3639, 10.5194/gmd-12-3609-2019, 2019.

Pendergrass, A. G., Conley, A., and Vitt, F. M.: Surface and top-of-atmosphere radiative feedback kernels for CESM-CAM5, Earth Syst Sci Data, 10, 317-324, 10.5194/essd-10-317-2018, 2018.

Stubenrauch, C. J., Rossow, W. B., Kinne, S., Ackerman, S., Cesana, G., Chepfer, H., Di Girolamo, L., Getzewich, B., Guignard, A., Heidinger, A., Maddux, B. C., Menzel, W. P., Minnis, P., Pearl, C., Platnick, S., Poulsen, C., Riedi, J., Sun-Mack, S., Walther, A., Winker, D., Zeng, S., and Zhao, G.: Assessment of Global Cloud Datasets from Satellites: Project and Database Initiated by the GEWEX Radiation Panel, B Am Meteorol Soc, 94, 1031-1049, 10.1175/Bams-D-12-00117.1, 2013.

Taylor, K. E.: Summarizing multiple aspects of model performance in a single diagram., J Geophys Res-Atmos, 106, 7183-7192, Doi 10.1029/2000jd900719, 2001.

Zelinka, M. D., Klein, S. A., and Hartmann, D. L.: Computing and Partitioning Cloud Feedbacks Using Cloud Property Histograms. Part I: Cloud Radiative Kernels, J Climate, 25, 3715-3735, 10.1175/Jcli-D-11-00248.1, 2012a.

Zelinka, M. D., Klein, S. A., and Hartmann, D. L.: Computing and Partitioning Cloud Feedbacks Using Cloud Property Histograms. Part II: Attribution to Changes in Cloud Amount, Altitude, and Optical Depth, J Climate, 25, 3736-3754, 10.1175/Jcli-D-11-00249.1, 2012b.

Zelinka, M. D., Klein, S. A., Taylor, K. E., Andrews, T., Webb, M. J., Gregory, J. M., and Forster, P. M.: Contributions of Different Cloud Types to Feedbacks and Rapid Adjustments in CMIP5, J Climate, 26, 5007-5027, 10.1175/Jcli-D-12-00555.1, 2013.

---

## Author Comment (AC3)

**Reply to Reviewer 3**

The manuscript presents comprehensive results from the model calibration effort to improve DOE EAM_v1. It explores the impacts of the model recalibration on the fidelity of model climate, and implications for cloud feedbacks and ERF. The documented technical details in parameter calibration are highly beneficial for the scientific community to understand the model at the process level. The calibration comprises four major parts that reflect the latest advancement in modeling cloud, convection, aerosol, and radiation. The manuscript is co-authored by leading scientists in the field. Overall, the model development processes and associated impacts on major science questions are thoroughly discussed. The work is appropriate for GMD and I recommend acceptance after the authors address the comments listed below.

Reply: We thank the reviewer for the positive comment.

1) As each experiment runs for 11 years, some more statistical analyses can be conducted. For example, for Fig. 3, are the differences significant compared with the natural variability in the control run? For those difference maps, the authors may consider wiping out those pixels with insignificant differences.

Reply: We have added the statistical significance test in Fig 4, 5, 6, 7, 8, 9, 10, 12, 15, 17, 20, and 21.

2) L139-141, the logic is unclear here. Why the smaller cloud feedback and aerosol radiative forcing can lead to a better surface temperature simulation in a couple run?

Reply: We have revised the text to clarify this point:

"…, the recalibrated atmosphere model, denoted as EAMv1P, shows lower cloud and precipitation sensitivity to aerosol perturbation and to surface warming. Because the notable biases in E3SMv1's simulated surface temperature evolution are due to a combination of high ECS (from cloud feedback) and strong aerosol forcing (Golaz et al., 2019), EAMv1P may lead to improvements to the simulation of the 20th century temperature evolution and a lower estimate of ECS when running as part of the fully coupled E3SM."

3) L343, should be "reduced conversion".

Reply: We have changed it to "insufficient conversion".

4) Fig. 4c&5c, tuning in MP apparently impacts LWP, but not cloud fraction in the Sc regions. Is it simply due to the diagnostic cloud fraction scheme which is independent with cloud microphysics? If yes, should such a disconnection be targeted in the future model development?

Reply: Indeed generally speaking cloud fraction and liquid water path are computed by different parts of an ESM, which can introduce inconsistencies. In E3SMv1, CLUBB is the macrophysics scheme that diagnoses the liquid cloud fraction and computes the condensation of water vapor and evaporation of liquid condensate which contributes to the changes in LWP. The microphysics scheme computes all the cloud microphysical processes which also affect LWP. The two parameterizations are coupled using 5-minute sub-time-step. The results show that LWP and cloud fraction changes are quite consistent with each other in marine subtropical regions: Figure A below shows that EAMv1_MP produces larger lower tropospheric LWP and larger low cloud fraction in the Sc regions.

[Figure]

Figure A. Lower tropospheric (below 700 hPa) LWP (left; unit = g m$^{-2}$) and lower tropospheric cloud fraction (right; unit = %).

5) L527-534, please provide how EIS is calculated from the model.

Reply: EIS was introduced in (Wood and Bretherton, 2006). We computed the EIS following the CFMIP diagnostics code catalogue (Tsushima et al., 2017). We have added the reference in the manuscript.

6) Fig. 14c, it is a little surprising to see the altered cloud-rain autoconversion does not impact Eaci significantly over the subtropical warm cloud regions. Any explanation?

Reply: In Fig 14, EAMv1_MP does produce a weaker ERFaci. We have also attached the zonal mean plot below, which shows that ERFaci in EAMv1_MP is about 0.5 Wm$^{-2}$ weaker than EAMv1 in the subtropics. However, we note that the EAMv1_MP includes 7 parameter changes as described in Table 3, and the autoconversion change is only one of them. Therefore, Fig 14c shows the effects of all 7 changes combined. We stated in Section 2.4 that "The effects and the mechanisms of each individual parameter adjustment require further investigation and will be documented in separate manuscripts."

[Figure]

Figure B. Zonal mean of ERFaci (Wm$^{-2}$).

7) Fig. 15a, on about the same latitude, why aerosols induce opposite land temperature changes over the northeast Eurasia and the northwest North America?

Reply: This suggests that the surface temperature changes are not determined only by local energy balance. Other processes in the climate system such as large-scale circulation changes also play a role. We have added these sentences in the manuscript.

8) Fig. 14e and L963-965, it is unclear to me why EAMv1_ZM shows less sensitivity of Nc and Ni to aerosols, but produces stronger ERFaci?

Reply: By combining the present-day Nc and Ni in Table 6 and the relative change of Nc and Ni due to anthropogenic aerosols in Table 10, we can derive that EAMv1_ZM produces higher pre-industrial Nc and Ni than EAMv1. EAMv1_ZM also produces a larger Nc increase ($4.79 \times 10^9$ m$^{-2}$) due to anthropogenic aerosols than EAMv1 ($4.58 \times 10^9$ m$^{-2}$). These results are consistent with the stronger ERFaci produced by EAMv1_ZM. We have revised the text for clarification.

9) It may be beyond the scope of the study, but I am curious of the additivity of the impacts of different tuning parts. Would the total impacts from EAMv1P be a linear addition of those from each individual configuration? In other words, are there significant nonlinear interactions among those different configurations?

Reply: We stated in Section 4 that "the recalibrated model produces more improvements than the sum of the improvements from individual intermediate configuration, demonstrating the nonlinearity in the climate system and the necessity of combining all of the improvements that target different biases in different regimes."

10) Near the end of the paper, it is worth discussing what are the unresolved outstanding biases in EAMv1P and whether they are likely to be resolved in the next stage.

Reply: This paper attempts to carefully document the approach, strategy, and results of model calibration, including the strengths and weaknesses of the approach and results. Our tuning target is the climatology of bulk cloud properties such as CRE and cloud fraction and show improvements in many other aspects of the model. Further reducing the model biases by improving parameterizations, numerics, resolution, and calibration is an ongoing effort for the E3SM team. Incorporating process-oriented diagnostics in model development and calibration will be useful for ensuring that the model get the right answer for the right reason. We have added the discussion in Section 4.

**Reference**

Golaz, J. C., Caldwell, P. M., Van Roekel, L. P., Petersen, M. R., Tang, Q., Wolfe, J. D., Abeshu, G., Anantharaj, V., Asay-Davis, X. S., Bader, D. C., Baldwin, S. A., Bisht, G., Bogenschutz, P. A., Branstetter, M., Brunke, M. A., Brus, S. R., Burrows, S. M., Cameron-Smith, P. J., Donahue, A. S., Deakin, M., Easter, R. C., Evans, K. J., Feng, Y., Flanner, M., Foucar, J. G., Fyke, J. G., Griffin, B. M., Hannay, C., Harrop, B. E., Hoffman, M. J., Hunke, E. C., Jacob, R. L., Jacobsen, D. W., Jeffery, N., Jones, P. W., Keen, N. D., Klein, S. A., Larson, V. E., Leung, L. R., Li, H. Y., Lin, W. Y., Lipscomb, W. H., Ma, P. L., Mahajan, S., Maltrud, M. E., Mametjanov, A., McClean, J. L., McCoy, R. B., Neale, R. B., Price, S. F., Qian, Y., Rasch, P. J., Eyre, J. E. J. R., Riley, W. J., Ringler, T. D., Roberts, A. F., Roesler, E. L., Salinger, A. G., Shaheen, Z., Shi, X. Y., Singh, B., Tang, J. Y., Taylor, M. A., Thornton, P. E., Turner, A. K., Veneziani, M., Wan, H., Wang, H. L., Wang, S. L., Williams, D. N., Wolfram, P. J., Worley, P. H., Xie, S. C., Yang, Y., Yoon, J. H., Zelinka, M. D., Zender, C. S., Zeng, X. B., Zhang, C. Z., Zhang, K., Zhang, Y., Zheng, X., Zhou, T., and Zhu, Q.: The DOE E3SM Coupled Model Version 1: Overview and Evaluation at Standard Resolution, J Adv Model Earth Sy, 11, 2089-2129, 10.1029/2018ms001603, 2019.

Tsushima, Y., Brient, F., Klein, S. A., Konsta, D., Nam, C. C., Qu, X., Williams, K. D., Sherwood, S. C., Suzuki, K., and Zelinka, M. D.: The Cloud Feedback Model Intercomparison Project (CFMIP) Diagnostic Codes Catalogue - metrics, diagnostics and methodologies to evaluate, understand and improve the representation of clouds and cloud feedbacks in climate models, Geosci Model Dev, 10, 4285-4305, 10.5194/gmd-10-4285-2017, 2017.

Wood, R., and Bretherton, C. S.: On the relationship between stratiform low cloud cover and lower-tropospheric stability, J Climate, 19, 6425-6432, Doi 10.1175/Jcli3988.1, 2006.